# Modelling hydrologic impacts of light absorbing aerosol deposition on snow at the catchment scale

Felix N. Matt[1], John F. Burkhart[1, 2], and Joni-Pekka Pietikäinen[3]

[1]Department of Geosciences, University of Oslo, Oslo, Norway
[2]Statkraft AS, Norway
[3]Finnish Meteorological Institute, Helsinki, Finland

*Correspondence to:* Felix N. Matt (f.n.matt@geo.uio.no)

**Abstract.** Light absorbing impurities in snow and ice (LAISI) originating from atmospheric deposition enhance the snow melt by increasing the absorption of short wave radiation. The consequences are a shortening of the snow duration due to increased snow melt and, at the catchment scale, a temporal shift in the discharge generation during the spring melt season.

In this study, we present a newly developed snow algorithm for application in hydrolgical models that allows for an additional class of input variables: the deposition mass flux of various species of light absorbing aerosols. To show the sensitivity of different model parameters, we first use the model as 1-D point model forced with representative synthetic data and investigate the impact of parameters and variables specific to the algorithm determining the effect of LAISI. We then demonstrate the significance of the radiative forcing by simulating the effect of black carbon (BC) deposited on snow of a remote south Norwegian catchment over a six years period, from September 2006 to August 2012. Our simulations suggest a significant impact of BC in snow on the hydrological cycle. Results show an average increase in discharge of 2.5 %, 9.9 %, and 21.4 %, depending on the applied model scenario, over a two months period during the spring melt season compared to simulations where radiative forcing from LAISI is not considered. The increase in discharge is followed by a decrease in discharge due to faster decrease of the catchment's snow covered fraction and a trend to earlier melt in the scenarios where radiative forcing from LAISI is applied. Using a reasonable estimate of critical model parameters, the model simulates realistic BC mixing ratios in surface snow with a strong annual cycle, showing increasing surface BC mixing ratios during spring melt as consequence of melt amplification. However, we further identify large uncertainties in the representation of the surface BC mixing ratio during snow melt and the subsequent consequences for the snowpack evolution.

## 1 Introduction

The representation of the seasonal snowpack is of outstanding importance in hydrological models aiming for application in cold or mountainous environments. In many mountain regions, the seasonal snowpack contributes a major portion of the water budget, with a contribution of up to 50 % and more to the annual discharge (e.g., Junghans et al., 2011). Snow melt plays a key role in the dynamic of the hydrology of catchments of various high mountain areas such as the Himalayas (Jeelani et al., 2012), the Alps (Junghans et al., 2011) and the Norwegian mountains (Engelhardt et al., 2014), and is an equally important contributor to stream flow generation as rain in these areas. Furthermore, timing and magnitude of the snow melt are major

predictors for flood (Berghuijs et al., 2016) and land slide (Kawagoe et al., 2009) forecasts, and important factors in water resource management and operational hydropower forecasting. Lastly, the extent and the temporal evolution of the snow cover is a controlling factor in the processes determining the growing-season of plants (Jonas et al., 2008). For all these reasons, a good representation of the seasonal snowpack in hydrological models is paramount. However, there are large uncertainties in many variables specifying the temporal evolution of the snowpack, and the snow albedo is one of the most important among those due to the direct effect on the energy input to the snowpack from solar radiation (Anderson, 1976). Fresh snow reflects most of the incoming solar radiation in the near UV and visible spectrum (Warren and Wiscombe, 1980). However, as snow ages and snow grain size increases, the snow albedo will drop as a result of the altered scattering properties of the larger snow grains (Flanner and Zender, 2006). Furthermore, ambient conditions also play a large role. The ratio of diffuse and direct incoming shortwave radiation, the zenith angle of the sun, and the albedo of the underlying ground in combination with the snow thickness can have a large impact on the snow albedo (Warren and Wiscombe, 1980). Of recent significance is the role light absorbing impurities, or particles, which absorb in the range of the solar spectrum, have on albedo when present in the snowpack (e.g., Flanner et al., 2007; Painter et al., 2007; Skiles et al., 2012). These light absorbing impurities in snow and ice (LAISI) can originate from fossil fuel combustion and forest fires in the form of black carbon, BC, and organic carbon (Bond et al., 2013; AMAP, 2015), mineral dust (Painter et al., 2012), volcanic ash (Rhodes et al., 1987), organic compounds in soils (Wang et al., 2013), and biological activity (Lutz et al., 2016), and have species-specific radiative properties.

As LAISI lower the snow albedo, the effect on the snow melt has the potential to alter the hydrological characteristics of catchments where snow melt significantly contributes to the water budget. Recent research investigates the impact of LAISI on discharge generation in mountain regions on different scales. Qian et al. (2011) used a global climate model to simulate the effect black carbon and dust in snow have on the hydrological cycle of the Tibetan Plateau and found a significant impact on the hydrology, with runoff increasing during late winter/early spring and decreasing during late spring/early summer due to a trend to earlier melt dates. Oaida et al. (2015) showed by implementing radiative transfer calculations to determine snow albedo in the Simple Simplified Biosphere (SSiB) land surface model implementation of the Weather Research and Forecasting (WRF) regional climate model that physically based snow albedo representation can be significantly improved by considering the deposition of light absorbing aerosols in the snowpack evolution. Qian et al. (2009) simulated hydrological impacts due to BC deposition in the western United States using WRF coupled with chemistry (WRF-Chem). They found a decrease in net snow accumulation and spring snowmelt due to BC-in-snow induced increase in surface air temperature.

Only a few studies developed model approaches to resolve the impact of LAISI on the snow melt discharge generation at the catchment scale. Painter et al. (2010) showed that dust, transported from remote places to the Colorado river basin, can have severe implications on the hydrological regime due to disturbances to the discharge generation from snow melt during the spring time, shifting the peak runoff in spring by several weeks and leading to earlier snow free catchments and a decrease in annual runoff. Kaspari et al. (2015) simulated the impact of BC and dust in snow on glacier melt on Mount Olympus, USA, by using measured concentrations in summer horizons and determining the radiative forcing via a radiative transfer model. Results indicate enhanced melt during a year of heavy nearby forest fires, coinciding with an increase of observed discharge from the catchment.

Despite these efforts, the direct integration of deposition mass fluxes of light absorbing aerosols in a catchment model is still lacking. To date, there is no rainfall-runoff model with focus on runoff forecast at the catchment scale that is able to consider aerosol deposition mass fluxes alongside snowfall. On the other hand, there is evidence that including the radiative forcing of LAISI has the potential to further the quality of hydrological predictions: Bryant et al. (2013) showed that during the melt pe-
riod errors in the operational stream flow prediction of the National Weather Service Colorado Basin River Forecast Center are linearly related to dust radiative forcing in snow and concluded that implementing the effect of LAISI on the snow reflectivity could improve hydrological predictions in regions prone to deposition of light absorbing aerosols on snow, which emphasizes the need for the development of a suitable model approach. Furthermore, we continuously move toward hydrological models with a increasing complex representation of the physical processes involved in the evolution of the seasonal snowpack. Hereto-
fore there has been little focus on the factors related to LAISI, such as the impact to albedo due to the deposition of aerosols, that may alter the timing and character of discharge generation at the catchment scale.

In this study we address this deficiency by introducing a rainfall-runoff model with a newly developed snow algorithm that allows for a new class of model input variables: the deposition mass flux of different species of light absorbing aerosols. The model integrates snowpack dynamics forced by LAISI and allows for analysis at the catchment scale. The algorithm uses a radiative transfer model for snow to account dynamically for the impact of LAISI on the snow albedo and the subsequent impacts on the snow melt and discharge generation. Aside from enabling the user to optionally apply deposition mass fluxes as model input, the algorithm depends on standard atmospheric input variables (precipitation, temperature, short wave radiation, wind speed, and relative humidity). To enable a critical evaluation of the newly developed snowpack algorithm, we conduct two independent analyses: i) a 1-D sensitivity study of critical model parameters, and ii) a catchment scale analysis of the impact of LAISI. In both analysis we use BC in snow from wet and dry deposition as a proxy for the impact of LAISI.

We first present an overview over the hydrological model used in this study and the newly developed snow algorithm to treat LAISI in Sect. 2. A description of the catchment used for our study and the input data sets is given in Sect. 3. Sect. 4 describes the 1-D model experiments and the model settings in the case study. Lastly, our results are presented in Sect. 5 and discussed in Sect. 6.

## 2 Modelling framework and snowpack algorithm

In the following section we provide descriptions of the hydrologic model (Sect. 2.1) and the formulation of a novel snowpack module used for the analyses (Sect. 2.2).

### 2.1 Hydrologic Model Framework

For the analysis, we use Statkraft's hydrologic forecasting toolbox (Shyft; https://github.com/statkraft/shyft), a model frame-
work developed for hydropower forecasting (Burkhart et al., 2016; Ghimirey, 2016; Westergren, 2016). Shyft provides the implementation of many well-known hydrological routines (conceptual parameter models, and more physically based ap-

proaches), and allows for distributed hydrological modelling. Standard model input variables are temperature, precipitation, wind speed, relative humidity, and shortwave radiation.

The methods used herein to simulate hydrological processes are (i) a single-equation implementation to determine the potential evapotranspiration, (ii) a newly developed snowpack algorithm using an online radiative transfer solution for snow to account for the effect of LAISI on the snow albedo, and (iii) a first order nonlinear differential equation to calculate the catchment response to precipitation, snow melt and evapotranspiration. (i) and (iii) are described in more detail herein, while (ii) is described in detail in Sect. 2.2.

To determine the potential evapotranspiration, $E_{pot}$, we use the method according to Priestley and Taylor (1972)

$$E_{pot} = \frac{a}{\lambda} \cdot \frac{s(T_a)}{s(T_a) + \gamma} \cdot R_n \qquad (1)$$

with $a = 1.26$ being a dimensionless empirical multiplier, $\gamma$ the psychrometric constant, $s(T_a)$ the slope of the relationship between the saturation vapour pressure and the temperature $T_a$, $\lambda$ the latent heat of vaporization and $R_n$ the net radiation.

The catchment response to precipitation and snow melt is determined using the approach of Kirchner (2009), who describes catchment discharge from a simple first order nonlinear differential equation. Following Kirchner (2009), we solve the log-transformed formulation

$$\frac{d(ln(Q))}{dt} = g(Q)(\frac{P - E}{Q} - 1) \qquad (2)$$

due to numerical instabilities of the original formulation. In Eq. (2), $Q$ is the catchment discharge, $E$ the evapotranspiration, and $P$ the precipitation.

We assume that the sensitivity function, $g(Q)$, has the same form as described in Kirchner (2009):

$$ln(g(Q)) \approx c_1 + c_2 ln(Q) + c_3 (ln(Q))^2 \qquad (3)$$

with $c_1$, $c_2$ and $c_3$ being the only catchment specific parameters, which we estimate by standard model calibration of simulated discharge against observed discharge. In contrast to Kirchner (2009)'s approach, we use the liquid water response from the snow routine instead of precipitation $P$ in Eq. (2) (Kirchner, 2009 used snow-free catchments). The response from the snow routine can be liquid precipitation, melt water, or a combination of both.

## 2.2   A new snowpack module for LAISI

To account for snow in the model, we developed a snow-algorithm to solve the energy balance

$$\frac{\delta F}{\delta t} = K_{in}(1 - \alpha) + L_{in} + L_{out} + H_s + H_l + R \qquad (4)$$

with the incoming shortwave radiation flux $K_{in}$, the incoming and outgoing longwave radiation fluxes $L_{in}$ and $L_{out}$, the sensible and latent heat fluxes $H_s$ and $H_l$, and the heat contribution from rain $R$. $\frac{\delta F}{\delta t}$ is the net energy flux into or out of the snowpack. Fluxes are considered to be positive when directed into the snowpack and as such an energy source.

$L_{in}$ and $L_{out}$ are calculated using the Stephan-Boltzmann law, with $L_{in}$ depending on the air temperature $T_a$ and $L_{out}$ on the snow surface temperature $T_{ss}$, calculated as $T_{ss} = 1.16 \cdot T_a - 2.09$ (Hegdahl et al., 2016). The latent and sensible heat fluxes are calculated using a bulk-transfer approach that depends on wind speed, temperature, and relative humidity (Hegdahl et al., 2016).

The main addition provided in the algorithm described herein is the implementation of a radiative transfer solution for the dynamical calculation of snow albedo, $\alpha$. The implementation allows a new class of model input variables, wet and dry deposition rates of light absorbing aerosols. From this, the model is able to simulate the impact of dust, black carbon, volcanic ash, or other aerosol deposition on snow albedo, snow melt, and runoff. To account for the mass balance of LAISI, while maintaining a representation of sub-grid snow variability and snow cover fraction (SCF), a tiling approach is applied, where a

grid-cell's snowfall is apportioned to sub-grid units. Energy-balance calculations are then conducted within each tile. Currently, a gamma distribution is used to distribute snowfall to the tiles.

     Below, we introduce the radiative transfer calculations required to represent LAISI (Sect. 2.2.1), and provide further details of the sub-gridscale tiling approach to represent snowpack spatial variability (Sect. 2.2.2).

### 2.2.1    Aerosols in the snowpack

Wiscombe and Warren (1980) and Warren and Wiscombe (1980) developed a robust and elegant model for snow albedo that remains today as a standard. Critical to their approach was the ability to account for: (i) wide variability in ice absorption with wavelength, (ii) the forward scattering of snow grains, and (iii) both diffuse and direct beam radiation at the surface. Furthermore, and of particular importance to the success of the approach, the model relies on observable parameters.

     Both the albedo of clean snow and the effect of LAISI on the snow albedo strongly depend on the snow optical grain radius

$r$ (Warren and Wiscombe, 1980), which alters as snow ages. $r$ can be related to the specific surface area $A_s$ via

$$r = \frac{3}{\rho_{ice} \cdot A_s}, \tag{5}$$

with $\rho_{ice}$ the density of ice. $A_s$ represents the ratio of surface area per unit mass of the snow grain (Roy et al., 2013).

     In our model, we compute the evolution of $A_s$ in dry snow following Taillandier et al. (2007) as

$$
\begin{aligned}
\quad A_s(t) = [0.629 \cdot A_{s,0} - 15.0 \cdot (T_s - 11.2)] - [0.076 \cdot A_{s,0} - 1.76 \cdot (T_s - 2.96)] \\
\ln\left\{ t + \exp\left( \frac{-0.371 \cdot A_{s,0} - 15.0 \cdot (T_s - 11.2)}{0.076 \cdot A_{s,0} - 1.76 \cdot (T_s - 2.96)} \right) \right\}, \quad (6)
\end{aligned}
$$

where $t$ is the age of the snow layer (hours), $A_{s,0}$ is $A_s$ at $t=0$ (cm$^2$ g$^{-1}$), and $T_s$ is the snow temperature (°C). The evolution of $A_s$ in wet snow is calculated according to Eq. 5 and Brun (1989) as

$$\Delta r = \frac{C_1 + C_2 \cdot \Theta^3}{r^2 \cdot 4\pi}, \tag{7}$$

where $C_1 = 1.1 \cdot 10^{-3}$ mm$^3$ d$^{-1}$ and $C_2 = 3.7 \cdot 10^{-5}$ mm$^3$ d$^{-1}$ are empirical coefficients. $\Theta$ is the liquid water content of snow in mass percentage. $A_{s,0}$ is set to 73.0 m$^2$ kg$^{-1}$ (Domine et al., 2007) and we set the minimum snowfall required to reset $A_s$ to 5 mm snow water equivalent (SWE).

To solve for the effect of light absorption of LAISI on the snow albedo, we have integrated a 2-layer adaption of the Snow, Ice, and Aerosol Radiative (SNICAR) model (Flanner et al., 2007, 2009) into the energy and mass budget calculations. By providing the solar zenith angle of the sun, the snow optical grain radius $r$, mixing ratios of LAISI in the snow layers and SWE of each layer, SNICAR calculates the snow albedo for a number of spectral bands. To achive this, SNICAR utilizes

the theory from Wiscombe and Warren (1980) and the two-stream, multilayer radiative approximation of Toon et al. (1989). Following Flanner et al. (2007), our implementation of SNICAR uses five spectral bands (0.3-0.7, 0.7-1.0, 1.0-1.2, 1.2-1.5, and 1.5-5.0 $\mu$m) in order to maintain computational efficiency. Flanner et al. (2007) compared results from the 5 bands scheme to the default 470 bands scheme in SNICAR and concluded that relative errors are less than 0.5%. The incident flux were simulated offline assuming mid-latitude winter clear- and cloudy-sky conditions.

The absorbing effect of LAISI is most efficient when the LAISI reside at or close to the snow surface (Warren and Wiscombe, 1980). As snow melts LAISI can remain near the surface due to inefficient melt scavenging, which leads to an increase in the near surface concentration of LAISI and thus a further decrease in the snow albedo - the so called melt amplification (e.g., Xu et al., 2012; Doherty et al., 2013; Sterle et al., 2013; Doherty et al., 2016). Field observations suggest that the magnitude of this effect is determined by the particle size and the hydrophobicity of the respective LAISI (Doherty et al., 2013). Conway

et al. (1996) observed vertical redistribution and the effect on the snow albedo by adding volcanic ash and hydrophilic and hydrophobic BC to the snow surface of a natural snowpack. Flanner et al. (2007) used the results from Conway et al. (1996) to determine the scavenging ratios, specifying the ratio of LAISI contained in the melting snow that is flushed out with melt water, of both hydrophilic and hydrophobic BC. They found the scavenging ratio for hydrophobic BC, $k_{phob}$, to be 0.03, and for hydrophilic BC, $k_{phil}$, 0.2. Doherty et al. (2013) found similar results by observing BC mixing ratios close to the surface of

melting snow. However, more recent studies report efficient removal of BC with melt water (Lazarcik et al., 2017), revealing large gaps in the understanding of the process.

To represent the evolution of LAISI mixing ratios near the snow surface, we treat LAISI in two layers in our model. The surface layer has a time invariant maximum thickness (further called maximum surface layer thickness). The mixing ratio of each LAISI species in this layer is calculated from a uniform mixing of the layer's snow with either falling snow with

a certain mixing ratio of aerosol (wet deposition), or aerosol from atmospheric dry deposition. The second layer (bottom layer) represents the snow exceeding the maximum thickness of the surface layer. Following Krinner et al. (2006), we apply a maximum surface layer thickness of 8 mm SWE. Krinner et al. (2006) suggests this value based on observations of 1 cm thick dirty layers in alpine firn cores used to identify summer horizons. Due to potential accumulation of LAISI in surface snow via dry deposition and melt amplification, we expect the simulated surface mixing ratios of LAISI to be sensitive to the maximum

surface layer thickness of our model. For this reason, we use a factor of 2 to the maximal surface layer thickness to account for the uncertainty of this model parameter.

To allow for melt amplification in the model, we include LAISI mass fluxes between the two layers during snow accumulation and snow melt. Generalizing Jacobson (2004)'s representation of LAISI mass loss due to meltwater scavenging for

multiple snow layers, we characterize the magnitude of melt scavenging using the scavenging ratio $k$ and calculate the temporal change of LAISI mass $m_s$ in the surface layer as

$$\frac{dm_s}{dt} = -kq_sc_s + D, \tag{8}$$

and the change of LAISI mass $m_b$ in the bottom layer as

$$\frac{dm_b}{dt} = k(q_sc_s - q_bc_b). \tag{9}$$

Herein, $q_s$ and $q_b$ are the mass fluxes of melt water from the surface to the bottom layer and out of the bottom layer, respectively, and $c_s$ and $c_b$ are the mass mixing ratios of LAISI in the respective layer. $D$ is the atmospheric deposition mass flux. A value for $k$ of <1 is equal to a scavenging efficiency of less than 100% and hence allows for accumulation of LAISI in the surface layer during melt. In our analysis, we account for hydrophobic and hydrophilic BC. By following Flanner et al. (2007), we set $k_{phob}$ to 0.03 and $k_{phil}$ to 0.2, and account for the large uncertainty in those estimates by using an order of magnitude variation on $k_{phob}$ and $k_{phil}$. Like Flanner et al. (2007), we treat aged, hydrophilic BC as sulphate coated to account for the net increase in the mass absorption cross section (MAC) by 1.5 at $\lambda$=550 nm compared to hydrophobic BC caused by the ageing of BC (reducing effect on MAC) and particle coating from condensation of weakly absorbing compounds (enhancing effect on MAC) suggested by Bond et al. (2006). As a consequence, hydrophilic BC absorbs stronger than hydrophobic BC under the same conditions. On the other hand, hydrophilic BC undergoes a more efficient melt scavenging. The competing mechanisms are subjects of the 1-D sensitivity study in Sect. 5.1.3.

### 2.2.2 Sub-grid variability in snow depth and snow cover

In order to allow for explicit treatment of snow layers while representing sub-grid snow variability, we follow (Aas et al., 2017) and assume that the sub-grid spatial distribution of each single event of solid precipitation follows a certain probability distribution function. From this distribution we calculate multiplication factors, which then are used to assign the snowfall of a model grid cell to a number of sub-grid computational elements, the so called tiles (Aas et al., 2017). The snow algorithm described herein is executed for each of the tiles separately, providing a mechanism to account for snow spatial distribution while preserving conservation of mass. Therefore, variables related to the snow state, such as SWE, liquid water content, LAISI mixing ratios, and snow albedo differ among the tiles. To calculate the multiplication factors, we assume that the sub-grid redistributed snow follows a gamma distribution (see e.g., Kolberg and Gottschalk, 2010; Gisnås et al., 2016), determined by the coefficient of variation (CV) of SWE at snow maximum. Gisnås et al. (2016) used Winstral and Marks (2002)'s terrain-based parametrization to model snow redistribution in Norway by accounting for wind effects during the snow accumulation period over a digital elevation model with 10 m resolution. In the case study presented in Sect. 5.2, we use the CV values from Gisnås et al. (2016) to derive a linear relationship between the model grid cell's elevation and the corresponding CV value by simple linear regression (see Fig.1a), which results in a $R^2$-value of 0.71 and a p-value of smaller than 2.0e-5 for the study

area. The linear relationship is only applied to grid cells with an areal forest cover fraction of lower than or equal to 0.5. For grid cells with a forest cover fraction of higher than 0.5, a constant snow CV value of 0.17 is used, following the findings of Liston (2004) for high latitude, mountainous forest. Examples of multiplication factors for forested grid cells and forest free grid cells for different CV values are shown in Fig. 1b.

## 3 Site description, meteorologic model input, and atmospheric deposition data

We selected the unregulated upper Atna catchment for our analysis. The catchment is located in a high elevation region of southern Norway (Fig. 2). The watershed covers an area of 463 km$^2$ and ranges in elevation from 700 masl at the outlet at lake Atnsjoen to over 2000 masl in the Rondane mountains in the western part of the watershed, with approximately 90 % of the area above the forest limit. The average annual precipitation in the watershed during the study period is approximately 655 mm. The mean annual discharge is approximately 11 m$^3$s$^{-1}$, with low flows of 1-3 m$^3$s$^{-1}$ during the winter months and peak flows of over 130 m$^3$s$^{-1}$ during the spring melt season.

For the meteorological model input of precipitation, temperature, relative humidity, and wind speed we use daily observations from the Norwegian Water Resources and Energy Directorate (NVE) and the Norwegian Meteorological Institute (MET). Four meteorological stations are located in the watershed at elevations between 701 and 780 masl along the Atna river, two of these measuring precipitation and two measuring temperature. Observations of relative humidity and wind speed originate from two stations at locations close by the catchment (not shown in Fig. 2). Further information about the stations are given in Table 1. Due to poor availability of continuous solar radiation observations in Norway, we use gridded global radiation data from the Water and Global Change (WATCH) Forcing Data methodology applied to ERA-Interim reanalysis data (WFDEI; Weedon et al., 2014) with a resolution of 0.5°. Discharge observations are from a station located at the outlet of the catchment at lake Atnsjoen and are used for model calibration and validation. In the following section (3.1) we present the development of atmospheric deposition rates of BC, which we use as a proxy for LAISI, due to a lack of available deposition rates for other species. For the 1-D sensitivity study of Sect. 5.1 we developed representative model input based on the meteorological conditions in this catchment.

### 3.1 Atmospheric deposition of black carbon from the REMO-HAM model

The wet and dry deposition rates of BC for the study area are generated using the regional aerosol-climate model REMO-HAM (Pietikäinen et al., 2012). The core of the model is a hydrostatic, three-dimensional atmosphere model developed at the Max Planck Institute for Meteorology in Hamburg. With the aerosol configuration, the model incorporates the HAM (Hamburg Aerosol Module) by Stier et al. (2005) and Zhang et al. (2012). HAM calculates the aerosols distributions using 7 log-normal modes and includes all the main aerosol processes.

For the simulations, we follow the approach of Hienola et al. (2013), but with changes to the emission inventory: Hienola et al. (2013) used emissions based on the AeroCom emission inventory for the year 2000 (see Dentener et al., 2006). In the REMO-HAM simulations conducted herein, emissions are made by the International Institute for Applied Systems Analysis

(IIASA) and are based on the Evaluating the Climate and Air Quality Impacts of Short-Lived Pollutants (ECLIPSE) V5a inventory for the years 2005, 2010, and 2015 (years in between were linearly interpolated) (Klimont et al., 2016a, b). We updated also other emissions modules (wildfire, aviation, and shipping) following the approaches presented in Pietikäinen et al. (2015). The only difference to Pietikäinen et al. (2015) in this work is that we used the Global Fire Emissions Database
(GFED) version 4 based on an updated version of van der Werf et al. (2010).

REMO-HAM was used for the same European domain as in Pietikäinen et al. (2012) using 0.44° spatial resolution (50 km), 27 vertical levels and 3 minutes time step. The ERA-Interim re-analysis data was utilized at the lateral boundaries for meteorological forcing (Dee et al., 2011) and for the lateral aerosol forcing, data from the global aerosol-climate model ECHAM-HAMMOZ (version echam6.1.0-ham2.2) was used. ECHAM-HAMMOZ was simulated in a nudging mode, i.e. the model's
meteorology was forced to follow ERA-Interim data, and the ECLIPSE emissions were used (plus other updated emission modules shown in Pietikäinen et al. (2015)). The boundaries of REMO-HAM were updated every 6 hours for both meteorological and aerosol related variables. Simulations with REMO-HAM were conducted for the time period of 01.07.2004 - 31.08.2012 and the time period used in the analysis herein is from 01.09.2006 onwards. The initial state for the model was taken from the boundary data, except for the soil parameters which were taken from a previous long-term simulation for the
same domain (a so called warm-start). The output frequency of REMO-HAM was 3 hours and the total BC deposition flux was calculated from the accumulated dry and wet deposition and sedimentation fluxes, and resampled to daily time resolution. Herein, dry deposition refers to the sum of REMO-HAM dry deposition and sedimentation.

## 4   Modelling experiments and calibration

Our analysis is conducted in two parts. First, in a 1-D sensitivity study, we investigate the sensitivity of parameters and variables
specific to the LAISI algorithm presented in Sect. 2.2. We then demonstrate the impact of BC at the catchment scale in a case study by simulating the impact of wet and dry deposition of BC on snow melt and discharge generation in a remote south Norwegian catchment (Sect. 5.2).

We assume uncertainties of the LAISI radiative forcing in snow to originate mainly from the model representation of surface layer thickness, melt scavenging of BC, and uncertainties in the deposition input data. To account for the uncertainties, we
declare minimum (min), central (mid), and maximum (max) effect estimates to each of the critical parameters, outlined together with further model parameters in Table 2. The min, mid, and max estimates are both subjects of analysis in the sensitivity study (further described in Sect. 4.1) and used in the case study to give an uncertainty estimate of the LAISI effect on the hydrologic variables (further described in Sect. 4.2). We investigate the impact of BC impurities on the response variables by comparing the results from Aerosol Radiative Forcing model experiments ("ARF" scenarios) to simulations in which all BC deposition
rates are set to zero ("no-ARF" scenario).

## 4.1 1-D sensitivity study experiments

The results of the 1-D sensitivity study are presented in Sect. 5.1, herein we describe the configurations to conduct our analysis. The purpose of the study is to isolate the impact of different model parameters: (i) maximum surface layer thickness (parameter *max_surface_layer*; see Table 2), (ii) scavenging ratio, and (iii) the impact of the scavenging ratio with respect to the BC species (parameters $k_{phob}$ and $k_{phil}$).

Our approach evaluates these parameters and the evolution of the snowpack under constant melting conditions. We run the 1-D simulations with model parameters as outlined in Table 2 and forcing data based on synthetic input data. The synthetic forcing data set is based on the average meteorological conditions during the melt season from mid March until mid July of the Atnsjoen catchment. In our sensitivity experiments, all snowpacks have 250 mm SWE of snow with a mixing ratio of 35 ng g$^{-1}$ in both surface and bottom layer at melt onset. These values are representative of the upper 50% of tiles at winter snow maximum in the Atnsjoen catchment during the study period of the case study. During the melt period, we exclude fresh snowfall and dry deposition, in order to isolate the effect of the tested model parameters on the snowpack evolution under melt conditions. This may lead to an underestimation of total BC mass in the snow column.

To investigate the impact of the maximum surface layer thickness (parameter *max_surface_layer*) of the model, we run simulations with synthetic forcing and use maximal surface layer thicknesses of 4.0 mm SWE (max estimate, see Tabel 2), 8.0 mm SWE (mid estimate), 16.0 mm SWE (min estimate). Additionally we include a single layer model with a vertically uniform distribution of BC in the analysis and for comparison a simulation with clean snow.

To explore the sensitivity to scavenging ratio, we apply different BC scavenging ratios in the range of the uncertainty of hydrophilic BC, which covers a wide range from very sufficient scavenging to inefficient scavenging. The scavenging ratios applied are based on the analysis conducted by Flanner et al. (2007) using data from Conway et al. (1996). The mid estimate for the hydrophilic BC scavenging ratio ($k_{phil}$=0.2) also compares well to field observations from Doherty et al. (2013). We further include in the analysis Flanner et al. (2007)'s upper bound uncertainty estimate for hydrophilic BC (2.0; efficient scavenging), the lower bound estimate (0.02; inefficient scavenging), and for comparison a scenario in which BC does not undergo any scavenging (0.0).

Hydrophilic BC absorbs stronger than hydrophobic BC under the same conditions due to an increased MAC for hydrophilic BC resulting from ageing of the aerosol during atmospheric transport (Bond et al., 2006). On the other hand, hydrophilic BC undergoes more efficient melt scavenging (Flanner et al., 2007), which impacts the snowpack evolution significantly. To explore this competing interplay we apply the mid estimate of the scavenging ratio of hydrophobic BC ($k_{phob}$=0.03) to both the hydrophobic BC and the hydrophilic BC species. In this manner we explore the isolated effect of the different absorption properties of the two species. We further apply the mid estimate for hydrophilic BC scavenging ratio ($k_{phil}$=0.2) to hydrophilic BC to quantify the gross effect. As in other cases, we include the no-ARF scenario to highlight the overall effect on the albedo and melt of the different scenarios.

## 4.2 Case study model setup and calibration

We investigate the impact of BC aerosol deposition on the catchment hydrology of a Norwegian catchment over a study period of 6 years, from September 2006 to August 2012. The station based input data described above (Sect. 3) is interpolated to the simulation grid cells (1x1 km$^2$ and accordingly smaller cells at the catchment boarders; see Fig. 2) using Shyft's interpolation

algorithms. For temperature Bayesian Kriging (Diggle and Ribeiro, 2007) is used. For precipitation, BC deposition rates, wind speed, and relative humidity interpolation to the model grid cells is via inverse distance weighting. A 5% increase in precipitation for every 100 m increase in altitude is used for the precipitation interpolation (Førland, 1979).

To calibrate the model against observed discharge, we first run a split-sample calibration (Klemes, 1986) using the first 3 years (1 September 2006 to 31 August 2009) of the study period as calibration period and the following 3 years (1 September

2009 to 31 August 2012) for model validation. For parameter estimation, we use the BOBYQA algorithm for bound constrained optimization (Powell, 2009). To asses the predictive efficiency of the model we use the Nash-Sutcliffe model efficiency (NSE)

$$E_{NS} = 1 - \frac{\sum_{t=0}^{T}(Q_o^t - Q_s^t)^2}{\sum_{t=0}^{T}(Q_o^t - \overline{Q_o})^2}, \tag{10}$$

where $Q_o^t$ and $Q_s^t$ are the observed and simulated discharge at time t, respectively, and $\overline{Q_0}$ is the mean observed discharge over the assessed period.

Model calibration is run with mid estimates for all model parameters impacting the handling and effect of LAISI and aerosol depositions as simulated from REMO-HAM during model calibration. Those parameters and further model parameters, including the parameters estimated during calibration, are listed in the left column of Table 2. We investigate the uncertainty in the effect of BC on snow melt by using the min and max effect parameter estimates from Table 2, while holding constant all other model parameters as estimated during calibration. To assess the gross effect of LAISI we compare the simulations to

equivalent simulations in which ARF is not included.

## 5 Results

In Sect. 5.1 we present the results from our sensitivity study based on the newly developed snow algorithm as a single point model. The results of the case study are presented in Sect. 5.2, where we examine the significance of the LAISI radiative forcing for hydrological processes by simulating the impact of BC deposition on the snow melt and discharge generation in a

25 snow dominated mountain catchment.

### 5.1 1-D sensitivity studies

#### 5.1.1 Sensitivity to surface layer thickness

Fig. 3a shows the effect of the different maximum surface layer thicknesses (parameter *max_surface_layer*) on the melting snowpack with other parameters set according to Table 2. The maximum surface layer thickness strongly determines the surface

BC mixing ratio over the melt season. During snow melt, surface BC increases up to a factor of circa 10, 20, and about 30 for maximum surface layer thicknesses of 16.0 mm SWE, 8.0 mm SWE, and 4.0 mm SWE, compared to the pre-melt season BC mixing ratio (35 ng g$^{-1}$). For those three 2-layer scenarios (green, purple and red curves in Fig. 3a), the resulting differences in albedo and melt rate are small, even though the increase in surface layer mixing ratio during the melt season differs strongly among the scenarios. Using the single layer model, the surface BC mixing ratio increases slower and stays comparably low in contrast to the 2-layer models until shortly before meltout. This leads to a less pronounced decrease of albedo compared to the 2-layer models and thus to a shorter meltout shift compared to a clean snowpack of about 5 days (yellow curves in Fig. 3a), whereas the 2-layer scenarios show earlier meltouts of about 7 days.

### 5.1.2 Sensitivity to scavenging ratio of BC

In the range of investigated scavenging ratios, we find sensitivity of the surface BC mixing ratio, the albedo, and the subsequent snow melt to this parameter (Fig. 3b). When applying a melt scavenging factor typical for the lower bound of hydrophilic BC (0.02, purple lines) there is little effect compared to the scenario without melt scavenging (green lines). Both show circa a factor 30 increase in surface BC concentration to the end of the melt season and only little differences in the development of albedo and snow melt. Similar results are achieved when using the mid estimate scavenging factor for hydrophobic BC (0.03, not shown). A distinction exists when using the mid estimate scavenging factor for hydrophilic BC (0.2, red line). In contrast to no scavenging and the lower bound hydrophilic scavenging, surface BC does not increase as rapidly during the melt period and in fact is completely flushed when applying a melt scavenging factor typical for the upper bound of hydrophilic BC (yellow line, the surface concentration drops continuously during the melt period).

The changes in the scavenging ratio lead to a considerable effect on the albedo and the snow melt. Meltout is delayed by circa 0.5 (purple lines), 3 (red lines), and 8 days (yellow lines) for scavenging ratios of 0.03, 0.2, and 2.0, respectively, compared to no scavenging (green lines). Compared to the no-ARF experiment (black lines), the presence of BC causes an earlier meltout of circa 9.5, 7, and 2 days for scavenging ratios of 0.03, 0.2, and 2.0, respectively, in our simulation.

### 5.1.3 Sensitivity to BC species

The column of graphs in Fig. 3c illustrate the net effect of the competing processes of more efficient absorption resulting from a larger MAC with more efficient wash out. A mid estimate of the scavenging ratio of hydrophobic BC (0.03) is applied and shown for the hydrophobic BC (green curve) and the hydrophilic BC (purple curves) species. These curves show the isolated effect of the different absorption properties of the two species. Further, the mid estimate scavenging ratio for hydrophilic BC (0.2) is also shown using radiative properties of hydrophilic BC to quantify the gross effect (red curves). The no-ARF scenario (black curves) highlights the overall impacts.

The isolated effect of the stronger absorption of hydrophilic BC leads to an earlier meltout by circa two days compared to hydrophobic BC (purple and green curves in graphs of Fig. 3c). However, when applying the mid estimate of the scavenging ratio for hydrophilic BC (0.2), the combined effects leads to a masking of the isolated effect of stronger absorption by hydrophilic BC (and vice versa). During the melt period, snow albedo, melt rate and the snowpack SWE barely differ between the scenarios

with the mid estimate scavenging for hydrophobic and hydrophilic BC applied (red and green curves). This reveals that both scenarios, hydrophobic BC with low scavenging efficiency and hydrophilic BC with high scavenging efficiency, lead roughly to an earlier meltout by circa 6 days compared to the no-ARF scenario.

## 5.2 Case study: Impact of BC deposition on the hydrology of a south Norwegian catchment

### 5.2.1 Performance of the model

In the split-sample test, the model performance is acceptable during both calibration and validation, with NSEs of 0.86 during the calibration period (green line in Fig. 4a) and 0.82 during the validation period (red line in Fig. 4a). However, in the winter season (circa November until March) the model generally underestimates the discharge and peaks in the beginning of the melt season are slightly underestimated. The scatter plot in Fig. 5 confirms the underestimation of low flow situations. For the different scenarios explored within the case study, all LAISI-relevant parameters are fixed to mid estimates and model parameters optimized for the full period (1 September 2006 to 31 August 2012; Fig. 4b) resulting in a NSE of 0.84. The optimized parameters are listed in Table 2. Note that switching ARF off entirely (no BC deposition) leads to a slight decrease of the model quality (NSE of 0.83 over the whole period; not shown).

### 5.2.2 Evolution of surface BC mixing ratio

For the min and mid estimate, the model simulates an average annual surface BC mixing ratio of about 18 ng g$^{-1}$ and 71 ng g$^{-1}$, respectively. Our max estimate yields 198 ng g$^{-1}$. The evolution of surface albedo driven by BC deposition is distinct in the accumulation period vs. the melt period. During the snow accumulation period (circa until end of March), only slight differences in albedo are noticeable. The average annual snow albedo from January $1^{st}$ until March $22^{nd}$ is 0.871 for the no-ARF scenario (Fig. 6a), while during the same time period, min, mid, and max estimates show relative albedo reductions of 0.003, 0.010, and 0.014, respectively from the no-ARF case. At the beginning of the melt period, surface layer concentrations of min, mid, and max estimate average to 12, 49, and 98 ng g$^{-1}$ (Fig. 6b).

With the start of the melt season, the difference in albedo between model experiments becomes increasingly larger over time. During the melt season, the mid estimate spatially averaged surface BC mixing ratio increases from 49 ng g$^{-1}$ to about 250 ng g$^{-1}$ (factor 5 increase) at the end of the melt season (beginning of July). For the max estimate, the increase is from roughly 100 ng g$^{-1}$ to over 2500 ng g$^{-1}$ (factor 25 increase). The min estimate on the other hand leads to a decrease in BC surface mixing ratio. The distinctly different evolution of surface BC in snow at the end of the melt season and among the three scenarios causes large differences in albedo decrease relative to the no-ARF case of about 0.03, 0.1 and over 0.3 for the min, mid, and max estimate, respectively.

### 5.2.3 BC induced radiative forcing

The radiative forcing in snow (RFS) induced by the presence of BC is calculated from the average radiative forcing over snow bearing tiles only. The RFS represents the additional uptake of energy from solar radiation per area snow cover due

to the presence of BC in the snow compared to clean snow with the same properties. Fig. 7a shows the daily mean RFS and demonstrates the increase of RFS during snow melt. Low RFS is observed during the snow accumulation period then steadily increasing through spring snow melt, reaching values of approximately 8, 18, and 57 $Wm^{-2}$ for the min, mid, and max estimates, respectively (see red solid line and shaded area in Fig. 7a). RFS in mid winter is small due low surface BC mixing

ratios and low solar irradiate.

However, most relevant for discharge generation (see Sect. 5.2.4), is the catchment-wide total daily energy uptake due to BC, or surface radiative forcing, calculated as the mean radiative forcing over all grid cells. As the snow cover fraction (SCF) in the catchment drops during spring (dotted line and yellow shaded area in Fig. 6 and 7), the effect of the RFS on the melt generation is limited by the increasing area of bare ground. The net effect is shown in Fig. 7b. The catchment mean surface

radiative forcing due to the presence of BC in snow shows a strong annual cycle and reaches a maximum of 1.3, 4.9, and 8.8 $Wm^{-2}$ (min, mid, and max estimates, respectively) around the beginning of May.

### 5.2.4   BC impact on catchment discharge and snow storage

Fig. 8a shows the simulated daily discharge and catchment SWE averaged over the 6 years simulation period for the mid (red lines), min, and max estimates (bounds of the shaded areas), and the no-ARF scenario (black lines). The differences in daily

discharge and catchment SWE of the min, mid, and max estimates to the no-ARF scenario are shown in Fig. 8b. All simulations with ARF show higher daily discharge from end of March until end of May and lower discharge from end of May until mid August relative to the no-ARF simulation. For the rest of the year, no effect on the discharge is noticeable. The net impact of RFS results in a shift in the timing of discharge. Higher discharge early in the melt season is observed, yet offset by lower discharge following May. The cumulative annual discharge remains nearly identical.

Min, mid, and max estimates all show the change from higher to lower discharge compared to the no-ARF scenario approximately at the same time (at the end of May; see blue marker in Fig. 8b). Therefore, we can quantify the absolute and relative effect of RFS on the discharge during the two periods: the early melt season from circa March 22 until May 29 and the late melt season from circa May 30 until August 10 (Fig. 8b and see Table 3). This yields an average percentage increase in daily discharge of 2.5 %, 9.9 % and 21.4 % for the min, mid, and max estimates for the early melt season and a decrease in discharge

of -0.8 %, -3.1 %, and -6.7 % during the late melt season.

The differences in discharge among the scenarios can be explained by understanding the evolution of the snowpack. In the all scenarios the catchment SWE (Fig. 8a) reaches a peak reduction relative to the no-ARF scenario of -4.6 %, -13.4 % and -34.4 % at mid May. The average difference in catchment SWE of the min, mid, and max estimates compared to the no-ARF scenario during the entire melt season is -1.5, -5.1, and -10.3 mm; or an average of 2.1 %, 7.4 %, and 15.1 % (see Table 3).

From mid May on, the differences in catchment SWE between scenarios drop continuously, which is equivalent to a higher catchment averaged snow melt rate in the no-ARF scenario compared to the ARF scenarios.

## 6  Discussion

The objective of this work is to provide a mechanism to assess the impact of light absorbing aerosols on runoff at the catchment scale in a rainfall-runoff modelling context. Prior investigations into LAISI indicate potentially significant impacts to the cryosphere (Flanner et al., 2007) with potential impacts to water resources (Qian et al., 2009, 2011). Earlier studies on
hydrologic impacts at the catchment scale have used altered radiative forcings to evaluate the impact on the timing of snow melt and hydrology (Painter et al., 2010; Skiles et al., 2012). With the approach presented herein, we seek to fill a gap between land-surface model approaches (e.g., Oaida et al., 2015) and approaches which apply modified radiative forcings to provide a novel tool for hydrologic forecasting.

### 6.1  Parameter sensitivity

To assess the sensitivity of the newly introduced algorithm and parameters, we conducted a sequence of 1-d sensitivity studies. In this context, we are able to remove complexities that arise when conducting a distributed simulation at the catchment scale.

We found the greatest sensitivity to lie in the parametrization of scavenging, as it relates to how likely the aerosol is to remain at the snow surface during melt. Field measurements indicate that only a fraction of BC is flushed out with the melt water and BC can accumulate near the snow surface (e.g., Xu et al., 2012; Doherty et al., 2013; Sterle et al., 2013; Doherty
et al., 2016). Our model is able to simulate this process by taking the scavenging ratio of BC during meltwater movement into account (Eq. 8 and 9). In the literature, the scavenging efficiency of BC is discussed controversially. Flanner et al. (2007)'s estimates for scavenging ratios of hydrophilic and hydrophobic BC, which are used in this study, are based on data from field experiments using artificially added soot (Conway et al., 1996). However, parameters derived from artificially added soot might not be directly transferable to the scavenging properties of naturally occurring BC. Even though field observations from
Doherty et al. (2013) agree well with the estimates of Flanner et al. (2007), and further studies highlight the importance of BC retention in the snowpack (e.g., Xu et al., 2012; Sterle et al., 2013), a large uncertainty remains on the magnitude of this effect (Lazarcik et al., 2017). These uncertainties are identified in our simulations as results show large differences in BC evolution and day of meltout at the boundaries of the applied scavenging ratios (Fig. 3b). Compared to the no-ARF experiment (black lines), the presence of BC causes an earlier meltout for all scavenging ratios applied, spanning from 2 days (upper boundary
scavenging) to about 9.5 days (lower boundary scavenging). Remarkable is that even when applying efficient melt scavenging (2.0, upper boundary of hydrophilic BC), resulting in nearly all BC removed from the snow, the melt out still happens circa two days earlier compared to the no-ARF experiment.

Further complicating the effect is the fact that hydrophilic BC (which undergoes more efficient melt scavenging) has a larger MAC (enhanced absorption) compared to hydrophobic BC (Flanner et al., 2007). Our results suggest distinguishing between
species may play a secondary role in the determination of the overall impact of BC on snow melt due to the compensating effect of stronger scavenging accompanied with stronger absorption and vice versa (Fig. 3c).

The 1-d model experiments further show that the definition of at least two layers for the snowpack model is important to allow for accumulation of impurities at the snow surface. This result in itself is not original, numerous prior studies have

identified the importance of having multiple layers (Krinner et al., 2006; Flanner et al., 2007; Oaida et al., 2015). However, we further find that the model surface layer thickness (parameter *max_surface_layer*; see Table 2) has great impact on the evolution of surface mixing ratios of BC, while at the same time the effect on albedo and snow melt is small. This results from the fact that for all 2-layer models the surface layer thickness is much thinner than the penetration depth of shortwave radiation.

For example, in clean snow with an optical grain radius of 50 $\mu$m, the radiative intensity diminishes to 1/e of its surface value (the so called penetration depth) in 25.5 mm SWE. For snow with an optical grain radius of 1000 $\mu$m, the penetration depth increases to 117 mm SWE (both results from Flanner et al., 2007, assuming a wavelength of 550 nm and a solar zenith angle of 60°). Thus, BC in the surface layer absorb efficiently in all 2-layer scenarios and the difference in the albedo is relatively large compared to the no-ARF scenario (solid black line in top graph of Fig. 3a), but relatively small among the 2-layer scenarios

(solid green, purple, and red curves in top graph of Fig. 3a). However, there is a critical difference when a single layer model is used (yellow curves in Fig. 3a) due to the aerosol being distributed uniformly throughout the snowpack instead of allowing accumulation at the surface. Thus, a large fraction of the BC is located at depths where the radiative intensity is much lower than in the top few mm of the snowpack, leading to a weaker absorption efficiency and results in a less pronounced decrease of albedo compared to 2-layer models and thus to a shorter meltout shift compared to a clean snowpack than in the 2-layer

scenarios.

Observations of BC in melting snow support the accumulation of BC near the surface (Xu et al., 2012; Doherty et al., 2013; Sterle et al., 2013; Delaney et al., 2015). In a sequence of snow pits, Sterle et al. (2013) showed that during the ablation season, BC mixing ratios increase significantly near the snow surface (sampled in the top two centimeter) relative to bulk BC concentrations. They suggest that most likely a large fraction of previously deposited BC becomes concentrated near the

surface. Delaney et al. (2015) also report of surface BC increase during melt, to which BC being trapped at the snow surface is likely to contribute. BC increase in surface snow of up to an order of magnitude (Sterle et al., 2013; Doherty et al., 2016) and more (Xu et al., 2012) have been observed in natural snow during melt. Over most of the melt period, our results show a factor increase between 5 and 15 for the 2-layer scenarios, which alignes well with observations. Higher values are mainly predicted shortly before meltout, when the snowpack is typically very thin and effects on discharge generation due to high increase in

surface BC should be small.

We argue therefore the importance of providing, at a minimum, a separate surface layer, but recognize simulated surface mixing ratios of BC are highly sensitive to the thickness of this layer. Since evaluation of model predictions for BC in snow is commonly performed by comparing simulated with observed BC mixing ratios in surface snow (e.g., Flanner et al., 2007; Forsström et al., 2013), this is a critical result. Snow is often sampled in top few centimeters (typically 2 to 5 cm, e.g.,

Doherty et al., 2010; Aamaas et al., 2011; Forsström et al., 2013). This raises an interesting challenge given that the surface layer assumed in models is not a measurable property of snow. A comparison of model simulations with observations should therefore include some quantification of the uncertainty resulting from the layer thickness parametrization.

## 6.2 Hydrologic response to BC deposition in a snowfall dominated catchment

We are interested in addressing the impact of BC deposition – and potentially other light absorbing aerosols – to the hydrology of snowfall dominated catchments. Studies have shown the potential impact LAISI may have on the timing of snowmelt (Skiles et al., 2012; Painter et al., 2012) while others have argued the impact to climate may be signficant (Flanner et al., 2007, 2009; Qian et al., 2009, 2011). Given the importance of snow for water resources for a significant portion of the population (Barnett et al., 2005; Sturm et al., 2017) and the rapid growth of BC emissions in certain regions of the world (e.g., Paliwal et al., 2016; Bond et al., 2013), our aim is to provide a mechanism to include this process in hydrologic forecasting to better address future impact studies.

Forsström et al. (2013) found BC seasonal mean snowpack concentrations from about 10 ng g$^{-1}$ to 80 ng g$^{-1}$ for different measurement locations and time periods in mainland Scandinavia. Generally our results are within those presented in Forsström et al. (2013), though our max estimate lies above. However, Flanner et al. (2007) evaluated the global impact of the radiative forcing of BC in snow using a model which was compared with globally distributed surface BC measurements. For south Norway, Flanner et al. (2007) predicted an annual mean surface BC concentration between 46 and 215 ng g$^{-1}$ for the year 1998, placing our simulations fully within a reasonable range of prior reported values.

The impact resulting from BC deposition in our study is seen in the timing of the annual water balance. Inclusion of ARF generally increases early season melt and causes the snowpack to melt out earlier. Comparing the ARF and no-ARF scenarios we see a general shift in the discharge, with the ARF scenario producing greater discharge early in the season, and having less discharge after June. Such a shift in seasonal water balance will potentially have impacts to soil moisture and agriculture (Blankinship et al., 2014), as well as regional climate (Qian et al., 2011). While we recognize significant uncertainties associated with conceptual hydrologic modelling that may impact the applicability of these results (Beven and Binley, 1992; see also uncertainty discussion in Sect. 6.3), we feel it provides a novel mechanism to address LAISI in a manner that, to date, is not available otherwise. As a reality check of the catchment scale process representation, we evaluate the impact of the incorporation of BC deposition on albedo, radiative forcing, and snowpack storage.

### 6.2.1 Surface BC and albedo

Albedo is a critical parameter in any snow melt model, having significant control over the energy balance. During the accumulation period, the average albedo of each scenario lies within the range of albedo of fresh snow with small optical grain radius combined with a high solar zenith angle (Gardner and Sharp, 2010) and is thus reasonable for a high latitude snowpack during snow accumulation. The differences in snow albedo during the accumulation season are mostly due to differences in aerosol deposition and in the maximum surface layer thickness of the snowpack. The time series of mid estimate modelled surface BC is within the range of values for locations in mainland Scandinavia presented in Forsström et al. (2013) during the accumulation period. The min estimate predicts values at the lower bound and lies in the range of the background surface BC level found in Svalbard in the European High Arctic (5 ng g$^{-1}$, Aamaas et al., 2011; 30 ng g$^{-1}$, Clarke and Noone, 1985). Compared to Forsström et al. (2013), the surface BC level of the max estimate seems to exceed the range of values reasonable

for mainland Scandinavia during snow accumulation and reflects a range of values that is rarely found in snowpacks outside Asia (Doherty et al., 2010; Forsström et al., 2013; Wang et al., 2013; AMAP, 2015).

During the melt season, the evolution of surface BC yields end of the melt season reductions in albedo relative to the no-ARF case of about 0.03, 0.1, and over 0.3 for the min, mid, and max estimate, respectively. This has two reasons: (i) with increasing grain radius during the melt season, the absorbing effect of BC gets more efficient due to deeper penetration of radiation into the snowpack leading to a stronger effect of the BC deposition on albedo. Snow of larger grains has a larger extinction coefficient and more effective forward scattering properties (Flanner et al., 2007). (ii) with the start of the melt season there is a widespread decrease of snow thickness, allowing BC to accumulate in the surface layer. This latter effect is strongly dependent on the applied scavenging ratios, as we demonstrated in the 1-D sensitivity study (Sect. 5.1). During the melt season, the mid estimate spatially averaged surface BC mixing ratio increases from 49 ng g$^{-1}$ to about 250 ng g$^{-1}$ (factor 5 increase) at the end of the melt season (beginning of July). Observations from Forsström et al. (2013) indicate that surface BC concentrations around 250 ng g$^{-1}$ are well within the range of reasonable values for a melting Scandinavian snowpack. Furthermore, an increase in surface BC by a factor of 5 and higher during snow melt is in line with observed BC trends in melting snow from different locations (Doherty et al., 2013, 2016; Xu et al., 2012). From this, we argue that our mid estimate simulation predicts a seasonal cycle in surface BC that is within reason.

For the max estimate, the increase is from roughly 100 ng g$^{-1}$ to over 2500 ng g$^{-1}$ (factor 25 increase). This strong seasonal cycle in surface BC is beyond what is observed for both, absolute BC values in Scandinavian snowpacks and increase relative to surface BC during snow accumulation. The min estimate, on the other hand, leads to a decrease in BC surface mixing ratio. Even though many studies report of an increase in surface BC during snow melt (e.g., Conway et al., 1996; Doherty et al., 2013, 2016; Xu et al., 2012), there exist observations showing that a large fraction of BC can be flushed efficiently from the snowpack with the beginning of snow melt (Lazarcik et al., 2017). This indicates that post-depositional enrichment processes and their significance on determining surface BC trends in melting snow require further exploration. We argue that the min estimate thus marks a reasonable lower bound estimate for the seasonal evolution of surface BC.

We recognize our max estimate results in a strong increase in surface BC mixing ratios mostly due to low BC scavenging with melt (note the strong increase from end of March on in Fig. 6). This divergent evolution of surface BC mixing ratios in the min, mid, and max estimates reveals uncertainty in the representation of the fate of BC in snow during melt, which is also reflected in the literature (Doherty et al., 2013, 2016; Xu et al., 2012; Lazarcik et al., 2017).

### 6.2.2 BC induced radiative forcing

The strong increase in RFS (Fig.7a) and surface radiative forcing (Fig.7b) during spring melt results from the combination of (i) the aforementioned decrease in snow albedo due to the increase in surface BC concentrations (e.g. melt amplification and the increasing optical grain radius in melting snow as discussed in Sect. 5.2.2) and, (ii) the increasing daily solar irradiation due to a lower solar zenith angle and longer days.

Annual mean surface radiative forcing in this study are 0.284, 0.844, and 1.391 Wm$^{-2}$ for the min, mid, and max estimates. Averaged over entire Scandinavia (including Finland), Hienola et al. (2016) calculated lower values around 0.145 Wm$^{-2}$.

However, Hienola et al. (2016)'s study includes large areas with shorter snow cover. Since the surface radiative forcing is strongly depended on the snow cover evolution, higher values compared to Hienola et al. (2016) are expected due to the long lasting snow cover in our case study region. The mid estimate annual cycle of surface radiative forcing due to the presence of BC in the study region is of similar magnitude of what is found over the Tibetan Plateau. Qian et al. (2011) reports of

similar snow cover duration and maximum mean forcing during May of over 6 $Wm^{-2}$ using a global climate model. Due to the generally much lower snow covered fraction in Qian et al. (2011)'s study region, however, RFS is presumably significantly higher on the Tibetan Plateau compared to our study region, which is in agreement with very high levels of BC reported for the Tibetan Plateau (Qian et al., 2011). Using a standalone version of SNICAR, we estimated RFS based on surface BC mixing ratios from Forsström et al. (2013) measured during melt in the 5 cm of Scandinavian snowpacks to 4.7 to 18.2 W $m^{-2}$ (95%

confidence interval; details described in Appendix A). These values agree well with our min and mid estimate RFS (Fig. 7), however, are significantly lower than our max estimate.

### 6.2.3  BC impact on catchment discharge and snow storage

We mention a shift in the seasonal water balance, with more melt early in the melt season resulting from enhanced RFS. However, from mid May the melt enhancement reduces and the differences in catchment SWE between the ARF and no-ARF

scenarios decreases (Fig. 8b). One would expect with more incoming radiation, later in the season, the RFS effect to become further enhanced. However, this counter-intuitive result becomes more clear when one considers the impact of fractional snow covered area and catchment scale processes. The dynamics driven by the faster development of SCF (see Fig. 6a) is a limiting factor to the catchment averaged snow melt. By comparing Fig. 7a, which shows the snowpack RFS enhancement, with Fig. 7b, which shows total daily energy uptake in the catchment, we see that a threshold period is reached and total daily energy uptake

decreases, while RFS is continually increasing. Intuitively, one would expect more melting due to enhanced solar radiative forcing. However, the SCF decrease with increased melt due to ARF counteracts the RFS effect itself, due to the reduction in area from which snow can actually melt. For discharge, this is manifested in the ARF scenarios as an enhancement during the beginning of the melt season attributed to RFS, whereas the decreased discharge later in the season is attributed to melt limitation caused by the faster growth of fractional bare ground areas.

Similar shifts in the annual water balance due to the impact from LAISI are reported for the Upper Colorado River Basin (Painter et al., 2010) and the Tibetan Plateau (Qian et al., 2011). Those regions are well known hotspots of LAISI disturbance to snow cover (Painter et al., 2007; Qian et al., 2014). Our results suggest that also the hydrologic cycle of regions that have not been into the focus hitherto (such as Norway) might be significantly affected by ARF.

Compared to observations, all simulations (ARF and no-ARF) tend to underestimate discharge during early melt season

and overestimate discharge during late melt season (Fig. 8a). However, the magnitude of over- and underestimation strongly differs between the scenarios. By including ARF the volume error is reduced in both the early melt season (by increasing melt), and in late melt season (by subsequently decreasing melt generation in the catchment due to reduced SCF). Expressed as seasonal mean volume error for early and late melt season, the difference to observed discharge is largest for the no-ARF scenario and smallest for the max estimate. The max estimate reduces the volume error by -75.1% during early melt season

and -89.9% during late melt season, relative to the no-ARF scenario (see Table 4). The min and mid estimates also reduce the volume error. Thus, on average, an improvement in simulated discharge is achieved during the melt season by accounting for BC RFS. Similar results are achieved when estimating model parameters using a no-ARF scenario (not shown). However, we acknowledge that further studies are needed in order to be able to confirm a general model improvement when accounting for

ARF in snow dominated catchments. Certain mechanisms can lead to model improvements for the wrong reason when applying ARF (Kirchner, 2006). Structural deficits of the model might lead to a negligence of processes that are important for the spring melt generation. The implementation of ARF could then optimize the model towards the observations and counteract errors coming (partly) from a missing process that is not related to ARF. A further potential mechanism is related to the equifinality of conceptual models. These implications coming from model parameter uncertainty are discussed in Sect 6.3 alongside with

further sources of uncertainty.

## 6.3   Uncertainties

There are numerous challenges associated with the development of an algorithm which mixes conceptual hydrologic parametrizations with physically based algorithms. Both the literature and our analysis highlight aspects that warrant a deeper investigation of ARF-induced uncertainty. The intent with this work is to introduce a new algorithm, however as indicated in Pappenberger

and Beven (2006), we feel it is important to provide an initial assessment of the uncertainty introduced with the addition of ARF terms. To achieve this we have conducted a Generalized Likelihood Uncertainty Estimation (GLUE; Beven and Binley, 1992) which provides an assessment of the degree of variability in behavioral models resulting from equifinality of parameters.

With respect to the implementation of a physical albedo model, the treatment of the darkening effect of LAISI adds additional degrees of freedom to the parameter space of the model due to the introduction of new parameters (scavenging ratios, surface

layer thickness, BC input scaling factor; see bottom 4 parameters in Table 2). In order to investigate the abilities and limits of the model with and without ARF to reflect the observed discharge, we quantify the parameter uncertainty prior and posterior to the implementation of ARF calculations (Fig. 9; details in Appendix B). Uncertainties are generally largest during snow melt and summer because various parameters only play an active role in calculating discharge during snow melt. Including ARF calculation in the model leads to a shift of the uncertainty band to higher values during April and May, and lower values

during June and July, due increased melt under the impact of ARF. From mid-May to mid-June, the ARF induced shift in the uncertainty band leads to observations being within or closer to the border of the uncertainty bands (shaded box in Fig. 9), which can be interpreted as an improvement to the model. This would imply that in the model without ARF, albedo decays not sufficiently enough during spring in order to generate enough snow melt, resulting in an underestimation of discharge in April and May. However, we admit that further testing is needed to draw a more accurate conclusion, as discussed above. Perhaps

more importantly, it appears that we have not increased uncertainty much by adding complexity. In general with and without ARF the results are acceptable, however, we enable the inclusion of a potentially important variable, particularly with respect to increased emissions due to population growth.

In our case study, further uncertainties result from mixing ratios of BC in the snowpack due to prescribed BC deposition, and LAISI other than BC not accounted for in the model:

### i) - prescribed BC deposition

In the approach presented here, we use prescribed BC deposition mass fluxes. Even though this is common practice (e.g., Goldenson et al., 2012; Lee et al., 2013; Jiao et al., 2014), it was showing by Doherty et al. (2014) that the decoupling of aerosol deposition from the water mass flux of falling snow can lead to an overestimation of surface mixing ratios by a factor of 1.5-2.5. However, we would like to highlight an important difference between our approach and the one Doherty et al. (2014) claim to be problematic: First, the high bias in surface snow BC mixing ratios described by Doherty et al. (2014) refers to global climate model simulations with prescribed aerosol deposition rates (wet and dry), where the input aerosol fields are interpolated in time from monthly means. Therefore, the episodic nature of aerosol deposition due to wet deposition is generally absent in the prescribed aerosol fields. The coupling of the interpolated fields with highly variable meteorology (in particular precipitation) results in the high bias (Doherty et al., 2014). In our case study, on the other hand, we use deposition fields originating from the regional aerosol climate model REMO-HAM, forced with ERA-Interim reanalysis data at the boundaries. REMO-HAM output is 3-hourly, which we re-sampled to daily means in order to have consistency between the deposition fields and the observed daily precipitation used as input data in the hydrological simulations. The daily timestep allows us to preserve the episodic nature of aerosol deposition. Moreover, the daily BC wet deposition rates should not be biased due to major inaccuracies in precipitation as REMO-HAM has been shown to reproduce the Scandinavian precipitation realistically (Pietikäinen et al., 2012). The high bias occurring when using interpolated monthly averages as input should therefore be minimized. Additionally, and significantly, Doherty et al. (2014) (and the critiques therein) address an objective with consideration to climate impacts. Our analysis is focused on the impact to the hydrological cycle. Our simulations suggest that BC RFS is mostly important during spring time, where surface BC mixing ratio are predominantly controlled by melt processes, and not by deposition processes (as shown in Fig. 3 and Fig. 6b).

### ii) - LAISI other than BC

By including only BC deposition in our simulation, we likely underestimate the additional effect of further LAISI species such as mineral dust (Di Mauro et al., 2015; Painter et al., 2010), mixing of the snow with soil from the underlying ground or local sources (Wang et al., 2013) and biological processes (Lutz et al., 2016). Neglecting additional RFS from LAISI other than BC is likely to result in an underestimation of the overall effect of LAISI on snow melt and discharge generation. Especially the contribution from dust is critical since it has been shown that in many regions such as the Rocky Mountains (Painter et al., 2012), Utah (Doherty et al., 2016), the southern edge of the Himalayas (Gautam et al., 2013), and Svalbard (Forsström et al., 2013), dust can play a significant role in terms of RFS or even is the dominating LAISI. For Norway, however, analysis conducted by Forsström et al. (2013) indicate that dust might only play a minor role. By comparing samples from Svalbard and near Tromsø, Norway, Forsström et al. (2013) showed that there exits a distinctive difference between the Arctic Archipelago and the mainland. BC mixing ratio from mineral-dust-rich Svalbard measured by the thermal/optical method used in Forsström et al. (2013) averaged about half the mixing ratio of insoluble light-absorbing particulates (including dust) measured by an optical method (ISSW: Integrating Sphere/Integrating Sandwich; e.g., Doherty et al., 2010). Samples collected close to Tromsø, on the other hand, resulted in BC that averaged about 1.3 times the ILAP mixing ratios. Due to the fact that the ISSW method overestimates BC for samples containing dust, Forsström et al. (2013) argues that the comparison of both methods can be used

to draw conclusions about the pollution regime. Yet, due to the small number of samples and the single-location analysis, this needs to be addressed more in future studies in order to identify the relative importance of different LAISI species.

With respect to our study, we acknowledge that including only BC is a shortcoming with respect to the overall effect of LAISI. However, by demonstrating the significant effect of BC on accelerating snow melt and discharge generation, our study gives a conservative estimate of the effect of LAISI and urges a more detailed investigation.

## 7 Conclusions

Herein we presented a newly developed snow algorithm for application in hydrologic models that allows a new class of model input variables: the deposition rates of light absorbing aerosols. By coupling a radiative transfer model for snow to an energy balance based snowpack model, we are providing a tool that can be used to determine the effect of various species of LAISI at the catchment scale. In this analysis we have focused solely on BC and acknowledge it therefore likely represents a conservative estimate. This work presents a novel analysis of the impact of BC deposition to snow on the hydrologic cycle through 1-D sensitivity studies and catchment scale hydrologic modelling. From a 1-D model study, presented in Sect. 5.1, we conclude that:

i - the implementation of at least two layers (a thin surface layer and a bottom layer) is of outstanding importance to capture the potential effect of melt amplification on the near surface LAISI evolution. The parametrization of the surface layer thickness (in SWE) has a rather little effect on the snow albedo and melt rate as long as it is sufficiently small (e.g. smaller than the penetration depth of shortwave radiation). However, the evolution of the LAISI surface mixing ratio is highly sensitive to the surface layer thickness. For this reason, we suggest to include a surface layer thickness variation in model studies when comparing simulated to observed LAISI mixing ratios sampled in the top few centimeters of snow.

ii - The determination on how LAISI is washed out of the snowpack with melt water has great effect on the evolution of LAISI concentration near the surface, snow albedo and melt rate. Due to rare observations of this effect under controlled conditions the uncertainties are high and our findings show the need for more detailed understanding of the processes involved due to the high importance for the overall effect of LAISI on the snowpack evolution.

To demonstrate the significance of BC radiative forcing for the hydrologic cycle at the catchment scale we demonstrated the effect of BC deposition and the subsequent implications for snow melt and discharge generation on a remote Norwegian mountain catchment. The study indicates that inclusion of BC in snow is likely to have a significant impact on melt timing, and that the effect on the discharge generation leads to a shift in the annual water balance. Our simulations further suggest that melt amplification can have severe implications on both, the snowpack evolution and the discharge regime of a catchment, which means that the seasonal cycle of surface BC mixing ratio is of great importance. However, large uncertainties are connected with the representation of surface enrichment of BC. A more robust understanding of the fate of BC in melting snow is essential to fully assess impacts to the hydrologic cycle.

Including radiative forcing from BC in the simulations leads to a reduction in volume error during the early and late melt season in our simulations. We conclude from our study that hydrological modelling can potentially be improved by including the effect of LAISI, especially when the model approach implicates a physically based representation of the snowpack in general and the snow albedo in particular. However, more research in the area of catchment scale impact of LAISI is needed to support this. The approach and algorithm presented in this analysis provides a tool to target this in future applications.

## Appendix A:  Radiative forcing in snow estimated from Forsström et al. (2013)

In order to calculate radiative forcing in snow (RFS) from surface concentrations during melt reported in Forsström et al. (2013), several assumptions have been made. For each input variable, a certain reasonable range is estimated, suiting to snow properties during melt conditions:

- snow optical grain radius: 500-1000 $\mu$m

- snow density: 400-600 kg m$^{-3}$

- BC mixing ratio: 50-200 ng g$^{-1}$ (from Forsström et al., 2013)

Forsström et al. (2013) reports of 6 time series of BC surface concentrations sampled in the top 5 cm of the snowpack. All of which cover the snow melt period at 3 locations in Scandinavia, however, only one location can be considered as remote without pollution from local sources (Abisko, Sweden). The range of BC concentrations during melt is estimated from this location. Global radiation during spring is estimated to 210 W m$^{-2}$. The value has been calculated from the input time series of our study region, in order to receive comparable results. The daily mean solar zenith angle has been set to 60° and BC concentration below the top 5 cm to 0, since no further information is available. The latter might lead to an underestimation of RFS and results can be seen as a conservative estimate. 1000 realizations with SNICAR have been conducted using different input variable sets, with random values for each input variable according to a uniform distributing in the stated range. Resulting RFS values are presented as 95% confidence interval to 4.7 to 18.2 W m$^{-2}$. The mean is 11.2 W m$^{-2}$.

## Appendix B:  Parameter uncertainty with GLUE

We determine parameter uncertainty using the Generalized Likelihood Uncertainty Estimation (GLUE) method (Beven and Binley, 1992). Lower and upper bounds of parameters used in the calculation are shown in Table A1. We use the Nash-Sutcliffe model efficiency (Eq. 10) as likelihood function and choose a threshold value of 0.74 (0.1 below best calibration result) for accepting parameter sets as behavioral parameter sets. To identify the impact of ARF on model uncertainty, we run GLUE twice, first without ARF applied, and in a second round of simulations accounting for ARF. Random parameter sets are created by choosing parameters according to a uniform distribution in the range of the parameter bounds. For each of the two uncertainty estimations, a total of 10000 model realizations was drawn of which 1435 (no-ARF) and 1831 (ARF) parameter sets were rated as behavioral parameter sets. This accounts for about 14% and 18% of the total samples, respectively.

*Acknowledgements.* This work was conducted within the Norwegian Research Council's INDNOR program under the Hydrologic sensitivity to Cryosphere-Aerosol interaction in Mountain Processes (HyCAMP) project (NFR no. 222195). We thank the Mitigation of Arctic warming by controlling European black carbon emissions (MACEB) project for their help concerning the REMO-HAM simulations. Furthermore, we thank the International Institute for Applied System Analysis (IIASA), especially Kaarle Kupiainen and Zbigniew Klimont, for providing the the emissions data. Sigbjorn Helset and Statkraft AS, in general, have been vital resources in the development of the algorithm and, in particular, the implmentation into Shyft. The ECHAM-HAMMOZ model is developed by a consortium composed of ETH Zurich, Max Planck Institut für Meteorologie, Forschungszentrum Jülich, University of Oxford, and the Finnish Meteorological Institute and managed by the Center for Climate Systems Modeling (C2SM) at ETH Zurich.

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

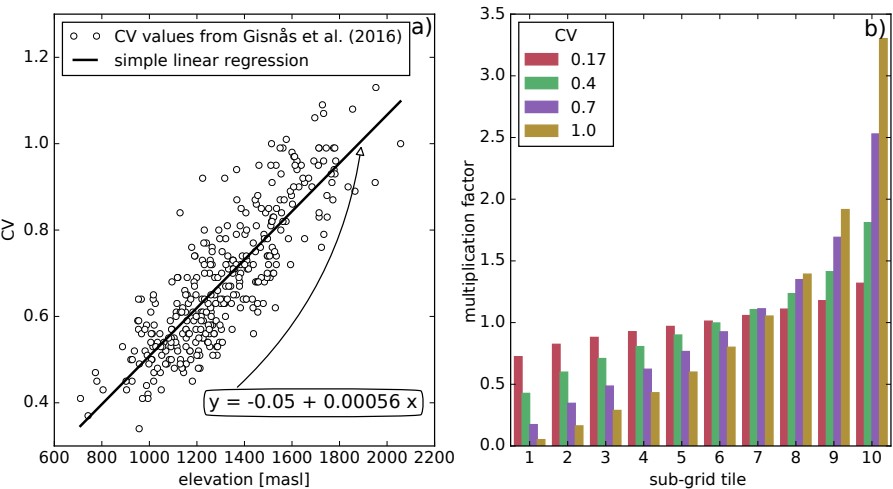

**Figure 1.** Left: elevation versus coefficients of variation (CV) of sub-grid snow distribution from Gisnås et al. (2016) of forest free areas in the Atnsjoen catchment (dots) and the relationship between the CVs and the elevation resulting from simple linear regression analysis (black line). Right: solid precipitation multiplication factors for the sub-grid snow tiles for different CVs.

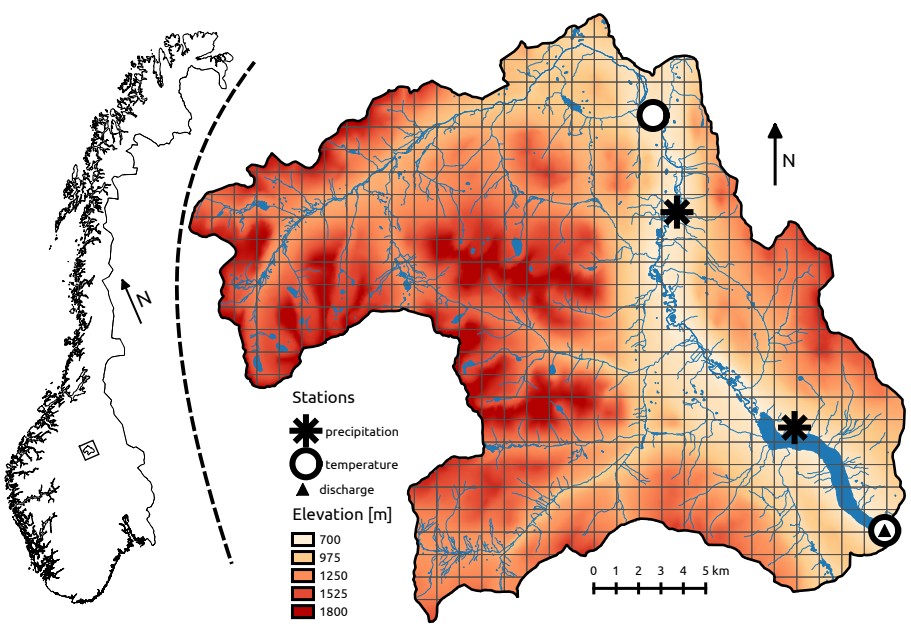

**Figure 2.** Location of the Atnsjoen catchment in Norway (black box in left map) and overview map of the Atnsjoen catchment (right).

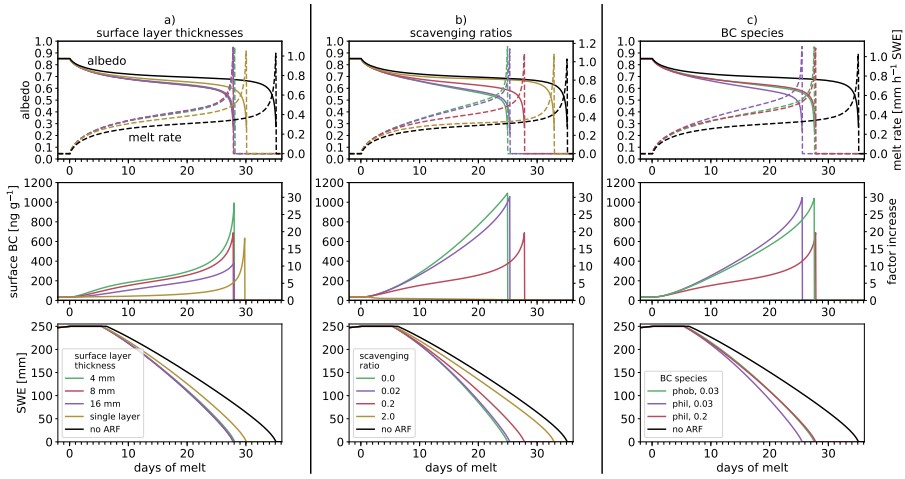

**Figure 3.** Snow albedo (top row of graphs; solid lines) and melt rate (top row of graphs; dashed lines), BC mixing ratio in the surface layer and factor increase of the mixing ratio during melt compared to the pre-melt BC mixing ratio (central row of graphs), and snowpack SWE (bottom row of graphs) for simulations forced with synthetic data based on the average meteorological conditions during the melt season from mid March until mid July of the Atnsjoen catchment and different model configurations: (a) different values for maximum surface layer thickness; (b) scavenging ratio; and (c) BC species with different melt scavenging ratios applied (*phob* and *phil* in legend stands for hydrophobic and hydrophilic BC, respectively). The black lines in all graph show simulation results of model runs without ARF applied (no-ARF).

**Table 1.** Information about observational stations.

| Station name | Station ID | Operator | Observational variable | Elevation |
|---|---|---|---|---|
| Atnsjoen 1 | 8720 | MET | precipitation | 749 |
| Atndalen-Eriksrud | 8770 | MET | precipitation | 731 |
| Atnsjoen 2 | 2.32.0 | NVE | temperature | 701 |
| Li Bru | 2.479.0 | NVE | temperature | 780 |
| Fokstuga | 16610 | MET | wind speed; relative humidity | 973 |
| Kvitfjell | 13160 | MET | wind speed | 1030 |
| Venabu | 13420 | MET | relative humidity | 930 |

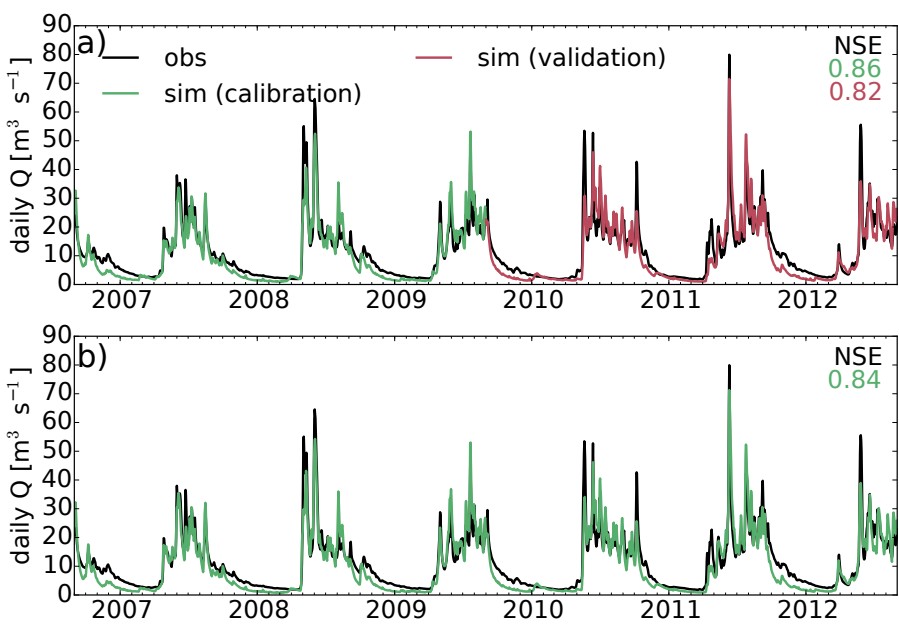

**Figure 4.** Simulated (green and red curves) and observed (black curve) daily discharge from the Atnsjoen watershed. Graph (a) is showing the simulation results for 3 years of calibration (green) and 3 years of validation (red). Graph (b) is showing the results for the 6 years calibration period. Parameters estimated in the latter are used in the case study. Parameters not included in the optimization are set to mid estimate values during the calibration process (see Table 2).

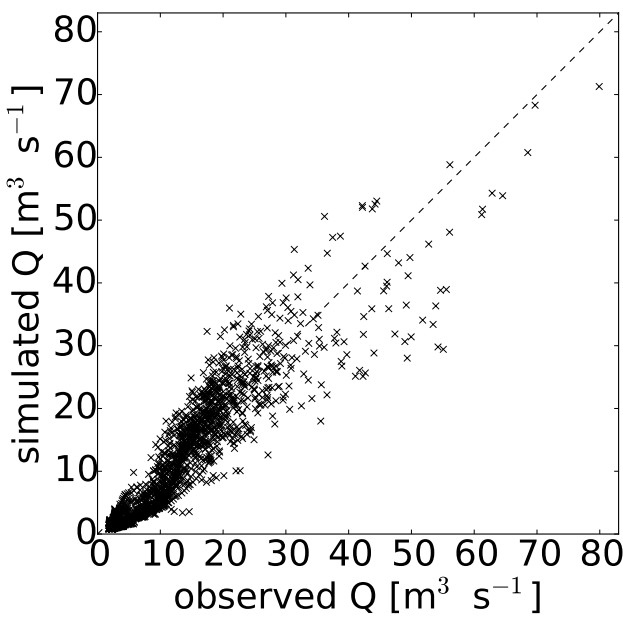

**Figure 5.** Comparison of observed and simulated daily discharge Q of the Atnsjoen catchment. The dashed black line demonstrates perfect agreement between simulation and observation.

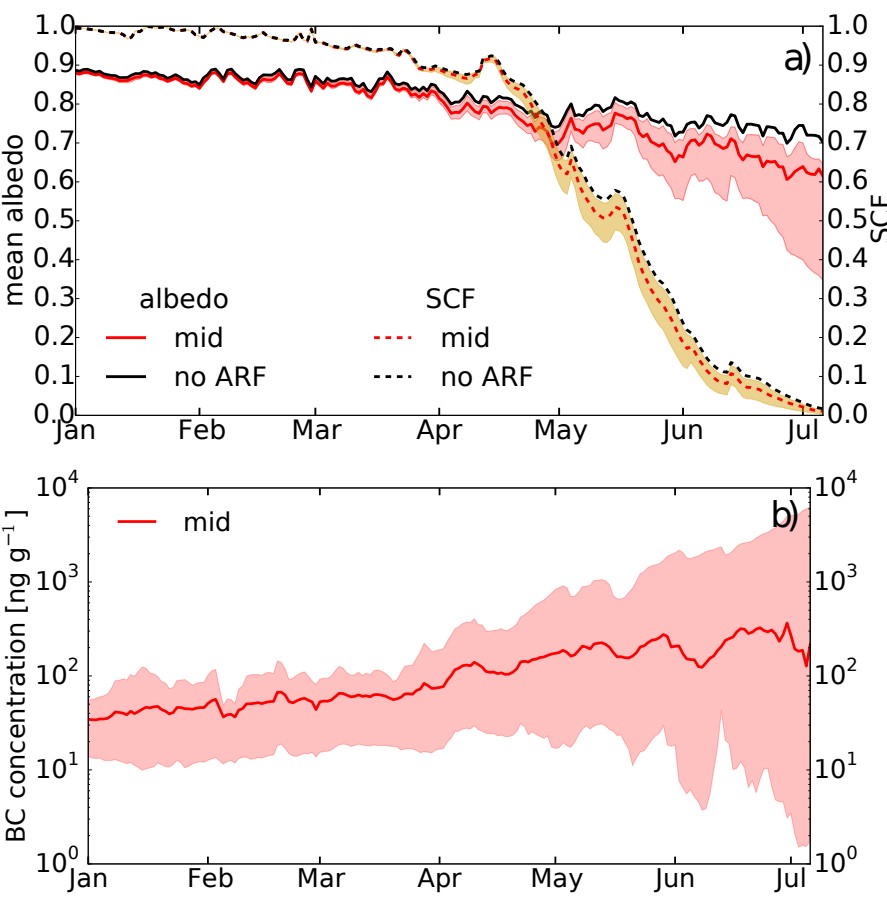

**Figure 6.** (a) Simulated mean catchment snow albedo (solid lines) and snow covered fraction (SCF; dashed lines) for the mid (red lines), low and high (shaded) estimates and for the scenario without ARF (no-ARF; black lines) averaged over the 6 years period. (b) Concentration of BC in the surface layer of the model for the mid (solid line), min (lower bound of shaded area) and max (upper bound of shaded area) estimates.

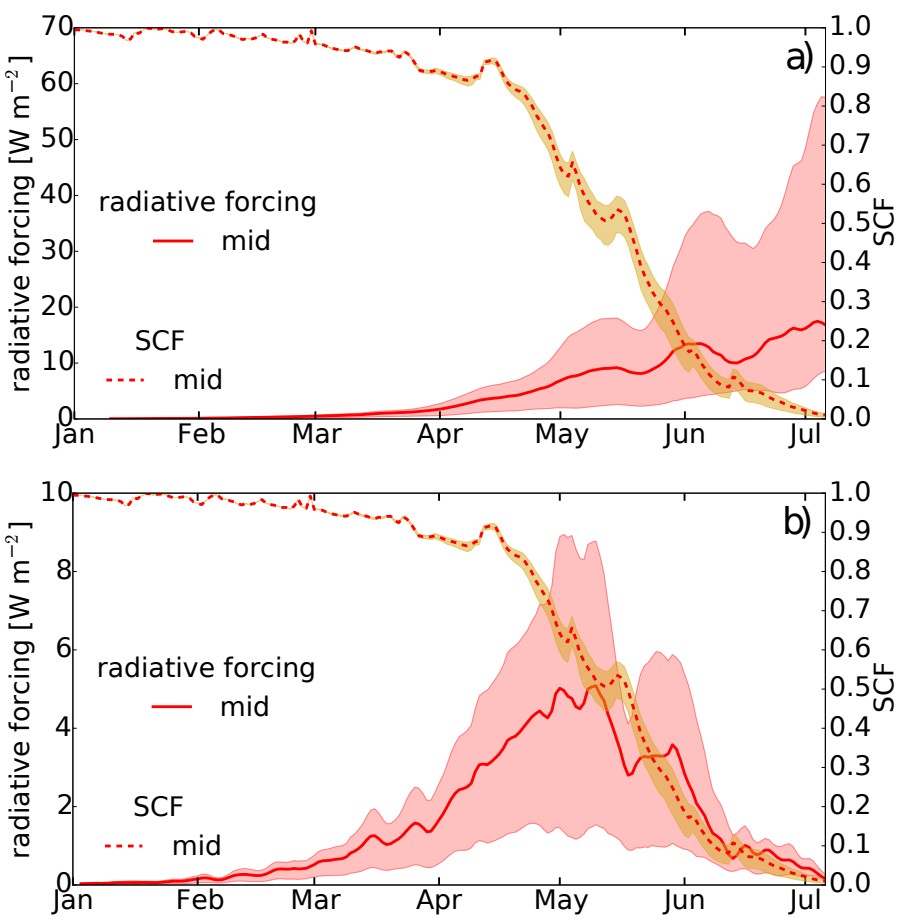

**Figure 7.** Catchment snow covered fraction (SCF; dashed lines) and (a) simulated mean radiative forcing in snow and (b) total daily energy uptake in the catchment due to BC for the mid (solid red lines), min (lower bound of shaded area) and max (upper bound of shaded area) estimates averaged over the 6 years period (daily means presented in Watts per square meter catchment area).

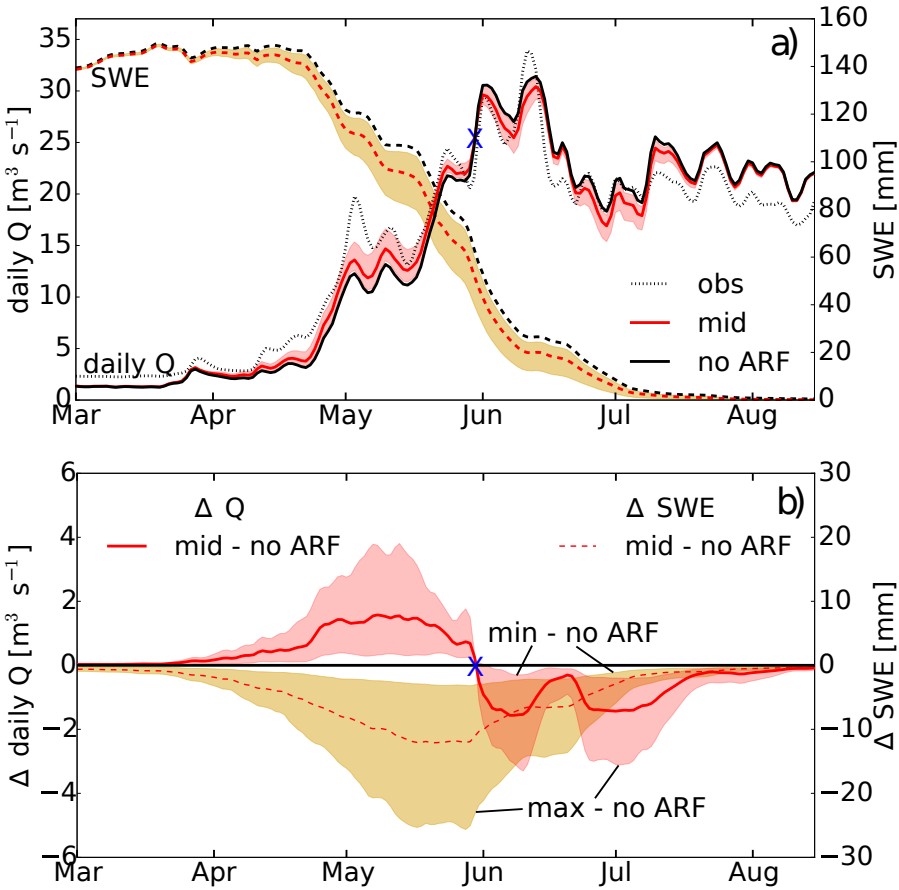

**Figure 8.** (a) Simulated daily discharge (Q; solid lines) and catchment mean snow water equivalent (SWE; dashed lines) for the mid (red lines), low and high (shaded) estimates and for the scenario without ARF (no-ARF; black lines) averaged over the 6 years period. (b) Differences in daily discharge and SWE of ARF scenarios to the scenario without ARF (no-ARF). The blue marker in (a) and (b) separates the periods where BC in snow has an enhancing (left of marker) and a decreasing (right of marker) effect on the discharge.

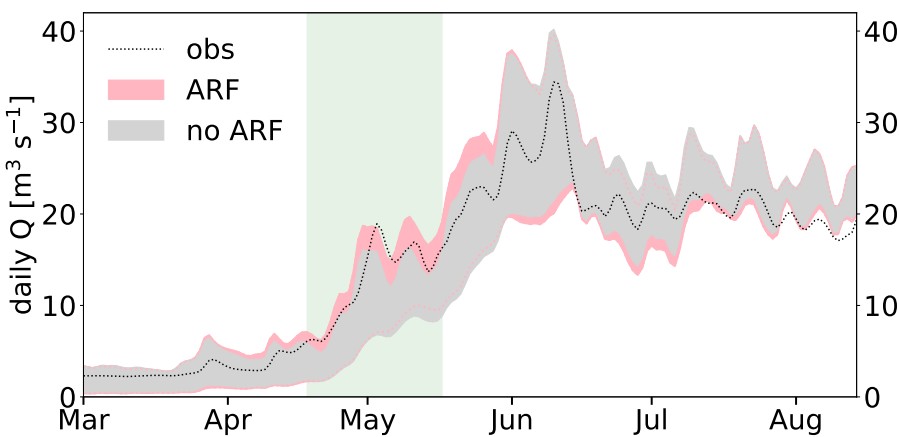

**Figure 9.** 95% confidence interval of discharge due to parameter uncertainty when allowing for ARF (red) and disregarding ARF (grey), calculated using the Generalized Likelihood Uncertainty Estimation (GLUE) method and averaged over the 6 years simulation period. The shaded box marks the period of the melt season, where observations tend to lie outside the uncertainty bounds of the no-ARF simulations.

**Table 2.** Model parameters used in sensitivity and case study. Parameters optimized during calibration are marked with *. Further parameters were pre-set and not open for calibration. Parameters with different values in the minimum (min), central (mid), and maximum (max) BC radiative forcing estimates are marked with **.

| Parameter | Description and unit | min estimate | optimized/set mid estimate | max estimate |
|---|---|---|---|---|
| $c_1$ * | kirchner parameter 1 (see Eq. 3) [-] | | -4.298 | |
| $c_2$ * | kirchner parameter 2 (see Eq. 3) [-] | | 0.3295 | |
| $c_3$ * | kirchner parameter 3 (see Eq. 3) [-] | | -0.07757 | |
| ae_scale_factor * | scaling factor for actual evapotranspiration [-] | | 1.43 | |
| tx * | temperature threshold rain/snow [°C] | | -0.92 | |
| wind_const * | determining wind profile [-] | | 6.32 | |
| wind_scale * | determining wind profile [-] | | 1.12 | |
| snowfall_reset_depth | minimum snowfall required to reset $A_s$ [mm SWE] | | 5.0 | |
| snow_cv_forest | snow CV in forested area [-] | | 0.17 | |
| snow_cv_intercept | intercept of linear elevation-CV relation [-] | | -0.05 | |
| snow_cv_slope | slope of linear elevation-CV relation [m$^{-1}$] | | 0.00056 | |
| max_water | fractional max water content of snow [-] | | 0.10 | |
| $A_{s,0}$ | $A_s$ of fresh snowfall [m$^2$ kg$^{-1}$] | | 73.0 | |
| surface_magnitude | Max snow depth for snow heat content [mm SWE] | | 30.0 | |
| max_surface_layer ** | Maximum thickness of surface layer [mm SWE] | 16.0 | 8.0 | 4.0 |
| depo_factor ** | Multiplication factor for deposition [-] | 0.5 | 1.0 | 1.5 |
| $k_{phob}$ ** | scavenging ratio of hydrophobic BC [-] | 0.3 | 0.03 | 0.003 |
| $k_{phil}$ ** | scavenging ratio of hydrophilic BC [-] | 2.0 | 0.2 | 0.02 |

**Table 3.** Average change in discharge during the early (March 22 to May 29) and late (May 30 to August 10) melt season of min, mid, and max estimates and average change in SWE during the melt season (March 22 to August 10) compared to the no-ARF scenario (zero BC mass deposition).

| scenario | early melt season discharge [m$^3$ s$^{-1}$] | [%] | late melt season discharge [m$^3$ s$^{-1}$] | [%] | melt season SWE [mm] | [%] |
|---|---|---|---|---|---|---|
| min estimate | 0.2 | 2.5 | -0.18 | -0.8 | -1.5 | -2.1 |
| mid estimate | 0.81 | 9.9 | -0.74 | -3.1 | -5.1 | -7.4 |
| max estimate | 1.74 | 21.4 | -1.60 | -6.7 | -10.3 | -15.1 |

**Table 4.** Season mean volume error in discharge during the early (March 22 to May 29) and late (May 30 to August 10) melt season of no-ARF, min, mid, and max scenario compared to observed discharge. The percentage change shows an increase (+) or decrease(-) of the volume error compared to the no-ARF volume error.

| scenario | early melt season discharge | | late melt season discharge | |
|---|---|---|---|---|
| | $[\text{m}^3 \text{ s}^{-1}]$ | [%] | $[\text{m}^3 \text{ s}^{-1}]$ | [%] |
| no-ARF | -2.32 | - | 1.78 | - |
| min estimate | -2.12 | -8.7 | 1.60 | -10.1 |
| mid estimate | -1.52 | -34.7 | 1.04 | -41.6 |
| max estimate | -0.57 | -75.1 | 0.18 | -89.8 |

**Table A1.** Model parameter bounds used in the uncertainty estimation with the Generalized Likelihood Uncertainty Estimation (GLUE) method. Parameters used to determine ARF are marked with *.

| Parameter | Unit | Lower bound | Upper bound |
|---|---|---|---|
| $c_1$ | [-] | -7.0 | -2.0 |
| $c_2$ | [-] | 0.1 | 1.0 |
| $c_3$ | [-] | -0.1 | 0.0 |
| ae_scale_factor | [-] | 0.7 | 2.0 |
| tx | [°C] | -2.0 | 1.0 |
| wind_const | [-] | 3.0 | 10.0 |
| wind_scale | [-] | 0.5 | 2.0 |
| snowfall_reset_depth | [mm SWE] | 3.0 | 7.0 |
| snow_cv_forest | [-] | 0.15 | 0.2 |
| snow_cv_intercept | [-] | -0.03 | -0.07 |
| snow_cv_slope | $[\text{m}^{-1}]$ | 0.0003 | 0.0007 |
| max_water | [-] | 0.5 | 0.15 |
| $A_{s,0}$ | $[\text{m}^2 \text{ kg}^{-1}]$ | 50.0 | 100.0 |
| surface_magnitude | [mm SWE] | 20.0 | 40.0 |
| max_surface_layer * | [mm SWE] | 4.0 | 16.0 |
| depo_factor * | [-] | 0.5 | 1.5 |
| $k_{phob}$ * | [-] | 0.003 | 0.3 |
| $k_{phil}$ * | [-] | 0.02 | 2.0 |