# Peer review of "Modelling hydrologic impacts of light absorbing aerosol deposition on snow at the catchment scale"

_Hydrology and Earth System Sciences, 2016_

## Referee Comment (RC1) · Anonymous Referee #1 · 22 Nov 2016

General comments:

There are four significant issues with the paper that need to be addressed before it is ready for publication:

1) The model independently treats the "wet deposition" of BC (pg 11, lines 12-134) from the deposition of snowfall with which it is nominally associated (pg 11, lines 5-6). This is an issue. Doherty et al. (2014) showed using the CESM/SNICAR model (after which the snow model described herein appears to be very closely modeled) that this results in a factor of 1.5-2.5 high bias in surface snow mixing ratios. The authors should look at this paper, consider the implications for their study, and add analysis/discussion of how this impacts their results (or justify why it doesn't). [Doherty, S. J., C. M. Bitz and M. G. Flanner, 2014: Biases in modeled surface snow BC mixing ratios in prescribed-

aerosol climate model runs, Atmos. Chem. Phys., 14, 11697-11709, doi:10.5194/acp-14-11697-2014.]

2) In the sensitivity study of how varying snowpack depth affects the impact of BC on snowpack melt (Section 5.1.4), snowpack SWE is varied while the total *mass* of BC deposited is also kept fixed (see pg. 18, lines 3-5). The justification is that this will "isolate the impact that the snowpack's SWE has on the effect of ARF in snow." However, that isn't quite correct. BC's impact on snow albedo/forcing/melt rate is a function of the mass mixing ratio of BC in snow water (ng BC per gram of SWE) – not of the total mass in the snow. By increasing SWE but not changing the mass of wet-deposited BC two things are being changed simultaneously: total snowpack SWE (definitively increasing) and the BC mass mixing ratio (definitively decreasing). I'd strongly argue that a better approach would have been to increase BC deposition in proportion to the increase in SWE so one can see to what degree having a "equally-polluted" but deeper snowpack changes the effect of the pollution on melt rate, versus a base case with the same pollution "level" (e.g. BC mass mixing ratio) but a shallower snowpack. Either this sensitivity study needs to be re-run or the paper needs to acknowledge that the results reflect these two simultaneous changes, discuss how this impacts their results, and note that this is not likely physically realistic – which makes me question the robustness of conclusion iii (pg 23, line 20).

3) pg 21, lines 17-19: "At the same time, tiles bearing large quantities of snow tend to also bear large quantities of BC (in terms of total BC mass) due to the dominantly wet-depositioned BC, which we chose in the model to follow the same redistribution as snow. Only dry deposition is assumed to deposit spatially homogeneous over the sub grid tiles." If I am reading this correctly, the wet-deposited BC is effectively concentrated only onto snow-covered areas. Thus, as the snow becomes increasingly patchy the remaining snow gets more and more BC mass wet-deposited to it. This is completely unphysical: wet-deposited BC falls to the ground, whether it's covered with snow or not. Perhaps I am misunderstanding and this is just an issue of needing better clarity

in the writing: The first sentence indicates the wet-deposited BC "follows the same distribution of the snow": Is this of the snow on the ground, or of the snowfall? If the latter, okay; if the former, as the second sentence seems to imply, I don't understand why this choice would be made since it's not physically reasonable. I'd expect this would significantly affect your results.

4) The paper is difficult to read. Much of it has run-on sentences that are overly convoluted. There are a few specific cases I note below ("Technical corrections") where outright corrections are needed but much more work is needed beyond this. I would strongly suggest that the co-authors work with the lead author to improve the clarity and conciseness of the writing.

Specific comments:

References need to be added to support statements made in a number of places: a) pg 2, lines 5 through 14 to support a string of assertions b) pg 7, line 10 re: radiative exchanges dominating snow melt in most snow melt scenarios (and perhaps qualify it, too; I'm assuming temperature is the dominant factor for many conditions) c) pg. 15, line 6 re: LAISI absorbing more efficiently in snow with larger grain size d) pg. 17, lines 16-17: "Hydrophilic BC absorbs stronger than hydrophobic BC under the same conditions due to an increased MAC compared to hydrophobic BC caused by the ageing of BC during atmospheric transport." (In reality, the degree to which this is the case is not very well-established so references supporting this assertion really are needed.)

pg 2, lines 13-14: organics from combustion and organics in soil also are LAISI and should be added to this list.

pg 2, lines 17-19: "Current theory indicates the absorbing effect of LAISI is most efficient when the LAISI reside at or close to the snow surface, and that subsequent snow fall burying the LAISI leads to a decline in or complete loss of the effect." The latter half of this statement is not accurate (and no reference is given to support this assertion).

This statement assumes that new snowfall is essentially clean (BC-free), then BC is subsequently deposited on top of the snow. In reality BC is deposited *with* the snow, in wet deposition – at least in the real world. So I don't think "current theory" indicates that it works as stated.

pg. 5, lines 29-30: Here and earlier in the test dust, black carbon, volcanic ash and other light-absorbing aerosols are mentioned, but only BC is included in the model. It would be good to be clear that the only LAISI you are currently accounting for in the model is BC.

pg. 6, lines 1-2: You need to be specific about what you mean by a gamma distribution.

pg. 8, line 14: What is the basis for using this specific formulation?

pg 8, lines 20-25. I found this discussion of what is "appropriate" for a surface layer thickness to be confusing. Snow doesn't have a defined "surface layer" so it's not as if this is fixed quantity that has some "true" value in the real world. What is appropriate to use for the model surface layer thickness would be a function of what metric you are interested in. Here it could be, for example, the e-folding depth of sunlight penetration, or it could be the depth over which most melt amplification of BC mass is concentrated. Or any number of other things, depending on what you're interested in.

pg 8, line 24-25: "Since we expect surface concentrations of LAISI in snow to be quite sensitive to the surface layer thickness in our model..." In reality this should only be the case for dry-deposited BC. Wet-deposited BC should be deposited with snow; if the mixing ratio of BC in snowfall were unchanged throughout a new snowfall event the mixing ratio of BC in the surface layer could be completely insensitive to the depth you select for the "surface layer". It should be made clear that surface concentrations of LAISI in the model might be sensitive to the selected surface layer thickness because you are decoupling BC mass deposition and SWE deposition, and not state this as if this were an inherent property of real ambient snowpacks.

[Figure]

pg. 9, lines 10-11: "We allow for melt from the bottom layer only when the potential melt per time step is exceeding the maximum depth of the surface layer (both in mm SWE)." It's unclear if you mean that no melt is allowed to occur in the bottom layer until the surface layer is saturated, or if you mean that no melt water is allowed to exit the bottom of the surface layer until the surface layer is saturated.

pg. 9, line 12: "To date, estimates of the scavenging ratio k are mostly based on experiments conducted by Conway et al. (1996)." Doherty et al. (2013) also estimated the scavenging ratio from ambient snowpacks in two locations. In fact their estimates agreed quite well with that used in Flanner et al.'s SNICAR model – values you seem to adopted here in your model. It would therefore be appropriate to note this, both because there is a study other than Conway et al. (1996) and because their results support the "mid" scavenging values you use.

pg. 9, line 24: The assumption of all wet-deposited BC being hydrophilic and all dry-deposited BC being hydrophobic is not justified, either here or in Section 3.

pg. 10, lines 27-28 and Figure 1: It's not at all clear what is meant by "multiplication factors" or how they are used. Figure 1, left panel: coefficient of variation in what? Specify in the figure caption. Figure 1, right panel: It's not at all clear what is being shown here. What are the "factor numbers"? What is the (unlabeled) vertical axis? Why different "factor numbers" for each CV?

pg 12, line 19 / Table 3: Why set radiation to zero during the "accumulation periods"?

pg. 12, line 21: "The forcing applied during the snow accumulation period of 180 days results in 250 mm of SWE at the end of the accumulation period." then pg 12, line 25-28: "After the snow accumulation period, we invoked a time invariant forcing to slowly melt the snowpack until meltout. The forcing applied for melt is based on the average forcing during the melt season from mid March until mid July of the Atnsjoen catchment and results in a melt period of ca. 25-35 days, depending on the scenario applied." I'm quite confused by the use of the term "forcing" here. I would assume

you mean radiative forcing, but that would make no sense in the first sentence. Which makes me wonder what you mean by "forcing" in the 2nd and 3rd sentences. Are you calling temperature and precip variations "forcings"? If so, this is quite unconventional, at least for someone from the climate community. Either some explanation or a revision is needed here.

pg 13, Sections 4.1.2 & 4.1.3 and Figure 3c: There aren't different "species" of BC. "Hydrophobic BC" generally refers to fresh – i.e. uncoated – BC, and "hydrophilic BC" is really BC that's been coated. The BC itself in each is essentially the same. I'd suggest a re-wording/re-naming.

pg 14, lines 8-9: "Bayesian Kriging" This needs a bit of an explanation or at least a reference.

pg. 14, lines 9-10: "For precipitation, BC deposition rates, wind speed and relative humidity this implies interpolation to the model cells via inverse distance weighting, with a constant vertical gradient applied for precipitation." Do you mean that precip varies with land altitude (with some constant gradient) or that precip is constant with altitude? If the latter, rewording is needed; if the former, some quantification of this vertical gradient is needed.

pg. 14, line 11 and Figure 5: a) It's not clear what is meant by a "split sample calibration". b) What is used to calibrate the model? What parameters are varied to achieve the best "calibration" / tuning? c) In Figure 5, the top panel shows data for 2007-2012 with the first three years as "calibration" data. In the bottom panel, these same three years are shown as "validation". Isn't that a bit circular? Or perhaps I don't understand what the difference is between the two panels. d) In Figure 5, it's not stated whether the model is run assuming a perfectly clean snowpack (BC deposition = 0), or something else. (See comment below re: Conclusions section and Figures 5-7.)

pg 15, lines 14-18: "The stronger increase in surface BC in model setups with thinner surface layer is due the inversely proportional relationship of the surface layer thick-

[Figure]

ness with the increase in impurity concentration under the same mass flux of LAISI into the surface layer (from deposition or melt amplification): halving the surface layer thickness, leaving the mass flux of LAISI into the surface layer unchanged, leads to a doubling of the increase in the LAISI concentration and thus to differences in the vertical distribution of LAISI. . ." Again, this is only true because you are decoupling BC wet deposition and snowfall deposition. (see comment above re: pg 8, line 24-25)

pg. 16, lines 8-9: "a doubling of the surface layer LAISI concentration occurs already when the accumulated melt equals the surface layer thickness": "Equals" in terms of what? SWE? Where is this shown? I don't see this in Figure 3.

pg. 16, lines 17-18: "the surface concentration of the aerosol simulated strongly depends on the magnitude of the surface layer," Poor wording. What do you mean by the "magnitude" of the surface layer? The surface layer depth?

pg. 16 Section 5.1.2 and Figure 3a: I'm confused by what is shown in Figure 3a middle panel. The BC concentration appears to start at zero, then increase from there. How can the surface snow concentration start at zero? How can there be a "factor increase" for a parameter that starts at zero?

pg. 17, lines 13-14: "showing that small amounts of BC in snow can impact the snow-pack evolution over the whole melt period even if it undergoes an efficient scavenging process." Up to here no results have been presented that indicate what pre-melt surface snow BC mixing ratios are. This isn't given until pg 20 Section 5.2.3. So the reader really can't know whether a) the model is giving reasonable surface snow mixing ratios of BC and b) what you mean by "small amounts of BC in snow". (Nominally Figure 3 would show this, but as noted above these values all start at zero so it's hard to know what point in the evolution of concentrations you're talking about here).

pg. 17, line 26: "which we assume to be the most suited": Based on what?

pg 18, line 16 and pg 23, lines 20 (bullet item iii): "are less impacted" (pg 18) and "are

more prone to be affected (pg 23): By what metric? The "melt shift days"? Is this the only metric of importance? Figure 4 shows the "meltout shift" vs snowpack SWE as a function of different BC scavenging ratios. The meltout shift of 60 days for the deepest snowpack is indeed impressive, but a) we don't know what the BC mass mixing ratios were in this model run and b) we don't know what the total number of melt days is so it's kind of hard to put these results into context. (It must've been at least a few months for the meltout shift result of 60 days. Is this correct??) Perhaps the relative change in number of meltout days (as a percentage?) would be a better metric.

pg. 18, line 23: "NSEs" needs to be defined.

pg. 18, lines 25 "winter discharge": what time period is "winter" here? Also, Figure 6 does not indicate seasonality. How do we know that the low flow cases are in winter?

pg. 19, Section 5.2.2.: Again it's difficult to interpret these results since we don't yet know what the model was calculating for surface snow BC mass mixing ratios, and whether they were even vaguely realistic. So I think some re-ordering of the presentation of results is needed.

pg. 21, lines 26-29: "Qualitatively, we feel this represents reality well, in that if we think about snow patches in a catchment at the end of the season, they tend to be 'dirty', as the concentration of impurities increases while the water melts away." Yes, but in the real world, on which you are basing your observations of reasonableness, this visible darkening of the snow is very likely dirt accumulating, not BC.

pg. 23, lines 24-25: "Even though our model approach is conservative due the lacking implementation of the effect of LAISI on the grain size growth and due to the choice of a remote northern catchment of only medium snow accumulation" It should have been spelled out sooner that grain size growth is not affected by the presence of LAISI (i.e. on pg 8, ∼ lines 15-18).

Conclusions:

a) Figures 5 - 7 and Results Discussion + Conclusions: The study indicates that inclusion of BC in snow has a significant impact on melt timing (Figure 7). Yet it's not at all clear whether the model calibration and validation (Figure 5) include the effects of BC or are based on using a clean snowpack. I was surprised that there was no testing or discussion of whether including BC in snow improves modeled vs observed catchment outflow volume/timing (Figures 5 & 6).

b) The discussion totally ignores the fact that real snowpacks have particulate absorbers other than BC. In this regard the impact of BC (pollution) on snow albedo, radiative forcing and melt rates in this study represent an upper limit. If other absorbers – i.e. naturally-occurring dust and dirt – were also included in the model study the impact of adding BC would be less. This needs to be noted and acknowledged.

Technical corrections:

pg 2, line 20: the wording that LAISI "can reappear and retain near to the surface" is both awkward and not accurate. It doesn't "reappear" – it just becomes more concentrated at the surface as the snow water runs out through the snowpack at a higher rate than the BC.

pg 4, lines 26: P & E need to be defined when they are first used (even though it's pretty obvious what is meant here. . .)

pg 5, lines 14-15: "Furthermore, the presence of a permanent snow layer and snow melt leads to a more challenging identification of periods when the change in liquid water storage is governed by discharge only." I can't figure out what it is you're trying to say here.

pg 18, line 25: "simulated over observed" should be "simulated versus observed"

pg. 20, lines 10-11: "We see the albedo of the max scenario having the largest drop and the one of the no ARF scenario being the lowest." Needs rewording. The *decline* in albedo is smallest; this reads as if the *albedo* is the lowest.
pg. 23, line 22: "To prove the significance. . ." I think you mean to *test* the significance.

---

## Referee Comment (RC2) · Anonymous Referee #2 · 22 Jan 2017

General Comments:

Major revisions are necessary before this paper is to be published. While I appreciate the various sensitivity tests to try and isolate the impact of various parameters as they relate to impact of light absorbing impurities (LAI) in snow in models, some are not executed or interpreted correctly in this study. The implementation of LAI in snow processes in a hydropower forecasting model, as attempted in this work, is important, as is indeed mostly lacking. However, I don't quite agree with some of the phrasing in the Introduction that claims LAI in snow in hydrologic models (or land surface models that have physically-based hydrologic processes) has been up to now understudied or lacking. Many examples can be found in Qian et al., 2015 (AAS), Light-absorbing particles in snow and Ice: measurement and modeling of climatic and hydrologic impact.

[Figure]

Generally, here are some of my concerns or things that are unclear based on the current manuscript: 1. Paper is very hard to read, and logic often hard to follow. There are many run-on sentences that are very wordy. Some terms seem to be used to refer to various different processes, and need clarification; e.g. "forcing" is at times used as LAI in snow forcing, whereas during other times it is used to refer to meteorological forcing. Being clearer in explanations is needed. 2. I also found the organization of the paper to be cumbersome, which plays into readability of the manuscript. a. For one, Section 2 on Modeling Framework is difficult to piece together, to understand how various components of the framework work together, and how the actually set up this modular "model platform for hydrologic purposes". Clear definitions of model setup and model meteorological inputs (sec 2.1) need to be specified. b. In addition, authors often describe several methods or quote values for parameters, but don't clearly state which they end up using in present study, or what modifications have been made. Reader is left lost in previous studies or potential methods. e.g. sec 2.2.3, or p8 – SNICAR implementation in hydro model not clearly laid out. 3. While it is true LAI in snow play a significant role on energy and water balance across many mountainous regions throughout the world, the authors don't state if this is in fact a problem in Norway, or for the catchment they chose for the case study. What is the motivation for choosing this catchment, if LAI in snow observations are lacking here, making drawing realistic conclusions difficult (which even they admit e.g. p22 lines 22-23)? Why not do a case study with this new model over a region that does have in situ observation of LAI in snow, e.g. Painter et al, 2012 (Dust radiative forcing in snow of the Upper Colorado River Basin: 1. A 6 year record of energy balance, radiation, and dust concentrations, WRR); or Kaspari et al., 2015 (Accelerated glacier melt on Snow Dome, Mount Olympus, Washington, USA, due to deposition of black and mineral dust from wildfire); or Zhao et al., 2014 (Simulating black carbon and dust and their radiative forcing in seasonal snow: a case study over Northern China with field campaign measurements). 4. If I understand their modeling framework correctly, BC deposition is decoupled from meteorological forcing applied, making the entire discussion of distribution of wet deposition of BC difficult to

rationalize realistically. Authors should at the very least discuss how this decoupling impacts their results and conclusions. I am also a bit concerned with the fact that BC is only deposited in snow during the accumulation phase. In reality, BC deposition, and LAI in general, in snow does not suddenly stop with melt onset, and in fact, certain LAI species such as dust are mostly deposited during the springtime (e.g. Painter et al., 2012, WRR) and therefore during the snowmelt period. This means deposition of LAI in snow during melt season play a significant role in melt magnitude and timing (and not only during accumulation). If deposition of BC in snow tends to indeed occur mostly during accumulation in Norway, or in the catchment in their case study, then authors should state that as an explanation (with appropriate references) for why they set up their experiments the way they did. 5. Authors use the phrasing of "addition of deposition rates of LAI" throughout manuscript (e.g p1 ln5, p23 ln 8) as a way of communicating the improvements they contribute – this is misleading, as what they really use is the LAI mass and concentration. By using "deposition rate" they suggest they are improving the atmospheric to surface deposition process, the rate and temporal distribution of LAI in snow, when really they are implementing a way for hydrologic model to account for LAI in snow (and in fact actual deposition and precip inputs are decoupled here). This needs to more accurately be represented throughout the paper.

Specific Comments:

P1 ln 12-13: confusing sentence; what is meant by "melt limitation" . . . this confusing term remains confusing throughout the paper (e.g. p20, ln20). A clearer description of the concept is needed.

P1, abstract: "Central effect" or "min, max, mid effect estimate" terminology hasn't yet been described, so use in Abstract leads to reader being confused as to what it's referring. Ln 14: "The central effect estimate produces reasonable surface BC concentrations in snow" The effect produces BC concentrations? Wouldn't BC in snow be the element producing an effect? Re-word sentence with clearer statement.

P1, ln 20: what's the difference between "mountainous" and "high mountain" – is there a need for mentioning both environments? Not the same?

P1 ln 24: "affected areas" is not adequate use of the phrase – these areas didn't experience an extreme event, and were not "affected". Suggest removing word "affected"

P2 ln 5-7 and ln 7-9: need references.

P2 Ln 18-19: statement needs reference.

P2 Ln 24-28: what's the point of that lit review? How does it impact the present work? What did you take from it, or how did you improve it?

P3 ln 1-2 and that paragraph in general: "investigating the impact of LAISI on the snow melt and runoff predominantly use empirical formulations to investigate the impact of LAISI on the radiative forcing in snow, by observing the net surface shortwave fluxes over snow and identifying the contribution from the LAISI through determination of the (hypothetical) clean snow albedo" – inaccurate, misrepresents previous work and the context of this work. There have been several improvements to LAI in snow representation in hydrologic or land surface models in recent years (albeit further developments continue to be needed), e.g. Zhao et al, 2014 (ACP), Oaida et al., 2015 (JGR), and see Qian et al., 2015 (AAS) for a more complete list and overview of observations and modeling of LAI in snow. Authors of present study show reframe their motivation or gap their new work is filling given these previous developments.

P3 ln 13-15: Entire sentence is awkward; what is meant by "complex abstractions"?

Sec 2.1: clearly state what variables the model needs as inputs (this is later alluded to on page 11, but needs to be more clearly stated under Modeling Framework section)

P4 ln 11: what is meant by "efficient simulation"?

P4 ln 12: why is ET module important? The whole modeling setup needs to be more clearly defined and laid out
P8 ln 1-9: how exactly did you integrate SNICAR within hydrologic framework? Which variables in 2.2.1 were updated by SNICAR output … and what does SNICAR output? P8 10-17: how is r connected to radiative transfer model? Where is r used in your implementation?

P10 ln 9-14: A bit unclear how these tiles are defined? Is it based on elevation? Also, "In our model, we further developed an approach assuming that the spatial distribution of each single event of 10 solid precipitation follows a certain probability distribution function." This newer approach is based on which previous method? What did you further develop?

P10: the concept of "multiplication factor" is not quite clear.

P11, ln 17-30+: is the REMO simulation ran offline, separately from the hydrologic model? Is there a discrepancy between deposition timing in REMO and hydrologic model meteorological precipitation input/events? How does that affect your study? (Also see General Comment 4 above).

P11, sec 3.1: what is the simulation period for hydrologic model, vs for REMO? Might want to even state the hydrologic model simulation period more clearly a bit earlier in the paper, in the intro to Section 3, before 3.1.

P12 ln 10-12: run on sentence. Please revise.

P12, ln 24-25: why did you chose to only deposit BC during accumulation period, and not throughout entire simulation period, or at least during both accumulation and ablation periods? Also see General Comment 4.

P13, ln 9-11: Sentence should be better integrated, and phrased more grammatically correct.

P14 ln 2-3: is BC distributed throughout the top layer, or entire snowpack?

P14, ln 19: what do you mean by "all free model parameters"?

P15, ln 4-6: confusing sentence. "The central graph in Fig. 3a shows that the choice of the maximum surface layer [insert "thickness"] strongly determines the increase in the [insert "magnitude of"] surface concentration over the melt season - leading to a strong increase in surface BC until [insert "through"] the end of the melt season with an increase in BC by a factor of circa 15, 30 and 60 for maximum layer thicknesses of 4.0, 8.0 and 16.0 mm, respectively, compared to the pre-melt season BC concentration."

P15, ln 4-9: These 2 sentences, if I understand correctly, seem to be at odds with each other: the latter, "The thinner the surface [...]" implies that the 4mm layer selection would have the strongest effect, yet it's only increasing BC by a factor of 15, smallest of them all.

P15, ln 19: "the mean radiative intensity diminishes with depth due to absorption in snow and LAISI and scattering, leading to a less effective absorption of LAISI in deeper snow. " needs a reference

P16, ln 21-24: what about new BC deposition? You mention on the ways the output of LAI from snowpack affects end of season LAISI amount, but what about the input, which may vary through time, and which again brings me back to general comment 4.

P18, section 5.1.4: The number of earlier meltout should probably be scaled by total length of meltout season of each snowpack to more realistically and accurately represent the impact of snowpack thickness

P19, ln 1-24: this entire section is rather convoluted and the conclusions not easy to follow. Because of that, some of the results seem at odds with each other. Please re-organize and be more concise in your analysis. ... ln 7: "total sum of daily discharge" refers to net annual sum of runoff? And it is about zero? Yet later in the paragraph the % change increases? Perhaps I am misunderstanding the stats – a more clear explanation would be helpful. One idea is to also put all these values in a table, for easier comparison. You mention ET, is there a plot to support the conclusions you are mentioning?

P19, ln 30: wouldn't the 1.5, 5.1, and 10.3 mm values be negative?

P20 ln 3-8: I am not quite sure what you are trying to say about having an analysis at the catchment scale. The links you are trying to draw don't seem that obvious or easy to follow.

P21 ln 2-4: scavenging ratio is not the only factor determining if BC accumulated in top snow layer. What about new snow?

P21, ln 26: "Qualitatively, [. . .]" – this sentence is not a very strong, supported, conclusion.

P21, ln 13-19: reason (iii) is rather confusing. The whole concept of "wet deposition" of BC in this explanation doesn't quite add up for me when this study has BC and precipitation "falling" separately (processes decoupled). It's possible I am misunderstanding the explanation, which might suggest a more clear explanation would help.

P23, ln 1-5: I would argue that normalizing SCF isn't necessarily more relevant to inpact on runoff, as total surface albedo (both snow and snow-free surfaces) influences snowmelt thought the snow-albedo feedback.

P23, ln 12-15: "The maximum thickness (in SWE) of the surface layer herein has rather little effect on the snow albedo and melt rate as long as the maximum layer thickness is sufficiently small." – is this clean snow, or LAISI case? "However, the evolution of the LAISI surface concentration is highly sensitive to the choice of the surface layer extent." If LAISI concentration is affected by snowpack thickness, then wouldn't snow thickness, somewhat indirectly, affect albedo, since surface snow layer LAI impact snow albedo?

P23 ln 27-29: I am not sure the evidence presented is enough to conclude improvement in hydrologic modeling. The shift found by comparing LAISI and no-LAISI scenarios certainly suggests an impact of LAISI on discharge timing, but one would have to compare LAISI, no-LAISI, and observed runoff over same period of time to conclude that a hydrologic model with LAISI processes present brings simulated runoff closer to

observations, over the no-LAISI simulated runoff. You could add no-LAISI discharge to figure 5 to have a more robust conclusion on model improvement.

Technical Corrections:

"LAISI in snow" is used in several parts of paper (e.g. p8 ln 18), which is redundant since LAISI already contains "in snow" by their own definition. Please revise.

P4 ln 2: too many "hydrological"/"hydrologic"/"hydropower" terms in one sentence. Please revise.

P5 ln 27: "central addition" is awkward. "Main addition"?

P15: word "Stronger" is repeated 2x. Remove one.

P22 ln 20: "are" is repeated 2x back to back.

---

## Referee Comment (RC3) · Anonymous Referee #3 · 26 Jan 2017

**General comments**

My knowledge of hydrological models is not broad, so I do not believe I am qualified to comment on the viability and implementation of the model. However, I have commented on the structure, content and more scientific issues that I see in this article. My first criticism is that the paper is long and should be shortened and restructured. Secondly, I did not find a useful quantification of how LAI from the ARF model are integrated in to the snowpack, as no field measurements of LAI from the area are available. Also, better parameterization of dust sources is needed. My final criticism, and one I take very seriously, is that the authors fail to cite and recognize substantial research that has been done in this field, leading to comments in the text that I believe to be speculative. Additionally, the authors only briefly put their research in the context of

other work on the subject matter, both modeling studies and field observations, which further needs to be addressed. Although I acknowledge that implementing processes observed in field observations is not always possible or practical in numerical models, as this model attempts to quantify and reproduce physical processes in the snowpack, far more heedance must be paid to this body of research. I have made specific comments to these issues in the section below.

Although, the paper is not publishable in its present state, I believe that this model, when presented clearly and in a manner that is standard to scientific papers, has the potential to serve as a valuable tool and compliment other models that integrate the dynamics of light absorbing impurities into snowpack evolution and hydrology.

**Scientific Comments**

- Introduction. I would recommend commenting more on the state of hydrological modelling and the need for integrating LAI into these models. Much of the information regarding snow physics can be condensed and put into the methods section.

- Pg 2, Line 14. This is an example of a comment that needs to be cited. Warren and Wiscombe present a model about snow, they do not address BC sources in a comprehensive manner. Mahowald, Ramanathan and Bond are some of the researchers who have explored this topic.

- Pg 2, Line 17. This topic has been discussed in several recent papers including Xu et. al. 2012, Hadely et. al. 2007, Delaney et. al. 2015, Sterle et. al. 2013, Skiles et al. 2016, and Adolph et. al. 2016. Additionally, this topic might be better put in the method section describing scavenging parameters.

- Pg 2, Line 30. I think that Kasapri et al. 2015 did relevant work on this topic.

- Pg 3, Line 3. From what I understand their albedo measurements are largely done with a spectrometer which calculates albedo over a broad range of values. I do not think that 'hypothetical' or 'empirical' are the proper descriptions of their methods.

- Pg 3, Lines 16-27. Here I think this needs to be clearer about the lack of knowledge in this field and the specific accomplishments of this article in reducing this knowledge gap.

- Pg 4, Lines 2-10. Please provide a more detail description of Shyft, are there other papers that have used it? If so, please cite . Also, if appropriate please outline your addition to the model framework here.

- Pg 5, Section 2.2. I think that this section should be condensed and restructured. I found much of the energy balance work to well known and possibly a bit too much detail. Also I believe that your contribution should be clarified from those whose work you implement.

- Pg 8, Lines 10-15. For the description of grain size evolution, did you develop this? or is this from someone else? If so, please cite. Has this method been applied to other studies, if it was, how well did in manifest real snow conditions?

- Pg 9, Lines 13-31. I would recommend looking in to other work about scavenging including Xu et. al. 2012, Delaney et. al. 2015, Sterle et al. 2012, Schwarz et. al. 2013. It is worth noting that the Conway et. al. 1996 experiments used synthetic soot, with properties and particle size distributions that may not occur naturally. Although Conway et. al. 1996 is an important paper, other such work has been done on this subject and should be considered. Additionally, I would recommend moving this section to a part that discusses the sensitivity study.

- Pg 10, I gather that there are 3 parts to your models, the hydrology component,

SNICAR, and your addition. I think the interaction of these components should be better described. Would it be possible to make a figure of this?

- Pg 11, Line2 16-33. A couple sentences from about Pietikäinen et al. 2012 would be good. Although, it is not my field of study I understand that dry deposition rates are quite poorly constrained, could you comment on this? Also, the REMO-HAM simulation period lies outside of the study period. Why?

–

- Pg 13, Lines 24-26. This is an example of statements where a citation must be added. Uncited statements, such as these, are not appropriate in scientific literature and are one of the reasons why I do not believe the paper publishable.

- Pg 14, Lines 11-15. Why is a spin up required? What parameters are modified to calibrate the model?

- Pg 15, Section 5. Put your modeled BC concentrations in the context of other measured concentrations.

- Pg 15, Section 5.1.1. These findings should be put in the context of existing literature. Also, it seems that in your experiments the various cause about 10 days of difference in meltout. Put this in the context of other hydrological modeling methods. Is this an improvement? is this amount of variability standard for say a T-index model?

- Pg 16, Section 5.1.2. Line 20-24. This amplification is far larger than has be documented in some field studies, compare.

- Pg 18, Section 5.2. Is there a BC dataset collected in a similar manner as Sterle et al. 2013, Delaney et al. 2015, Xu et al. 2012, Adolph et al. 2016 that could be used to see how well the model reproduces BC concentrations in the snowpack? Also, what values do you use as background values? Pre-industrial? early

season? Additionally, how do you account for the effects of dust in this case study?

- Pg 18, Line 25. What are reasons for underestimates? 15 m$^3$ s$^{-1}$ is quite a bit.

- Pg 19, Lines 1-24. I found this paragraph hard to follow. I would recommend focusing on the trends as opposed to the specific numbers.

- Pg. 19, Line 25. Why was this time period chosen?

- Pg. 23, Section 6. In the conclusions section, I would recommend adding some comments about the case study.

---

## Author Response (AR1)

**Structural changes and changes with regard to content**

We thanks the reviewers for the significant and helpful comments in order to improve the manuscript.

Common to all reviewers was the request for an improved structure of the manuscript. We lay out the structural changes and major changes regarding content as follows. Further changes to the manuscript are listed point by point under the reviewer's comments further down.

- Introduction
  - Reworded/ according to reviewer's suggestions
    - Adding fundamental literature regarding hydrologic impacts of LAISI
    - A improved description of the gap of knowledge we are targeting
  - Moving snow physics details to methods part of the paper
- Methods (Modeling framework and snowpack algorithm)
  - shortening hydrological framework descriptions; more focus used methods
  - shortening general energy balance descriptions; more focus used methods
  - more focus on implementation of LAISI implementation and coupling to SNICAR
  - adding important literature
- Site description, meteorologic model input and atmospheric deposition data
  - Only minor changes
- Model experiments and calibration
  - General rewording and major shortening of the section
- Discussion
  - Sensitivity study
    - Reran model for Fig. 3A; with reasonable scavenging to lay focus on surface layer thickness impact under otherwise reasonable conditions
    - Reran model for Fig. 4 according to reviewers suggestion with constant BC mixing ratio, but different SWE. Changed metric to percentaged melt period duration compared to clean case.
    - General rewording of the discussion
  - Case study
    - Restructured the discussion; beginning with albedo/surface BC mixing ratio; then radiative forcing, then impacts on hydrology (from cause to effect).
    - Added/restructured/reworded large parts of the discussion, including improved literature comparison.
    - Included a discussion on model improvement. Therefore included observations of discharge in Fig. 9.
    - Added an additional section to discuss model uncertainties
- Conclusion
  - Reworded conclusion

Interactive comment on
"Modelling hydrologic impacts of light absorbing aerosol deposition on snow at the catchment scale"

by Felix N. Matt et al.

Anonymous Referee #1

Responses:

There are four significant issues with the paper that need to be addressed before it is ready for publication:

1) The model independently treats the "wet deposition" of BC (pg 11, lines 12-134) from the deposition of snowfall with which it is nominally associated (pg 11, lines 5-6). This is an issue. Doherty et al. (2014) showed using the CESM/SNICAR model (after which the snow model described herein appears to be very closely modeled) that this results in a factor of 1.5-2.5 high bias in surface snow mixing ratios. The authors should look at this paper, consider the implications for their study, and add analysis/discussion of how this impacts their results (or justify why it doesn't). [Doherty, S. J., C. M. Bitz and M. G. Flanner, 2014: Biases in modeled surface snow BC mixing ratios in prescribed aerosol climate model runs, Atmos. Chem. Phys., 14, 11697-11709, doi:10.5194/acp-14-11697-2014.]

RESPONSE:
This is a very important point, and we agree with including a discussion about the implications for our model. We would like to point out, however, that there are some significant differences in the approach of Doherty and the CESM/SNICAR coupling and our own. CESM/SNICAR couples of land surface model with SNICAR whereas we are interested in coupling a hydrologic rainfall-runoff model. There are similarities, of course, in the treatment of the snow, but over all the two approaches prioritize different objectives. As such, there are some significant differences between our approach and the one Doherty et al. (2014) claim to be problematic:
The high bias in surface snow BC mixing ratios described by Doherty et al. (2014) refers to global climate model simulations with prescribed aerosol deposition rates (wet and dry), where "the input aerosol fields are often interpolated in time from monthly means. Therefore the episodic nature of aerosol deposition in reality (owing to wet deposition) is generally absent in prescribed-aerosol fields." This then results in the high bias, due to the coupling of the interpolated fields with highly variable meteorology (in particular precipitation). In our case study, however, we use deposition fields originating from the regional aerosol climate model REMO-HAM, forced with ERA-Interim reanalysis data at the boundaries. REMO-HAM output is 3-hourly, which we resample to daily means in order to have consistency between the deposition field and the daily observations used as input data in the hydrological simulations. Due to the use of ERA-Interim at the boundaries (which are not far away from Norway) we argue that the REMO-HAM precipitation is realistic and at least on daily scales should reproduce realistic values in terms of BC deposition. The high bias occurring when using interpolated monthly averages as input should therefore be minimized.

We do appreciate this comment, however, and intend to include a more inclusive discussion of these aspects in a revised manuscript.

REVIEW:
We have included a discussion on uncertainties (Sect. 5.2.5), discussing among other the decoupling of BC deposition mass fluxes from precipitation and how this potentially effects our simulation.

2) In the sensitivity study of how varying snowpack depth affects the impact of BC on snowpack melt (Section 5.1.4), snowpack SWE is varied while the total \*mass\* of BC deposited is also kept fixed (see pg. 18, lines 3-5). The justification is that this will "iso-late the impact that the snowpack's SWE has on the effect of ARF in snow." However, that isn't quite correct. BC's impact on snow albedo/forcing/melt rate is a function of the mass mixing ratio of BC in snow water (ng BC per gram of SWE) – not of the total mass in the snow. By increasing SWE but not changing the mass of wet-deposited BC two things are being changed simultaneously: total snowpack SWE (definitively increasing) and the BC mass mixing ratio (definitively decreasing). I'd strongly argue that a better approach would have been to increase BC deposition in proportion to the increase in SWE so one can see to what degree having a "equally-polluted" but deeper snowpack changes the effect of the pollution on melt rate, versus a base case with the same pollution "level" (e.g. BC mass mixing ratio) but a shallower snowpack. Either this sensitivity study needs to be re-run or the paper needs to acknowledge that the results reflect these two simultaneous changes, discuss how this impacts their results, and note that this is not likely physically realistic – which makes me question the robustness of conclusion iii (pg 23, line 20).

RESPONSE:

We acknowledge the suggestions and will evaluate the results with constant mixing ratio to also include this case. We also plan to replace the "meltout days" as metric for impact of SWE with a relative change in meltout days (compare to comment in the "specific comment" section, "pg 18, line 16 and pg 23, lines 20"). However, we feel the comment: "BC's impact on snow albedo/forcing/melt rate is a function of the mass mixing ratio of BC in snow water (ng BC per gram of SWE) – not of the total mass in the snow." does not take into account that over the course of a melt season, BC can accumulate in the top layer and thus the total BC mass in the snowpack can have a large impact on the snow melt. Constant mixing ratio at different SWE would therefor be a different experiment leading to different conclusions, but not necessarily oppose our results. But as we mentioned, we acknowledge that this experiment should be evaluated also in a revised manuscript.

REVIEW:

We have rerun the simulation with snowpacks of constant mixing ratio but different SWE, as suggested from reviewer 1. Furthermore, we replaced "meltout days" as metric with a relative shortening of melt period duration compared to the "no ARF" case. We adapted the discussion of the results accordingly. Furthermore, conclusion (iii) has been reworded.

3) pg 21, lines 17-19: "At the same time, tiles bearing large quantities of snow tend to also bear large quantities of BC (in terms of total BC mass) due to the dominantly wet-depositioned BC, which we chose in the model to follow the same redistribution as snow. Only dry deposition is assumed to deposit spatially homogeneous over the sub grid tiles." If I am reading this correctly, the wet-deposited BC is effectively concentrated only onto snow-covered areas. Thus, as the snow becomes increasingly patchy the remaining snow gets more and more BC mass wet-deposited to it. This is completely unphysical: wet-deposited BC falls to the ground, whether it's covered with snow or not. Perhaps I am misunderstanding and this is just an issue of needing better clarity in the writing: The first sentence indicates the wet-deposited BC "follows the same distribution of the snow": Is this of the snow on the ground, or of the snowfall? If the latter, okay; if the former, as the second sentence seems to imply, I don't understand why this choice would be made since it's not physically reasonable. I'd expect this would significantly affect your results.

RESPONSE:

Yes, this is just a misunderstanding. We will clarify our explanation taking into consideration this comment. When writing we chose wet deposition in the model to follow the same redistribution as snow, we do in fact mean snow fall. We will accordingly rewrite this paragraph to avoid confusion about our methods.

REVIEW:

We have reworded the according paragraph and hope this leads to a better understanding.

4) The paper is difficult to read. Much of it has run-on sentences that are overly convoluted. There are a few specific cases I note below ("Technical corrections") where outright corrections are

needed but much more work is needed beyond this. I would strongly suggest that the co-authors work with the lead author to improve the clarity and conciseness of the writing.

RESPONSE:

This comment appears in all three reviews given – we take it very serious and will accordingly work together to improve the structure and the conciseness of the writing.

REVIEW:

We have reworded and restructured large parts of the paper in order to improve the readability:

- Introduction
  - Reworded/ according to reviewer's suggestions
    - Adding fundamental literature regarding hydrologic impacts of LAISI
    - A improved description of the gap of knowledge we are targeting
  - Moving snow physics details to methods part of the paper
- Methods (Modeling framework and snowpack algorithm)
  - shortening hydrological framework descriptions; more focus used methods
  - shortening general energy balance descriptions; more focus used methods
  - more focus on implementation of LAISI implementation and coupling to SNICAR
  - adding important literature
- Site description, meteorologic model input and atmospheric deposition data
  - Only minor changes
- Model experiments and calibration
  - General rewording and major shortening of the section
- Discussion
  - Sensitivity study
    - Reran model for Fig. 3A; with reasonable scavenging to lay focus on surface layer thickness impact under otherwise reasonable conditions
    - Reran model for Fig. 4 according to reviewers suggestion with constant BC mixing ratio, but different SWE. Changed metric to percentaged melt period duration compared to clean case.
    - General rewording of the discussion
  - Case study
    - Restructured the discussion; beginning with albedo/surface BC mixing ratio; then radiative forcing, then impacts on hydrology (from cause to effect).
    - Added/restructured/reworded large parts of the discussion, including improved literature comparison.
    - Included a discussion on model improvement. Therefore included observations of discharge in Fig. 9.
    - Added an additional section to discuss model uncertainties
- Conclusion
  - Reworded conclusion

Specific comments:

References need to be added to support statements made in a number of places:

a) pg 2, lines 5 through 14 to support a string of assertions

REVIEW:

References added.

b) pg 7, line 10 re: radiative exchanges dominating snow melt in most snow melt scenarios (and perhaps qualify it, too; I'm assuming temperature is the dominant factor for many conditions)

REVIEW:

We have reworded the respective paragraph.

c) pg. 15, line 6 re: LAISI absorbing more efficiently in snow with larger grain size

REVIEW:

This refers to text where the no absorption efficiently with larger grain size is discussed.

We assume pg. 17, line 9 is meant instead. Reference added there.

d) pg. 17, lines 16-17: "Hydrophilic BC absorbs stronger than hydrophobic BC under the same conditions due to an increased MAC compared to hydrophobic BC caused by the ageing of BC during atmospheric transport." (In reality, the degree to which this is the case is not very well-established so references supporting this assertion really are needed.)
RESPONSE:
The required references will be added.
REVIEW:
Added Bond (2006) and Flanner (2007) as references to support the statements and the following one.

pg 2, lines 13-14: organics from combustion and organics in soil also are LAISI and should be added to this list
RESPONSE:
We will add missing LAISI species to the list.
REVIEW:
We completed the list and added references to support our statements.

pg 2, lines 17-19: "Current theory indicates the absorbing effect of LAISI is most efficient when the LAISI reside at or close to the snow surface, and that subsequent snow fall burying the LAISI leads to a decline in or complete loss of the effect." The latter half of this statement is not accurate (and no reference is given to support this assertion). This statement assumes that new snowfall is essentially clean (BC-free), then BC is subsequently deposited on top of the snow. In reality BC is deposited *with* the snow, in wet deposition – at least in the real world. So I don't think "current theory" indicates that it works as stated.
RESPONSE:
Current theory indicates that due to a limited penetration of light in the snow, only LAISI relatively close to the surface is acting decreasing on the albedo. Furthermore, BC can accumulate close to the surface due to sublimation or inefficient melt scavenging, or by dry deposition, an thus exceed LAISI mixing ratios given by that of falling snow. Subsequent snowfall with a lower LAISI mixing ratio in the falling snow than in the surface snow can lead to a burying of layers with higher LAISI mixing ratios (e.g. observed in dust on snow events by Painter et al., 2012).
And even if the subsequent snow has the same mixing ration in BC, the optical grain size is typically smaller in fresh snow, so the effect of LAISI will be less in fresh snow then in the previous older snow – which lead to a decline of the effect after snow fall events.
However, I agree that we should clarify this section to separate what theory actually says (BC closer to the surface absorbs more efficient than BC further down in the snow pack) and what is process related.
REVIEW:
We have reworded the paragraph. Furthermore, we moved this part to the methods (on recommendation of reviewer 3) – and focus in the introduction more on the current state of hydrological models.

pg. 5, lines 29-30: Here and earlier in the test dust, black carbon, volcanic ash and other light-absorbing aerosols are mentioned, but only BC is included in the model. It would be good to be clear that the only LAISI you are currently accounting for in the model is BC.
RESPONSE:
We will do this. We will also add a discussion how the presence of other LAISI would impact our results.
REVIEW:
We now made this clear at the end of the introduction: "a catchment scale analysis of the impact of LAISI, with BC in snow as a proxy for the impact of LAISI"
Furthermore, we added a discussion in Sect. 5.3 on uncertainties, including how the presence of further LAISI species would impact our results.

pg. 6, lines 1-2: You need to be specific about what you mean by a gamma distribution.

RESPONSE:
We will clarify this.
REVIEW:
The lines referred to herein by the reviewer gives only an introduction to the sub-grid tiling approach. The approach is described in detail in Sect. 2.2.2. ("Sub-grid variability in snow depth and snow cover"). There we gave a reference to specify the gamma-distribution.

pg. 8, line 14: What is the basis for using this specific formulation?
RESPONSE:
We accidentally stated the wrong equation in the paper. In an older version of the current snow routine, we were using this equation, which we developed by ourselfs. However, we then changed to a formulation by Taillandier et al. (2007) for dry snow and Brun (1989) for wet snow, on which our here presented model results are based on. This formulation has been used in other studies, e.g. Gabbi et al. (2015). We will change the paper accordingly and add the correct equation and references.
REVIEW: We added the correct equations and references.

pg 8, lines 20-25. I found this discussion of what is "appropriate" for a surface layer thickness to be confusing. Snow doesn't have a defined "surface layer" so it's not as if this is fixed quantity that has some "true" value in the real world. What is appropriate to use for the model surface layer thickness would be a function of what metric you are interested in. Here it could be, for example, the e-folding depth of sunlight penetration, or it could be the depth over which most melt amplification of BC mass is concentrated. Or any number of other things, depending on what you're interested in.
RESPONSE:
We will clarify this in the text  and agree the specifics regarding snow stratigraphy and specifying a surface layer (let alone sampling one!) are difficult.
REVIEW:
We removed the discussion. We base our mid-estimate on Krinner (2006), who suggests this value based on observations of 1~cm thick dirty layers in alpine firn cores used to identify summer horizons. Thus the metric for our estimate is based on  the depth over which most melt amplification of BC mass is concentrated. However, since we expect that the model surface layer strongly determines the BC mixing ratio during melt, we investigate the effect of changing this model parameter (Sect. 5.1.1 Sensitivity to surface layer thickness). See P6, line 31-32.

pg 8, line 24-25: "Since we expect surface concentrations of LAISI in snow to be quite sensitive to the surface layer thickness in our model … " In reality this should only be the case for dry-deposited BC. Wet- eposited BC should be deposited with snow; if the mixing ratio of BC in snowfall were unchanged throughout a new snowfall event the mixing ratio of BC in the surface layer could be completely insensitive to the depth you select for the "surface layer". It should be made clear that surface concentrations of LAISI in the model might be sensitive to the selected surface layer thickness because you are decoupling BC mass deposition and SWE deposition, and not state this as if this were an inherent property of real ambient snowpacks.
RESPONSE:
"In reality this should only be the case for dry-deposited BC":
This is only correct during snow accumulation. However, during melt (as later shown in the sensitivity study), the surface layer thickness strongly defines the effect of melt amplification on the surface layer mixing ratio. However, we should clarify this in the text, and in fact need an improved definition of surface layer as discussed in the prior comment.
REVIEW: We have stated in the text that BC mixing ratio in the surface layer can be sensitive to the thickness of the surface layer due to surface accumulation via dry deposition and melt amplification. See pg. 7, line 1-3.

pg. 9, lines 10-11: "We allow for melt from the bottom layer only when the potential melt per time step is exceeding the maximum depth of the surface layer (both in mm SWE)." It's unclear if you mean that no melt is allowed to occur in the bottom layer until the surface layer is saturated, or if you mean that no melt water is allowed to exit the bottom of the surface layer until the surface layer is saturated.
RESPONSE:
We mean the latter: no melt water is allowed to exit the bottom of the surface layer until the surface layer is saturated. We will clarify this in the text.
REVIEW: We have reworded the paragraph.

pg. 9, line 12: "To date, estimates of the scavenging ratio k are mostly based on experiments conducted by Conway et al. (1996)." Doherty et al. (2013) also estimated the scavenging ratio from ambient snowpacks in two locations. In fact their estimates agreed quite well with that used in Flanner et al.'s SNICAR model – values you seem to adopted here in your model. It would therefore be appropriate to note this, both because there is a study other than Conway et al. (1996) and because their results support the "mid" scavenging values you use.
RESPONSE:
We will adapt the text accordingly.
REVIEW:
We have reworded the paragraph and added Doherty (2013) and further important literature (see pg. 6, line 15-26)

pg. 9, line 24: The assumption of all wet-deposited BC being hydrophilic and all dry-deposited BC being hydrophobic is not justified, either here or in Section 3.
RESPONSE:
The hygroscopicity of BC particles defines which removal process (wet or dry deposition) will be more effective (e.g. Croft et al., 2005). REMO-HAM accounts for this by applying hygroscopicity depended scavenging parameters to aerosols (e.g. Hienola, 2013). From this, we assume that wet deposited BC has the optical properties of aged, hydrophobic BC.

pg. 10, lines 27-28 and Figure 1: It's not at all clear what is meant by "multiplication factors" or how they are used. Figure 1, left panel: coefficient of variation in what? Specify in the figure caption. Figure 1, right panel: It's not at all clear what is being shown here. What are the "factor numbers"? What is the (unlabeled) vertical axis? Why different "factor numbers" for each CV?
RESPONSE:
* Coefficient of variation:
We assume falling snow in a cell is spatially distributed according to gamma distribution, defined by its coefficient of variation (CV). The Cvs of the gamma-distributed snow are taken from Gisnås (2016), who calculated them on a 1x1 km grid over Norway. They represent the spatial snow distribution in the 1x1km cells at snow maximum. To simulate the gamma-distributed snow in a cell, we divide each cell into 10 tiles.

* Multiplication factors:
During snowfall, each of these tiles then gets snow input from falling snow, multiplied with a factor, according to the gamma distribution. These are then ten multiplication factors. These multiplication factors for each tile are constant over time, but vary from cell to cell, according to the CV. The mean of the multiplication factors is 1, so that the mass balance beween falling snow and snow input to a cell is not violated.

* Factor number:
We should rather call this "tile number". We will change this in the figure.

* Vertical axis:
The factor, with which falling snow is multiplied. We will add this to the figure.

* "Why different "factor numbers" for each CV?"
One multiplication factor for each subgrid tile.

We acknowledge that the explanation of our approach needs to be clarified. We will reword the paragraph accordingly.
REVIEW:
We have partly reworded the paragraph and added a key reference describing the approach in detail (Aas et al., 2017). We also added information about the CVs in the caption of Fig. 1, and corrected/added axis labels in the right panel of Fig.1.

pg 12, line 19 / Table 3: Why set radiation to zero during the "accumulation periods"?
RESPOSNE:
The purpose of the accumulation period is only to accumulate a snowpack for the purposes of our sensitivity study. We then slowly melt the snowpack with constant meteorological forcing to explore the parameters specifically in the melt period.
REVIEW:
We have reworded the paragraph. Instead of describing the accumulation phase, we pre-set a snowpack with certain properties at the begin of the melt season and apply them spring time meteorology base on the meteorologic conditions in the Atnsjoen catchment.

pg. 12, line 21: "The forcing applied during the snow accumulation period of 180 days results in 250 mm of SWE at the end of the accumulation period." then pg 12, line 25-28: "After the snow accumulation period, we invoked a time invariant forcing to slowly melt the snowpack until meltout. The forcing applied for melt is based on the average forcing during the melt season from mid March until mid July of the Atnsjoen catchment and results in a melt period of ca. 25-35 days, depending on the scenario applied." I'm quite confused by the use of the term "forcing" here. I would assume you mean radiative forcing, but that would make no sense in the first sentence. Which makes me wonder what you mean by "forcing" in the 2nd and 3rd sentences. Are you calling temperature and precip variations "forcings"? If so, this is quite unconventional, at least for someone from the climate community. Either some explanation or a revision is needed here.
RESPONSE:
In the hydrologic community it is not untypical to simply use the term 'forcing' to refer to the suite of meteorological forcing data. However, we acknowledge this should be clarified and will do so in revised text to better distinguish between the meteorological forcing (radiation, precipitation, temperature, relative humidity, wind speed, aerosol deposition) and the aerosol radiative forcing (which we refer to as the additional absorption of incoming SW radiation due to BC in snow compared to hypothetical clean snow).
REVIEW:
We changed naming from (meteorological) model "forcing" to "model input", and name all forcing connected to LAISI "radiative forcing".

pg 13, Sections 4.1.2 & 4.1.3 and Figure 3c: There aren't different "species" of BC. "Hydrophobic BC" generally refers to fresh – i.e. uncoated – BC, and "hydrophilic BC" is really BC that's been coated. The BC itself in each is essentially the same. I'd suggest a re-wording/re-naming.
RESPONSE:
We were defining "species" of LAISI according to radiative properties – in which the two are different (Flanner, 2012 uses similar wording). However, we can clarify this in the text.
REVIEW:
We decided to leave the naming "species" of BC due to similar naming in other publications (Flanner, 2012; Doherty, 2013). In the introduction, we say that different LAISI have "species-specific radiative properties". By doing this, we define that we differ between species by different radiative properties. This is the case for coated and uncoated BC.

pg 14, lines 8-9: "Bayesian Kriging" This needs a bit of an explanation or at least a reference.
RESPONSE:
We will add the according reference to the paragraph.

REVIEW:
We added the reference.

pg. 14, lines 9-10: "For precipitation, BC deposition rates, wind speed and relative humidity this implies interpolation to the model cells via inverse distance weighting, with a constant vertical gradient applied for precipitation." Do you mean that precip varies with land altitude (with some constant gradient) or that precip is constant with altitude? If the latter, rewording is needed; if the former, some quantification of this vertical gradient is needed.

RESPONSE:
We refer to Førland (1979), who investigated the elevation dependency of precipitation in Norway, and apply a 5% increase in precipitation for every 100 m increase in altitude. We will add the reference to the text and reword the description of our method.

REVIEW:
We reworded the paragraph and added Førland (1979) as reference.

pg. 14, line 11 and Figure 5: a) It's not clear what is meant by a "split sample calibration". b) What is used to calibrate the model? What parameters are varied to achieve the best "calibration" / tuning? c) In Figure 5, the top panel shows data for 2007-2012 with the first three years as "calibration" data. In the bottom panel, these same three years are shown as "validation". Isn't that a bit circular? Or perhaps I don't understand what the difference is between the two panels. d) In Figure 5, it's not stated whether the model is run assuming a perfectly clean snowpack (BC deposition = 0), or something else. (See comment below re: Conclusions section and Figures 5-7.)

RESPONSE:
a) "The split-sample test is a classical test in hydrological modeling, which can be used when sufficient long time series of control data for both calibration and validation period are available and catchment conditions are stationary, which we assume to be true during the simulation period. If the split sample test gives acceptable results, a final calibration can be conducted, making use of the full control data" (from "Distributed Hydrological Modelling; edited by Michael B. Abbott and Jens Christian Refsgaard"; page 50).
The above described procedure is the one we used in our analysis. We will clarify the meaning of the split-sample tests by adding the above reference to the text.
b) We use observed discharge for model calibration. This is mentioned later in the paper (Sect. 5.2.1), but should of course be mentioned here as well. A table with the calibration parameters and the final estimates of the parameters after calibration will be added to the paper.
c) In Fig. 5, both panels show the six years simulation period, form Sept. 2006 – Aug. 2012. Referring to the answer given in a), after receiving acceptable results from the split-sample test (shown in the upper panel of Fig. 5, green curve shows calibration, red curve validation period), we ran a final calibration, making use of the full control data (lower panel of Fig. 5), which results in a similar NSE. By describing the split sample-test in more detail (see a)), we hope that our procedure becomes clear.
d) For model calibration we assume the mid-scenario, since this is what we expect to represent "reality" the best. Max, min and no-ARF scenarios then use different deposition rates and parameters related to the LAISI representation in the snowpack, but otherwise the same settings as used in the model calibration. We will clarify the use of the deposition scenario during calibration in the text.

REVIEW:
- we added a reference describing the split-sample test.
- we stated the algorithm used to estimate parameters, and mention that we calibrate simulated against observed discharge
- we included a table (Table 2) listing all model parameters. In the table, we differ between pre-set parameters (physically based parameters), parameters that have to be estimated during calibration, and parameters set to different values in the min, mid and max scenarios. We also state that we use mid-estimate parameters for the calibration.

pg 15, lines 14-18: "The stronger increase in surface BC in model setups with thinner surface layer is due the inversely proportional relationship of the surface layer thickness with the increase in

impurity concentration under the same mass flux of LAISI into the surface layer (from deposition or melt amplification): halving the surface layer thickness, leaving the mass flux of LAISI into the surface layer unchanged, leads to a doubling of the increase in the LAISI concentration and thus to differences in the vertical distribution of LAISI … " Again, this is only true because you are decoupling BC wet deposition and snowfall deposition. (see comment above re: pg 8, line 24-25)
RESPONSE:
In this paragraph, we try to investigate how the choice model surface layer impacts the BC concentration in the layer and thus the snow melt. In the experiment, we simulate a snow pack with a certain mixing ratio of BC at melt onset until all snow is melted. During the snow melt, no deposition of BC is applied to the snow pack (see Table 3). Thus, the increase in BC mixing ratio in the surface layer is due to **melt amplification solely**, and has nothing to do with decoupling BC wet deposition and snowfall deposition (however, we should mention this specifically in the text, since this obviously led to some confusion). When stating "The stronger increase in surface BC in model setups with thinner surface layer is due the inversely proportional relationship of the surface layer thickness with the increase in impurity concentration under the same mass flux of LAISI into the surface layer (from deposition or melt amplification): ...", we generally describe that any mass input of BC in the surface layer without a mass input of snow, will lead to an increase in the BC mixing ratio, and the increase is inversely proportional to the thickness of the surface layer. This mass input can originate from deposition (in particular dry deposition) or during melt from the bottom layer (melt amplification). However, as mentioned above, in our model experiment no deposition of any kind is applied during the melt phase, and the increase in the surface layer BC mixing ratio is due to the BC mass input from the bottom layer sole.
REVIEW:
We re-wrote the paragraph and address that increase in surface BC is due to melt amplification sole. Furthermore we re-ran the model with scavenging ratios set to mid-estimate values, so that all model parameters except the varying surface thickness are set to mid-estimate or as estimated during calibration. This leads to more realistic model results and allows for a comparison of surface BC increase due to melt amplification with literature values.

pg. 16, lines 8-9: "a doubling of the surface layer LAISI concentration occurs already when the accumulated melt equals the surface layer thickness": "Equals" in terms of what? SWE? Where is this shown? I don't see this in Figure 3.
RESPONSE:
"Equals" is meant in terms of mm SWE. We will add this in the text.
The doubling follows logically from the model representation of the surface layer.
REVIEW:
We re-ran the model with scavenging ratios set to mid-estimate values, so that all model parameters except the varying surface thickness are set to mid-estimate or as estimated during calibration. Since the above mentioned discussion does not make sense in the context of the new simulation, we removed it from the text.

pg. 16, lines 17-18: "the surface concentration of the aerosol simulated strongly depends on the magnitude of the surface layer," Poor wording. What do you mean by the "magnitude" of the surface layer? The surface layer depth?
RESPONSE:
We mean the "surface layer thickness". We will change wording.
REVIEW:
Reworded to  "surface layer thickness".

pg. 16 Section 5.1.2 and Figure 3a: I'm confused by what is shown in Figure 3a middle panel. The BC concentration appears to start at zero, then increase from there. How can the surface snow concentration start at zero? How can there be a "factor increase" for a parameter that starts at zero?
RESPONSE:

This is misleading. The mixing ratio of BC at the begin of the melt season is set to 11 ng/g (which is equivalent with the min-estimate pre-season BC). The curves shown in Figure 3, middle panel don't start at zero, but at 11 ng/g. The point of choosing this value is to show the potential of impact of relatively small concentrations of BC, and at the same time investigate the impact of the model specific parameterization on the impact. However, we missed to mention the pre-season BC mixing ratio in the text of Sect. 4.1. We will add this to the description of the sensitivity study.
REVIEW:
We added it to the the description of the sensitivity study (Sect. 4.1).

pg. 17, lines 13-14: "showing that small amounts of BC in snow can impact the snowpack evolution over the whole melt period even if it undergoes an efficient scavenging process." Up to here no results have been presented that indicate what pre-melt surface snow BC mixing ratios are. This isn't given until pg 20 Section 5.2.3. So the reader really can't know whether a) the model is giving reasonable surface snow mixing ratios of BC and b) what you mean by "small amounts of BC in snow". (Nominally Figure 3 would show this, but as noted above these values all start at zero so it's hard to know what point in the evolution of concentrations you're talking about here).
RESPONSE:
It is correct that we need to mention the pre-melt surface snow BC mixing ratio in the sensitivity study description in Sect. 4.1. As described above, the values don't start at 0.
REVIEW:
We added the pre-melt BC mixing ratio to the the description of the sensitivity study (Sect. 4.1).

pg. 17, line 26: "which we assume to be the most suited": Based on what?
RESPONSE:
"Most suited" in term of "representing reality the best". The parameters of the mid estimate are based on literature values (for a in depth description see e.g. Sect 2.2.2: Aerosols in the snow pack).
REVIEW:
We reworded this part and added Flanner (2007) as reference, who suggests a hydrophilic BC scavenging ratio of 0.2.

pg 18, line 16 and pg 23, lines 20 (bullet item iii): "are less impacted" (pg 18) and "are more prone to be affected (pg 23): By what metric? The "melt shift days"? Is this the only metric of importance? Figure 4 shows the "meltout shift" vs snowpack SWE as a function of different BC scavenging ratios. The meltout shift of 60 days for the deepest snowpack is indeed impressive, but a) we don't know what the BC mass mixing ratios were in this model run and b) we don't know what the total number of melt days is so it's kind of hard to put these results into context. (It must've been at least a few months for the meltout shift result of 60 days. Is this correct??) Perhaps the relative change in number of meltout days (as a percentage?) would be a better metric.
RESPONSE:
We acknowledge that "meltout shift in days" is an insufficient metric. We will accordingly rerun the test and change the metric to "relative change in meltout days".
REVIEW:
We re-ran the model according to the reviewer's suggestions:
  • holding the mixing ratio of BC constant while changing SWE (total mass of BC changes)
  • we use %-change in melt season duration compared to the "clean snow" melt season duration
We reworded the discussion accordingly.

pg. 18, line 23: "NSEs" needs to be defined.
RESPONSE:
NSE is already defined earlier as Nash-Sutcliffe model efficiency.

pg. 18, lines 25 "winter discharge": what time period is "winter" here? Also, Figure 6 does not indicate seasonality. How do we know that the low flow cases are in winter?
RESPONSE:
We refer to "winter discharge" as to the period from circa beginning of November until end of March, when discharge slowly drops to a minimum at the end of the winter season. Here, relatively low flows between 0 and 15 m3 s-1 are predominant, which the model underestimates (see Fig. 5). Even though no seasonally is shown in Fig 6, one can clearly see that low flows are underestimated, as shown in Fig. 5, whereas higher flows are better represented.
We will rephrase this passage of the text, making clear that no seasonality is mentioned in relation to the scatter plot in Fig.6.
REVIEW:
We reworded the paragraph accordingly.

pg. 19, Section 5.2.2.: Again it's difficult to interpret these results since we don't yet know what the model was calculating for surface snow BC mass mixing ratios, and whether they were even vaguely realistic. So I think some re-ordering of the presentation of results is needed.
RESPONSE:
We will change the text accordingly to discussing the BC mixing ratios before the BC impact on discharge and aerosol radiative forcing from BC.
REVIEW:
We restructured the case-study discussion: first albedo/BC mixing ratios, then radiative forcing, then impacts on hydrology (from cause to effect).

pg. 21, lines 26-29: "Qualitatively, we feel this represents reality well, in that if we think about snow patches in a catchment at the end of the season, they tend to be 'dirty', as the concentration of impurities increases while the water melts away." Yes, but in the real world, on which you are basing your observations of reasonableness, this visible darkening of the snow is very likely dirt accumulating, not BC.
RESPONSE:
We acknowledge this reality, and would like to indicate that dirt is a component of LAISI – in the most broad sense. Granted, it was not what this study focused on, but we feel the BC would follow the same general pattern.
REVIEW:
We removed the statement from the text.

pg. 23, lines 24-25: "Even though our model approach is conservative due the lacking implementation of the effect of LAISI on the grain size growth and due to the choice of a remote northern catchment of only medium snow accumulation" It should have been spelled out sooner that grain size growth is not affected by the presence of LAISI (i.e. on pg 8, ~ lines 15-18).
RESPONSE:
We will add this as suggested to Sect. 2.2.2 (Aerosols in the snowpack).
REVIEW:
Since grain size growth depend on the liquid water content, which in turn depends on the melt rate, forcing from BC has in fact an impact on the grain growth. We removed the statement from the text.

Conclusions:

a) Figures 5 - 7 and Results Discussion + Conclusions: The study indicates that inclusion of BC in snow has a significant impact on melt timing (Figure 7). Yet it's not at all clear whether the model calibration and validation (Figure 5) include the effects of BC or are based on using a clean snowpack. I was surprised that there was no testing or discussion of whether including BC in snow improves modeled vs observed catchment outflow volume/timing (Figures 5 & 6).
RESPONSE:

* "Yet it's not at all clear whether the model calibration and validation (Figure 5) include the effects of BC":
The calibration include BC – as mentioned before, we missed to mention this in Sect. 4.2 (Case study model setup and calibration). We will add this to the text.
* "no testing or discussion of whether including BC in snow improves modeled vs observed catchment outflow volume/timing":
We will add and investigation about model improvement and add a discussion part about potential improvement in simulation in the revised manuscipt.
REVIEW:
We have edited Sect. 4.2 (Case study model setup and calibration) to clarify the calibration procedure. Furthermore, we included observation in Fig. 9 and added a discussion on potential for model improvement in Sect. 5.2.4 (BC impact on catchment discharge and snow storage).

b) The discussion totally ignores the fact that real snowpacks have particulate absorbers other than BC. In this regard the impact of BC (pollution) on snow albedo, radiative forcing and melt rates in this study represent an upper limit. If other absorbers – i.e. naturally-occurring dust and dirt – were also included in the model study the impact of adding BC would be less. This needs to be noted and acknowledged.
RESPONSE:
We will add a discussion about how including of further absorbers would impact our results. There are two main points that need to be discussed in this context:
* Including further absorbers would lead to an increase of the total effect of LAISI on snow melt and discharge.
* Including further absorbers would lead to a decrease of the impact of BC.
REVIEW:
We have added a discussion about model uncertainties and discuss there the potential impact of other LAISI species than BC. Sect. 5.3 (Uncertainties).

Technical corrections:

pg 2, line 20: the wording that LAISI "can reappear and retain near to the surface" is both awkward and not accurate. It doesn't "reappear" – it just becomes more concentrated at the surface as the snow water runs out through the snowpack at a higher rate than the BC.
RESPONSE:
We will reword this statement
REVIEW:
We reworded the statement.

pg 4, lines 26: P & E need to be defined when they are first used (even though it's pretty obvious what is meant here...)
RESPONSE:
We will define both.
REVIEW:
We have defined both.

pg 5, lines 14-15: "Furthermore, the presence of a permanent snow layer and snow melt leads to a more challenging identification of periods when the change in liquid water storage is governed by discharge only." I can't figure out what it is you're trying to say here.
RESPONSE:
Kirchner (2009) suggests to estimate the catchment specific parameters $c_1$, $c_2$ and $c_3$ (see Eq. 5) via an analysis of the observed discharge time-series. To estimate the parameters, discharge values are needed at times where both P and E are zero, which he claims to happen mostly during "rainless night hours". When this condition is fulfilled, storage is governed by discharge only

according to Eq. 3. However, "rainless night hours" are not accessible in our dataset since we use daily data. Furthermore, Kirchner analyses discharge of a catchment where contribution to discharge generation from snow melt is rare. In our catchment, snow discharge from snow melt plays a large role, which limits the times, where storage is governed by discharge even more (since it also might be governed by snow melt). For this reason we don't follow Kirchner's approach of estimating the catchment specific parameters $c_1$, $c_2$ and $c_3$ via time series analysis of the observed discharge, but instead use standard model calibration of simulated discharge against observed discharge using the Nash-Sutcliffe model efficiency as objective function.
We will reword the paragraph accordingly.
REVIEW:
Due to suggestions from the other reviewers, we reworded and shortened the general description of the Hydrologic Model Framework, including the description of the Kirchner method (see Sect. 2.1).

pg 18, line 25: "simulated over observed" should be "simulated versus observed"
RESPONSE:
We will reword this.
REVIEW:
We have reworded this.

pg. 20, lines 10-11: "We see the albedo of the max scenario having the largest drop and the one of the no ARF scenario being the lowest." Needs rewording. The *decline* in albedo is smallest; this reads as if the *albedo* is the lowest.
RESPONSE:
We will reword the sentence.
REVIEW:
We have reworded this.

Interactive comment on
"Modelling hydrologic impacts of light absorbing aerosol deposition on snow at the catchment scale"

by Felix N. Matt et al.

Anonymous Referee #2

Responses:

Major revisions are necessary before this paper is to be published. While I appreciate the various sensitivity tests to try and isolate the impact of various parameters as they relate to impact of light absorbing impurities (LAI) in snow in models, some are not executed or interpreted correctly in this study.
RESPONSE:
We acknowledge that the sensitivity study partly needs to be revised and further discussed. Please see the response to general comment 4, and the responses to the specific comments on P12, ln 24-25; P16, ln 21-24 and P18, section 5.1.4 further down for more details.

The implementation of LAI in snow processes in a hydro-power forecasting model, as attempted in this work, is important, as is indeed mostly lacking. However, I don't quite agree with some of the phrasing in the Introduction that claims LAI in snow in hydrologic models (or land surface models that have physically-based hydrologic processes) has been up to now understudied or lacking. Many examples can be found in Qian et al., 2015 (AAS), Light-absorbing particles in snow and Ice: measurement and modeling of climatic and hydrologic impact.
RESPONSE:
We acknowledge a lot of work has been done on the topic of LAI, largely with respect to climate. We are working at a different scale, however, and also – to our knowledge – applying for the first time an 'online' calculation of the albedo response to aerosol deposition in a hydrologic forecasting framework. Prior analyses have applied prescribed albedo forcings, or have used land surface models. Regardless, we will make a better effort to place the work in the context of the extensive existing literature.
REVIEW:
We reviewed Qian et al., 2015 (AAS) plus further literature and accordingly referenced significant prior work.

Generally, here are some of my concerns or things that are unclear based on the current manuscript:

1. Paper is very hard to read, and logic often hard to follow. There are many run-on sentences that are very wordy. Some terms seem to be used to refer to various different processes, and need clarification; e.g. "forcing" is at times used as LAI in snow forcing, whereas during other times it is used to refer to meteorological forcing. Being clearer in explanations is needed.
RESPONSE:
* To "Paper is very hard to read, and logic often hard to follow":
This comment appears in all three reviews given – we take it very serious and will accordingly work together to improve the structure and the conciseness of the writing.
* To "forcing": We use the expression "forcing" to describe the meteorological forcing of the model, in particular the input variables temperature, precipitation, wind speed, relative humidity and aerosols deposition. Furthermore, the expression radiative forcing is used to describe the additional energy uptake from solar radiation by snow due to light absorbing impurities in snow and ice (LAISI), compared to snow with the same properties, but without LAISI. We will clarify those two definitions in the text and replace misleading statements with the correct expressions.

REVIEW:
We re-structured large part of the paper and hope readability is improved (please see general remark no. 2). We also reviewed the paper with focus on run-on sentences and removed wordy sentences. We replaced "forcing" with "model input" when connected to meteorological forcing, and with "radiative forcing" when radiative forcing from BC is addressed.

2. I also found the organization of the paper to be cumbersome, which plays into readability of the manuscript.
a. For one, Section 2 on Modeling Framework is difficult to piece together, to understand how various components of the framework work together, and how the actually set up this modular "model platform for hydrologic purposes". Clear definitions of model setup and model meteorological inputs (sec 2.1) need to be specified.
RESPONSE:
We will work on the readability of Section 2 and clarify how the model is put together. This goes in hand with removing unnecessary repetitions of methods-details, outlined in other studies (e.g. Kirchner, 2009 method in Sect. 2.1). By shortening the methods parts to the core methods we used, we hope to increase the understandability and the readability of this section.
REVIEW:
We have reworded and restructured large parts of the paper in order to improve the readability.
This includes the Model Framework descriptions. In order to lay more focus on our implementations, we shortened the general Model Framework descriptions. Further restructuring is listed as follows:

- Introduction
  - Reworded/ according to reviewer's suggestions
    - Adding fundamental literature regarding hydrologic impacts of LAISI
    - A improved description of the gap of knowledge we are targeting
  - Moving snow physics details to methods part of the paper
- Methods (Modeling framework and snowpack algorithm)
  - shortening hydrological framework descriptions; more focus used methods
  - shortening general energy balance descriptions; more focus used methods
  - more focus on implementation of LAISI implementation and coupling to SNICAR
  - adding important literature
- Site description, meteorologic model input and atmospheric deposition data
  - Only minor changes
- Model experiments and calibration
  - General rewording and major shortening of the section
- Discussion
  - Sensitivity study
    - Reran model for Fig. 3A; with reasonable scavenging to lay focus on surface layer thickness impact under otherwise reasonable conditions
    - Reran model for Fig. 4 according to reviewers suggestion with constant BC mixing ratio, but different SWE. Changed metric to percentaged melt period duration compared to clean case.
    - General rewording of the discussion
  - Case study
    - Restructured the discussion; beginning with albedo/surface BC mixing ratio; then radiative forcing, then impacts on hydrology (from cause to effect).
    - Added/restructured/reworded large parts of the discussion, including improved literature comparison.
    - Included a discussion on model improvement. Therefore included observations of discharge in Fig. 9.
    - Added an additional section to discuss model uncertainties
- Conclusion
  - Reworded conclusion

b. In addition, authors often describe several methods or quote values for parameters, but don't clearly state which they end up using in present study, or what modifications have been made. Reader is left lost in previous studies or potential methods. e.g. sec 2.2.3, or p8 – SNICAR implementation in hydro model not clearly laid out.
RESPONSE:
To give the reader a better overview, we will give a summary table of parameters (+ values) used in the model study. Furthermore we will more specifically line out the methods used.
REVIEW:
During the restructuring process of the "Methods" part of our paper, we focused shortening the methods part and clearly state which equations and parameters we use.

3. While it is true LAI in snow play a significant role on energy and water balance across many mountainous regions throughout the world, the authors don't state if this is in fact a problem in Norway, or for the catchment they chose for the case study. What is the motivation for choosing this catchment, if LAI in snow observations are lacking here, making drawing realistic conclusions difficult (which even they admit e.g. p22 lines 22-23)? Why not do a case study with this new model over a region that does have in situ observation of LAI in snow, e.g. Painter et al, 2012 (Dust radiative forcing in snow of the Upper Colorado River Basin: 1. A 6 year record of energy balance, radiation, and dust concentrations, WRR); or Kaspari et al., 2015 (Accelerated glacier melt on Snow Dome, Mount Olympus, Washington, USA, due to deposition of black and mineral dust from wildfire); or Zhao et al., 2014 (Simulating black carbon and dust and their radiative forcing in seasonal snow: a case study over Northern China with field campaign measurements).
RESPONSE:
The bodies funding our activities, are interested in the potential impact of BC deposition to hydropower operations in Norway and India. We are working in both regions, however, data paucity presents a challenge in both cases. For India, hydrologic data for the regions of interest are challenging to obtain. For Norway, as you mention, there are sparse observations of BC in snow. We selected Norway initially due to the high quality hydrologic data and availability of deposition model output for the region. The modeling and observations available to validate the BC transport were published and found scientifically robust (e.g. Hienola et al., 2013), so we selected to use this region initially.

4. If I understand their modeling framework correctly, BC deposition is decoupled from meteorological forcing applied, making the entire discussion of distribution of wet deposition of BC difficult to rationalize realistically. Authors should at the very least discuss how this decoupling impacts their results and conclusions. I am also a bit concerned with the fact that BC is only deposited in snow during the accumulation phase. In reality, BC deposition, and LAI in general, in snow does not suddenly stop with melt onset, and in fact, certain LAI species such as dust are mostly deposited during the springtime (e.g. Painter et al., 2012, WRR) and therefore during the snowmelt period. This means deposition of LAI in snow during melt season play a significant role in melt magnitude and timing (and not only during accumulation). If deposition of BC in snow tends to indeed occur mostly during accumulation in Norway, or in the catchment in their case study, then authors should state that as an explanation (with appropriate references) for why they set up their experiments the way they did.
RESPONSE:
* To "BC deposition is decoupled from meteorological forcing applied, making the entire discussion of distribution of wet deposition of BC difficult to rationalize realistically. Authors should at the very least discuss how this decoupling impacts their results and conclusions.":
We acknowledge this concern, and in fact it is common to other reviewers. Please see our response to Reviewer #1, comment 1. We will address it within a revised manuscript.
* To "I am also a bit concerned with the fact that BC is only deposited in snow during the accumulation phase"
In our experiments, we aim to show the contribution of different model parameters and settings to the accumulation of LAISI in the top layer and the resulting differences in the response. For this

reason, we try to exclude factors that have the potential to mask the isolated effects or lead to speculative results. One of those factors is the input of aerosol to the snowpack via deposition during the melt period. Furthermore, we in our experiments we investigate the snowpack evolution under idealized conditions, e.g. no precipitation during the melt period. For this reason, we don't expect a large input aerosol from deposition to the snow pack during the melt period, since by far the largest fraction of aerosol deposition is from wet deposition. This idealization is limited to the sensitivity study. In the case study, we use aerosol deposition as prognosed by REMO-HAM on a daily timestep.

REVIEW:
We have included a discussion on uncertainties (Sect. 5.2.5), discussing among others the decoupling of BC deposition mass fluxes from precipitation and how this potentially effects our simulation.

5. Authors use the phrasing of "addition of deposition rates of LAI" throughout manuscript (e.g p1 ln5, p23 ln 8) as a way of communicating the improvements they contribute – this is misleading, as what they really use is the LAI mass and concentration. By using "deposition rate" they suggest they are improving the atmospheric to surface deposition process, the rate and temporal distribution of LAI in snow, when really they are implementing a way for hydrologic model to account for LAI in snow (and in fact actual deposition and precip inputs are decoupled here). This needs to more accurately be represented throughout the paper.

RESPONSE:
We need to clarify our text to explain that we are in fact using deposition rates as input time series to our model. This is in fact original, and as we indicate: "allows for an additional class of input variables" (p1 ln5, p23 ln 8). By stating " new class of input variable", we intended to make clear that we provide the possibility for a new 'forcing variable' but never claim we are "improving the atmospheric to surface deposition process". Clearly, some improvements in our wording our required and will be included in a revised manuscript taking into consideration this comment.

REVIEW:
We have considered this comment when re-wording and re-structering the manuscript. However, by stating that in our implementation we "allow for an additional class of input
variables: the deposition rate of various species of light absorbing aerosols", we feel we accurately describe our contribution. We do not claim to improve atmospheric to surface deposition process, but allow deposition rates of light absorbing aerosols as additional input data to our rainfall-runoff model. This is a new contribution in that there is no study available to date following this approach.

Specific Comments:

P1 ln 12-13: confusing sentence; what is meant by "melt limitation" … this confusing term remains confusing throughout the paper (e.g. p20, ln20). A clearer description of the concept is needed.

RESPONSE:
In our case study, the discharge of the scenarios where BC is applied (ARF scenario) is lower compared to the scenario where no BC deposition is applied (no-ARF scenario), even though the snow albedo is lower in the ARF scenarios. The reason for this is that even though the potential melt in the catchment is higher in the ARF scenarios, the actual melt is lower simply due to a combination of lower snow storage in the catchment and lower snow covered area in the ARF scenarios – the melt is limited by this. This is what we refer to as "melt limitation". We will reword and clarify this in the text.

REVIEW:
We have reworded the statement.

P1, abstract: "Central effect" or "min, max, mid effect estimate" terminology hasn't yet been described, so use in Abstract leads to reader being confused as to what it's referring. Ln 14: "The

central effect estimate produces reasonable surface BC concentrations in snow" The effect produces BC concentrations? Wouldn't BC in snow be the element producing an effect? Re-word sentence with clearer statement.
RESPONSE:
We will reword the sentence and clarify the statement.
REVIEW:
We have removed "min, mid, max estimate" and reworded the statement.

P1, ln 20: what's the difference between "mountainous" and "high mountain" – is there a need for mentioning both environments? Not the same?
RESPONSE:
We will remove "high mountain".
REVIEW:
Removed

P1 ln 24: "affected areas" is not adequate use of the phrase – these areas didn't experience an extreme event, and were not "affected". Suggest removing word "affected"
RESPONSE:
We will remove the word "affected".
REVIEW:
Removed

P2 ln 5-7 and ln 7-9: need references.
RESPONSE:
We will add the required references to the revised paper.
REVIEW:
P2 ln 5-7: added Anderson, 1976
P2 ln 7-9: added Warren and Wiscombe, 1980 and Flanner, 2006

P2 Ln 18-19: statement needs reference.
RESPONSE:
We will add the according reference.
REVIEW:
Reference added. Furthermore, this part has been moved to the methods part and information was densified (recommended by rev 3).

P2 Ln 24-28: What's the point of that lit review? How does it impact the present work? What did you take from it, or how did you improve it?
RESPONSE:
Flanner (2007) estimated scavenging ratio's for hydrophilic and hydrophobic BC based on work done by Conway et al. (1996). We use the values estimated by Flanner (2007) in our study to simulate melt scavenging of BC (see Eq. 15 and 16 in Sect. 2.2.2, Aerosols in the snowpack).
REVIEW:
This part has been moved to the methods part, where we think is it more suitable (also recommended by rev 3). Stated in the methods, the link to where we use this information should be clear.

P3 ln 1-2 and that paragraph in general: "investigating the impact of LAISI on the snow melt and runoff predominantly use empirical formulations to investigate the impact of LAISI on the radiative forcing in snow, by observing the net surface shortwave fluxes over snow and identifying the contribution from the LAISI through determination of the (hypothetical) clean snow albedo" – inaccurate, misrepresents previous work and the context of this work. There have been several

improvements to LAI in snow representation in hydrologic or land surface models in recent years (albeit further developments continue to be needed), e.g. Zhao et al, 2014 (ACP), Oaida et al., 2015 (JGR), and see Qian et al., 2015 (AAS) for a more complete list and overview of observations and modeling of LAI in snow. Authors of present study show reframe their motivation or gap their new work is filling given these previous developments.
RESPONSE:
As mentioned above, we acknowledge that the introduction needs to be revised and a broader overview about significant contribution from recent literature needs to be given. We will provide this in the revised paper. However, including the literature in the here stated comment, there is to date no hydrologic catchment model allowing deposition of LAISI as additional meteorologic forcing. The motivation of our work is thus justified and we are convinced to fill an important gap with the contribution of our model.
REVIEW:
We have re-structured and re-written the second half of the introduction according to the recommendations of the reviewers (included the here criticized statement). Part of this is an extended literature review and reframing of our motivation.

P3 ln 13-15: Entire sentence is awkward; what is meant by "complex abstractions"?
RESPONSE:
Maybe a better choice would be "increasing complex representation of the physical processes"
REVIEW:
We have reworded the sentence.

Sec 2.1: clearly state what variables the model needs as inputs (this is later alluded to on page 11, but needs to be more clearly stated under Modeling Framework section)
RESPONSE:
Meteorological forcing: Temperature, precipitation, wind speed, relative humidity, radiation, aerosol deposition.
We will add this to Sect. 2.1.
REVIEW:
We have added this to Sect. 2.1.

P4 ln 11: what is meant by "efficient simulation"?
RESPONSE:
With "it is optimized for highly efficient simulation" we mean "computational efficient", in the sense that it uses computational resources very efficient (or in other words: simulations run fast). We will clarify this in the text.
REVIEW:
The sentence has been removed during the restructuring process.

P4 ln 12: why is ET module important? The whole modeling setup needs to be more clearly defined and laid out
RESPONSE:
Yes, we acknowledge an improved discussion of the model framework is required.
REVIEW:
The sentence has been removed during the restructuring process.

P8 ln 1-9: how exactly did you integrate SNICAR within hydrologic framework? Which variables in 2.2.1 were updated by SNICAR output … and what does SNICAR output?
RESPONSE:
Our SNICAR implementation calculates the broadband hemispheric reflactance of snow ("**snow albedo**") as function of

* snow optical grain size
* solar zenith angle
* thickness of the snow layers (in mm SWE)
* mixing ratios of hydrophilic and hydrophobic BC in each of the layers
which are calculated each time step by the snow routine. SNICAR is called in an intermediate step and used to update the snow albedo, before the time step's energy and mass balance is calculated. We will clarify this in the text.
REVIEW:
We have added this to the text

P8 10-17: how is r connected to radiative transfer model? Where is r used in your implementation?
RESPONSE:
r is the optical grain size of snow, one of the input variables to SNICAR. The snow albedo strongly depends on r. We will add to the text what role r plays in determining the snow albedo.
REVIEW:
We have clarified this in the text.

P10 ln 9-14: A bit unclear how these tiles are defined? Is it based on elevation? Also, "In our model, we further developed an approach assuming that the spatial distribution of each single event of solid precipitation follows a certain probability distribution function." This newer approach is based on which previous method? What did you further develop?
RESPONSE:
The tiles are a representation of subgrid snowpacks, used to represent the subgrid snow distribution. Each solid precipitation event is assigned to those tiles, according to a multiplication factor. The multiplication factor for each tile is based on a gamma distribution, assuming that the the subgrid spatial distribution of precipitation is well represented by this distribution. The coefficient of variation of each grid cell, which defines the gamma distribution, originates from work done by Gisnås et al. (2016) [Small-scale variation of snow in a regional permafrost model]. The method is similar to the one used in Aas et al. (2017) [A Tiling Approach to Represent Subgrid Snow Variability in Coupled Land Surface–Atmosphere Models].
We will try to describe this more clear in the text and add the missing Aas et al. (2017) reference.
REVIEW:
We have added Aas et al. (2017) as referance, who describes the approach in detail. We furthermore reworded parts of the section in order to clarify the approach.
We also changed the axis label of right Fig. 1.

P10: the concept of "multiplication factor" is not quite clear.
RESPONSE:
As described in the comment above, we will re-write this paragraph and describe the concept in more detail. The  Aas et al. (2017) reference also should help to clarify the concept.
REVIEW:
See above.

P11, ln 17-30+: is the REMO simulation ran offline, separately from the hydrologic model? Is there a discrepancy between deposition timing in REMO and hydrologic model meteorological precipitation input/events? How does that affect your study? (Also see General Comment 4 above).
RESPONSE:
The REMO simulation ran offline, separately from the hydrologic model. We acknowledge that we should include a discussion about precipitation timing in REMO and the used observations in the hydrological model, and the resulting implications for our study.
REVIEW:
See above.

P11, sec 3.1: what is the simulation period for hydrologic model, vs for REMO? Might want to even state the hydrologic model simulation period more clearly a bit earlier in the paper, in the intro to Section 3, before 3.1.
RESPONSE:
Hydrological model: 01.09.2006 to 31.08.2012
REMO-Ham: 01.07.2004 – 31.12.2014
The discrepancy between the two periods might lead to some confusion. The important information is that the REMO-Ham simulation period covers the hydrologic simulation period – we might only state the hydrologic simulation period in the paper, and that we have full coverage of this period from REMO-HAM simulation.
REVIEW:
We reworded the respective text in Section 3.1 so that there is no confusion about periods discrepancy.

P12 ln 10-12: run on sentence. Please revise.
RESPONSE:
We will reword the sentence.
REVIEW:
We have reworded the sentence.

P12, ln 24-25: why did you chose to only deposit BC during accumulation period, and not throughout entire simulation period, or at least during both accumulation and ablation periods? Also see General Comment 4.
RESPONSE:
Since we melt the snowpacks under idealized conditions, e.g. undisturbed from precipitation (sold and liquid), this is realistic in the scenario in the sense that BC input mostly happens as wet-deposition, and as such during precipitation events. The idealized conditions are required to identify the contribution of certain model concepts to the evolution of BC concentration and impact on melt. However, we should discuss this during this in the text and also discuss the implications of idealized versus real conditions.
In the case study, we use of course continuous input data from Remo-HAM.

P13, ln 9-11: Sentence should be better integrated, and phrased more grammatically correct.
RESPONSE:
We will reword the sentence.
REVIEW:
The sentence has been removed during rewording and restructuring of the section.

P14 ln 2-3: is BC distributed throughout the top layer, or entire snowpack?
RESPONSE:
The BC is uniformly distributed in the snow at melt onset, such that the mixing ratio of BC is the same in both layers. We will clarify this in the text.
REVIEW:
We clarified this in the text.
On recommendation of reviewer 1 (see general comment no. 2 of reviewer 1), we have changed the model setup for this section (no called "Sensitivity to snowpack SWE at melt-onset"). Instead of holding the total mass of BC constant while changing SWE, we hold the mixing ratio of BC constant while changing SWE.

P14, ln 19: what do you mean by "all free model parameters"?
RESPONSE:

Model parameters/tuning parameters that are estimated during the calibration process of the simulation. We will clarify this in the text.
REVIEW:
We have added an overview table (Table 2), giving an overview over all model parameters estimated during calibration. We also have reworded the sentence referred to above.

P15, ln 4-6: confusing sentence. "The central graph in Fig. 3a shows that the choice of the maximum surface layer [insert "thickness"] strongly determines the increase in the [insert "magnitude of"] surface concentration over the melt season - leading to a strong increase in surface BC until [insert "through"] the end of the melt season with an increase in BC by a factor of circa 15, 30 and 60 for maximum layer thicknesses of 4.0, 8.0 and 16.0 mm, respectively, compared to the pre-melt season BC concentration."
RESPONSE:
We will reword the sentence accordingly.
REVIEW:
We have reworded the sentence.

P15, ln 4-9: These 2 sentences, if I understand correctly, seem to be at odds with each other: the latter, "The thinner the surface [...]" implies that the 4mm layer selection would have the strongest effect, yet it's only increasing BC by a factor of 15, smallest of them all.
RESPONSE:
Thank you for catching this error. The correct statement is: "... increase in BC by a factor of circa 15, 30 and 60 for maximum layer thicknesses of 16.0, 8.0 and 4.0 mm, respectively,..." instead of of "... 4.0, 8.0 and 16.0 mm, respectively, ..." (as shown in Fig. 3a). We will change this in the paper.
REVIEW:
We have reworded the sentence.

P15, ln 19: "the mean radiative intensity diminishes with depth due to absorption in snow and LAISI and scattering, leading to a less effective absorption of LAISI in deeper snow. " needs a reference
RESPONSE:
e.g. Warren and Wiscombe (1980) [A Model for the Spectral Albedo of Snow. II: Snow Containing Atmospheric Aerosols] and Flanner et al. (2007) [Present-day climate forcing and response from black carbon in snow]. We will add a reference to support this statement.
REVIEW:
We have reworded paragraph and added references where needed.

P16, ln 21-24: what about new BC deposition? You mention on the ways the output of LAI from snowpack affects end of season LAISI amount, but what about the input, which may vary through time, and which again brings me back to general comment 4.
RESPONSE:
Again, this is a good comment and it should be discussed in the paper. However, since we use idealized conditions with a melt period which is not interrupted with neither snow nor rain events, the exclusion of BC input to the snow pack is arguable, since the main mechanism contributing to BC input in the snow pack is due to wet-deposition.
REVIEW:
We have added a short discussion about neglecting dry deposition and  snowfall during the melt period in Section 5.1.1.

P18, section 5.1.4: The number of earlier meltout should probably be scaled by total length of meltout season of each snowpack to more realistically and accurately represent the impact of snowpack thickness

RESPONSE:
This is correct. We plan to include this in the paper.
REVIEW:
Done.

P19, ln 1-24: this entire section is rather convoluted and the conclusions not easy to follow. Because of that, some of the results seem at odds with each other. Please reorganize and be more concise in your analysis. … ln 7: "total sum of daily discharge" refers to net annual sum of runoff? And it is about zero? Yet later in the paragraph the % change increases? Perhaps I am misunderstanding the stats – a more clear explanation would be helpful. One idea is to also put all these values in a table, for easier comparison. You mention ET, is there a plot to support the conclusions you are mentioning?

RESPONSE:
"total sum of daily discharge" refers to the sum of daily discharge over the simulation period (so the sum over several years, not only the annual sum). This is the same for all scenarios. Our argue is then that it follows that the impact on the ET between the different scenarios is negligible – but we can look deeper into this and support our argument with a plot.
Furthermore, we see that differences in discharge of our ARF scenarios to the no ARF are counter balancing, meaning that a decrease of discharge in the beginning of the melt season is followed by a decrease later in the melt season (comparing ARF with the no-ARF scenario). By splitting up the melt season into those two periods, we quantify these increases/decreases. This is visualized in Fig. 7b.
REVIEW:
We have reworded the paragraph and added an overview table (Table 3).

P19, ln 30: wouldn't the 1.5, 5.1, and 10.3 mm values be negative?
REVIEW:
We have corrected this.

P20 ln 3-8: I am not quite sure what you are trying to say about having an analysis at the catchment scale. The links you are trying to draw don't seem that obvious or easy to follow.
RESPONSE:
We acknowledge that the paragraph requires rewording and a more clear explanation of our intentions.
REVIEW:
We have reworded the paragraph.

P21 ln 2-4: scavenging ratio is not the only factor determining if BC accumulated in top snow layer. What about new snow?
RESPONSE:
This is correct, and we will mention this in the text. However, the during the melt period (and that's what we refer to here), fresh snowfall doesn't play a large role, as one can see in the continuous drop of SWE during the melt season in Fig 7a).
REVIEW:
We have added a short discussion about the role of new sow during the melt season.

P21, ln 26: "Qualitatively, [...]" – this sentence is not a very strong, supported, conclusion.
RESPONSE:
We will revise this sentence (also compare with comment on "pg. 21, lines 26-29" of reviewer #1).
REVIEW:
We have removed the statement.

P21, ln 13-19: reason (iii) is rather confusing. The whole concept of "wet deposition" of BC in this explanation doesn't quite add up for me when this study has BC and precipitation "falling" separately (processes decoupled). It's possible I am misunderstanding the explanation, which might suggest a more clear explanation would help.
RESPONSE:
Even though we use decoupled precipitation and wet deposition, we expect observed daily precipitation (used in the hydrologic model as meteorological forcing) and wet-deposition from REMO-HAM to be  consistent. Since we calculate BC mixing ratios in falling snow before redistributing it to the tile level, we think that the discussion referred to herein is legitimate (also compare with the response to comment 1 of reviewer #1). We do appreciate this comment, however, and intend to include a more inclusive investigation and discussion of these aspects in a revised manuscript.
REVIEW:
We have reworded the paragraph and hope argumentation is more clear now.

P23, ln 1-5: I would argue that normalizing SCF isn't necessarily more relevant to impact on runoff, as total surface albedo (both snow and snow-free surfaces) influences snowmelt thought the snow-albedo feedback.
RESPONSE:
Since our model is not coupled to an atmospheric model, no feedback between the land surface and the atmosphere is represented. Thus, the albedo of snow-free surfaces does not impact runoff through the snow-albedo feedback. However, the evapotranspiration is impacted – which then has implications for the discharge generation. We will ad this to our discussion.
REVIEW:
We have reworded the discussion of this section and focused on trying to clarify what we mean by normalizing with SCF.

P23, ln 12-15: "The maximum thickness (in SWE) of the surface layer herein has rather little effect on the snow albedo and melt rate as long as the maximum layer thickness is sufficiently small." – is this clean snow, or LAISI case? "However, the evolution of the LAISI surface concentration is highly sensitive to the choice of the surface layer extent." If LAISI concentration is affected by snowpack thickness, then wouldn't snow thickness, somewhat indirectly, affect albedo, since surface snow layer LAI impact snow albedo?
RESPONSE:
To " is this clean snow, or LAISI case?":
LAISI case. We will clarify this in the revised manuscript.
To "wouldn't snow thickness, somewhat indirectly, affect albedo, since surface snow layer LAI impact snow albedo?"
This is correct – the choice of the maximum layer thickness has an impact on the snow albedo – primarily due to LAISI accumulation in the surface layer during melt. We discuss this in the sensitivity study (see Sect. 5.1.1).
REVIEW:
We have reworded the paragraph in order to clarify this.

P23 ln 27-29: I am not sure the evidence presented is enough to conclude improvement in hydrologic modeling. The shift found by comparing LAISI and no-LAISI scenarios certainly suggests an impact of LAISI on discharge timing, but one would have to compare LAISI, no-LAISI, and observed runoff over same period of time to conclude that a hydrologic model with LAISI processes present brings simulated runoff closer to observations, over the no-LAISI simulated runoff. You could add no-LAISI discharge to figure 5 to have a more robust conclusion on model improvement.

RESPONSE:
We acknowledge the weakness in our conclusion and will consider the suggestion for improving our reasoning in the revised manuscript.
REVIEW:
We have added further analysis and how that including BC helps to minimize the volume error in discharge during the spring time (see Sect. 5.2.4)

Technical Corrections:

"LAISI in snow" is used in several parts of paper (e.g. p8 ln 18), which is redundant since LAISI already contains "in snow" by their own definition. Please revise.
RESPONSE:
We will remove "in snow".
REVIEW:
We have removed "in snow" wherever we used it in combination with "LAISI".

P4 ln 2: too many "hydrological"/"hydrologic"/'hydropower" terms in one sentence. Please revise.
RESPONSE:
We will revise the sentence.
REVIEW:
We reworded the sentence.

P5 ln 27: "central addition" is awkward. "Main addition"?
RESPONSE:
We will replace "central".
REVIEW:
We reworded to "Main addition".

P15: word "Stronger" is repeated 2x. Remove one.
RESPONSE:
We will remove one.
REVIEW:
Removed.

P22 ln 20: "are" is repeated 2x back to back
RESPONSE:
We will remove one.
REVIEW:
Removed.

Interactive comment on
"Modelling hydrologic impacts of light absorbing aerosol deposition on snow at the catchment scale"

by Felix N. Matt et al.

Anonymous Referee #3

General comments

My knowledge of hydrological models is not broad, so I do not believe I am qualified to comment on the viability and implementation of the model. However, I have commented on the structure, content and more scientific issues that I see in this article. My first criticism is that the paper is long and should be shortened and restructured.

RESPONSE:
We received criticism about the structure and length about our manuscript from all referees – and take this criticism accordingly serious. We work together and will reword and restructure parts of the paper. This includes:
* Rewording the introduction, with focus on citing recent published literature that is of interest for the here presented paper, and that have been missing in the first submission. Also, we will move some of the aspects (espacially LAI in snow physics) mentioned in the introduction to the methods part.
* Shortening the methods part: We will lay more focus on our approach of handling LAI in the snowpack and the implementation of SNICAR in our model and remove large parts that are not necessary for the here discussed implementation.
* Furthermore, we will focusing on putting our research in  the context of other work.

REVIEW:
We have shortened (24 pages of text to 20 pages of text) and restructured the paper in order to improve the readability. In order to lay more focus on our implementations, we shortened the general  Model Framework descriptions. Further restructuring is listed as follows:
- Introduction
  - Reworded/ according to reviewer's suggestions
    - Adding fundamental literature regarding hydrologic impacts of LAISI
    - A improved description of the gap of knowledge we are targeting
  - Moving snow physics details to methods part of the paper
- Methods (Modeling framework and snowpack algorithm)
  - shortening hydrological framework descriptions; more focus used methods
  - shortening general energy balance descriptions; more focus used methods
  - more focus on implementation of LAISI implementation and coupling to SNICAR
  - adding important literature
- Site description, meteorologic model input and atmospheric deposition data
  - Only minor changes
- Model experiments and calibration
  - General rewording and major shortening of the section
- Discussion
  - Sensitivity study
    - Reran model for Fig. 3A; with reasonable scavenging to lay focus on surface layer thickness impact under otherwise reasonable conditions
    - Reran model for Fig. 4 according to reviewers suggestion with constant BC mixing ratio, but different SWE. Changed metric to percentaged melt period duration compared to clean case.
    - General rewording of the discussion
  - Case study

- Restructured the discussion; beginning with albedo/surface BC mixing ratio; then radiative forcing, then impacts on hydrology (from cause to effect).
- Added/restructured/reworded large parts of the discussion, including improved literature comparison.
- Included a discussion on model improvement. Therefore included observations of discharge in Fig. 9.
- Added an additional section to discuss model uncertainties
- Conclusion
  Reworded conclusion

Secondly, I did not find a useful quantification of how LAI from the ARF model are integrated in to the snowpack, as no field measurements of LAI from the area are available.
RESPONSE:
We calculate BC mixing ratios in snow from wet- and dry-deposition fields determined with REMO-HAM. Similar REMO-HAM simulations (similar setup and same region) and observations available to validate the aerosol transport were published and found scientifically robust (see Hienola et al., 2013). However, we acknowledge the need for a better discussion of our results – especially the magnitude of order of BC mixing ratios in the surface layer throughout the melt season in the case study. This includes the comparison with observations of BC mixing ratio in snow collected in the proximity of our study region (e.g. Forsström et al., 2013).
REVIEW:
We have restructured and reworded our discussion on surface BC mixing ratios, including further comparison with literature.

Also, better parameterization of dust sources is needed.
RESPONSE:
We use BC as only source in our simulations – however, we will add a discussion how this simplification is impacting our results – in particular how this is influencing the impact of BC on albedo and snow melt (additional other LAISI lower the impact of BC) and the overall impact of LAISI on BC (the overall impact of LAISI on albedo and snow melt would be higher than our model suggests).
REVIEW:
We have included a discussion on uncertainties (Sect. 5.2.5), discussing among other things how the presence of further LAISI species would impact our results.

My final criticism, and one I take very seriously, is that the authors fail to cite and recognize substantial research that has been done in this field, leading to comments in the text that I believe to be speculative. Additionally, the authors only briefly put their research in the context of other work on the subject matter, both modeling studies and field observations, which further needs to be addressed. Although I acknowledge that implementing processes observed in field observations is not always possible or practical in numerical models, as this model attempts to quantify and reproduce physical processes in the snowpack, far more heedance must be paid to this body of research.
RESPONSE:
We will review and reword our paper with a focus on excluding "speculative" conclusions. Furthermore, we will work on our discussion part of the paper to better put our research in the context of published work on the subject matter.
REVIEW:
We have restructured the introduction, including and extended literature review and reviewed the paper for speculative comment. Furthermore, we extended the discussion part to better put our contribution in the context of other work.

I have made specific comments to these issues in the section below. Although, the paper is not publishable in its present state, I believe that this model when presented clearly and in a manner that is standard to scientific papers, has the potential to serve as a valuable tool and compliment

other models that integrate the dynamics of light absorbing impurities into snowpack evolution and hydrology.
RESPONSE:
Thank you for recognizing our contribution. We do feel strongly this is a unique contribution, and one that is missing presently from the hydrologic modeling community (more so than from the climate modeling community). We hope to address this deficit.

Scientific Comments

Introduction. I would recommend commenting more on the state of hydrological modelling and the need for integrating LAI into these models. Much of the information regarding snow physics can be condensed and put into the methods section.
RESPONSE:
We will reword the introduction and put more focus on the state of LAI implementation in hydrological model and the gap of knowledge that we address.
REVIEW:
We have restructured the introduction and focused on the state of hydrological models and the need for LAISI integration. Parts describing snow physics in detail (especially scavenging of LAISI with melt) have been moved to the methods part.

Pg 2, Line 14. This is an example of a comment that needs to be cited. Warren and Wiscombe present a model about snow, they do not address BC sources in a comprehensive manner. Mahowald, Ramanathan and Bond are some of the researchers who have explored this topic.
RESPONSE:
We will give an appropriate reference for this statement.
REVIEW:
We added appropriate references for this statement.

Pg 2, Line 17. This topic has been discussed in several recent papers including Xu et. al. 2012, Hadely et. al. 2007, Delaney et. al. 2015, Sterle et. al. 2013, Skiles et al. 2016, and Adolph et. al. 2016. Additionally, this topic might be better put in the method section describing scavenging parameters.
RESPONSE:
We will move most of the paragraph (including the here cited statement) into the methods part. We will furthermore put the topic into a broader context, including the above mentioned references.
REVIEW:
We put this topic to the methods, shortened the paragraph and added references where suited.

Pg 2, Line 30. I think that Kasapri et al. 2015 did relevant work on this topic.
RESPONSE:
Assuming Kaspari et al. (Accelerated Glacier Melt on Snowdome, Mt. Olympus, Washington due to Deposition of Black Carbon and Mineral Dust from Wildfire) we acknowledge similarities in topic. However, they estimated the impact on snow melt and runoff by doing a "first-order estimate of the impact on snowmelt by doing a simple energy analysis." which is quite distinct from this work. We respect, however, it should be cited.
REVIEW:
We have included Kaspari et al. (2015) In our introduction.

Pg 3, Line 3. From what I understand their albedo measurements are largely done with a spectrometer which calculates albedo over a broad range of values. I do not think that 'hypothetical' or 'empirical' are the proper descriptions of their methods.
RESPONSE:

Their results are in fact empirical using the definition: "a relationship supported by experiment and observation". We only wish to show that the prior approaches are using observations and prescribing albedo changes, rather than including on online calculation of albedo based on LAISI deposition rates. We will refine the text in consideration of this comment.
REVIEW:
We have reworded the paragraph.

Pg 3, Lines 16-27. Here I think this needs to be clearer about the lack of knowledge in this field and the specific accomplishments of this article in reducing this knowledge gap.
RESPONSE:
We will revise and reword the paragraph with focus on the specific accomplishments of this article in reducing the raised knowledge gap.
REVIEW:
We partly reworded the paragraph. Further up in the introduction we focused on better demonstrate the knowledge gap we are addressing.

Pg 4, Lines 2-10. Please provide a more detail description of Shyft, are there other papers that have used it? If so, please cite . Also, if appropriate please outline your addition to the model framework here.
RESPONSE:
SHyFT is a new Hydrological model framework developed by Statkraft (https://github.com/statkraft/shyft). We are currently working on a manuscript.
REVIEW:
A long overview over the Shyft framework would lead to a loss of readability. For this reason, we focus on clearly stating which methods we are using in the present study. We restructured and reworded large parts of the methods with a clear focus on our contribution.

Pg 5, Section 2.2. I think that this section should be condensed and restructured. I found much of the energy balance work to well known and possibly a bit too much detail. Also I believe that your contribution should be clarified from those whose work you implement.
RESPONSE:
This will be part of the restructuring and rewording of the methods part. Since many of the Energy balance formulations are well know, we will shorten this part of the methods an focus on the description of our implementation to the existing model framework.
REVIEW:
We shortened the general energy balance description and focused on clearly stating our contribution.

Pg 8, Lines 10-15. For the description of grain size evolution, did you develop this? or is this from someone else? If so, please cite. Has this method been applied to other studies, if it was, how well did in manifest real snow conditions?
RESPONSE:
We accidentally stated the wrong equation in the paper. In an older version of the current snow routine, we were using this equation, which we developed by ourselfs. However, we then changed to a formulation by Taillandier et al. (2007) for dry snow and Brun (1989) for wet snow, on which our here presented model results are based on. This formulation has been used in other studies, e.g. Gabbi et al. (2015). We will change the paper accordingly and add the correct equation and references.
REVIEW:
We have included the correct equation and references.

Pg 9, Lines 13-31. I would recommend looking into other work about scavenging including Xu et. al. 2012, Delaney et. al. 2015, Sterle et al. 2012, Schwarz et. al. 2013. It is worth noting that the Conway et. al. 1996 experiments used synthetic soot, with properties and particle size distributions that may not occur naturally. Although Conway et. al. 1996 is an important paper, other such work

has been done on this subject and should be considered. Additionally, I would recommend moving this section to a part that discusses the sensitivity study.

RESPONSE:

We will move most of the here described to the discussion of the scavenging ratio sensitivity study, including a discussion of the above listed literature.

REVIEW:

We have shortly raised the thematic in the methods to state scavenging ratio estimates we are using in our study. We then discuss the topic in the sensitivity study.

Pg 10, I gather that there are 3 parts to your models, the hydrology component, SNICAR, and your addition. I think the interaction of these components should be better described. Would it be possible to make a figure of this?

RESPONSE:

This is correct. SHyFT provides the model stack, which defines the hydrological model. We exchanged the "default" snow-routine in the model stack with the snow routine we developed. A part of our new snow routine is the couplinig of to SNICAR: The snow routine handles alongside standard energy balance and mass balance calculations the mixing ratio of aerosols in the snow pack, the zenith angle of the sun and the optical grain size of snow, which are input to SNICAR. From this, SNICAR calculates and returns the broadband albedo of snow – which is then used in the energy balance calculations of the snow routine.

We will make this clear by adding a more detailed description and consider to  support our description with a sketch of the coupling.

REVIEW:

We shortened the methods description, especially the description of the general model framework since it only constitutes the computational infrastructure. Instead we focus on describing the methods we use for our analysis and the development of our implementation.

Pg 11, Line2 16-33. A couple sentences from about Pietikäinen et al. 2012 would be good. Although, it is not my field of study I understand that dry deposition rates are quite poorly constrained, could you comment on this? Also, the REMO-HAM simulation period lies outside of the study period. Why?

RESPONSE:

We will add some more detailed information about REMO-HAM from Pietikäinen et al. 2012 and a short discussion about limitations (e.g. problems with dry deposition handling in REMO-HAM) and how this potentially effects our results.

" Also, the REMO-HAM simulation period lies outside of the study period. Why?":

Simulations with REMO-HAM, which is used to calculate deposition rates offline, are conducted for the period 01.07.2004 – 31.12.2014. The hydrologic simulations for the case study, using the deposition rates from REMO-HAM as input, are conducted from September 2006 to September 2012. Thus the REMO-HAM output covers the total time period of the hydrologic simulations. However, we acknowledge that the mismatch in dates can lead to confusion. For this reason, we will reword the REMO-HAM simulation description.

REVIEW:

- We have added more information about  REMO-HAM (see beginning of section 3.1)

- We reworded parts of the paragraph to clarify the date mismatch.

- To "poorly constrained dry deposition rates": The dry deposition model uses some measurement based parameters for different species, but there are very few measurement overall and not so many over different surfaces. This, for many species (like aerosols) the parameters have been more or less guessed based on measurement for gases, for example. So it is true that dry deposition has error sources, but so does the whole model.

Pg 13, Lines 24-26. This is an example of statements where a citation must be added. Uncited statements, such as these, are not appropriate in scientific literature and are one of the reasons why I do not believe the paper publishable.

RESPONSE:

We will add the appropriate reference. We also will review and reword our paper with a focus on excluding uncited statements as referred to herein.
REVIEW:
We added the reference.

Pg 14, Lines 11-15. Why is a spin up required? What parameters are modified to calibrate the model?
RESPONSE:
We use a spinup time of one year (1 September 2005 to 31 August 2006) to in order to achieve good estimates for the model state variables. We will add this to the revised manuscript.
Furthermore, there is a mixup of dates in this paragraph: First we write we use a "study period of 6 years, from September 2006 to September 2012" (Pg 14, Line 6). Later we write, we run the model until October 31 (Pg 14, Lines 11-12). We will state the correct dates in the revised manuscript.
REVIEW:
We added an overview table with all model parameters listed (Table 2).

Pg 15, Section 5. Put your modeled BC concentrations in the context of other measured concentrations.
RESPONSE:
We will do this in the revised manuscript.
REVIEW:
We added an extended discussion about the modeled surface BC mixing ratios and increase during spring time including comparison with other model studies and field observations.

Pg 15, Section 5.1.1. These findings should be put in the context of existing literature. Also, it seems that in your experiments the various cause about 10 days of difference in meltout. Put this in the context of other hydrological modeling methods. Is this an improvement? is this amount of variability standard for say a T-index model?
RESPONSE:
We will add a discussion to put this in a broader context in the revised manuscript.
REVIEW:
We reran the model to study the sensitivity of surface layer thickness variation. In the old manuscript, we assumed that all BC stays in the snowpack. Since this is not a realistic assumption, we now use the mid-estimates for scavenging instead to be able to compare results to the literature. We put our findings in the context of Doherty (2013), Sterle (2013) and others.
In the sesitivity study, we don't discuss a potential improvement of the model. We do this in the case study, where we show that we can reduce the discharge volume error during spring time by including aerosols in the snowpack.

Pg 16, Section 5.1.2. Line 20-24. This amplification is far larger than has be documented in some field studies, compare.
RESPONSE:
To identify the isolated effect of the maximum model surface layer thickness, we chose to set the scavenging ratio to 0 (all LAISI stays in the snowpack during melt). This does not necessarily result in realistic results but demonstrates that results can significantly depend on the choice of this parameter. However, we acknowledge that we need to discuss our model experiment results in a broader context. We will do this in the revised manuscript.

Pg 18, Section 5.2. Is there a BC dataset collected in a similar manner as Sterle et al. 2013, Delaney et al. 2015, Xu et al. 2012, Adolph et al. 2016 that could be used to see how well the model reproduces BC concentrations in the snowpack? Also, what values do you use as

background values? Pre-industrial? Early season? Additionally, how do you account for the effects of dust in this case study?

RESPONSE:

To "BC dataset":

There is no data specifically on the Atnsjoen catchment. However, there is data from Scandinavia available that can allow evaluation of the here presented results (e.g. Forsström 2013; Elemental carbon measurements in European Arctic snow packs) due to the proximity of our study region and the sampling site. We will add an extended discussion about this in the revised manuscript.

To "What values do you use as background values":

"No ARF scenario" refers to a scenario, in which deposition of BC is set to zero, simulating a hypothetical clean snowpack. Results from these runs are used to identify the contribution of BC to snowmelt and discharge generation.

To "effects of dust in this case study":

We don't include dust in our study. However, we will include a discussion on how this affects our results in the revised manuscript (see also comment response to comment to "conclusions b" of reviewer #1).

REVIEW:

We have re-structured and rewritten large parts of this section and extended the comparison with other studies.

Pg 18, Line 25. What are reasons for underestimates? 15 m3 s−1 is quite a bit.

RESPONSE:

There is actually not an underestimating of 15 m3 s-1, but the model underestimates flows where the observation shows flows between 0-15 m3 s-1. The reason for this might be that the parameters chosen for Kirchner are not perfect for the low decrease of discharge during winter.

REVIEW:

We have reworded the respective paragraph

Pg 19, Lines 1-24. I found this paragraph hard to follow. I would recommend focusing on the trends as opposed to the specific numbers.

RESPONSE:

We will consider this suggestion in the revised manuscript.

REVIEW: We have reworded this paragraph and added an overview table with all numbers instead of naming them in the text.

Pg. 19, Line 25. Why was this time period chosen?

RESPONSE:

We refer to this time period as "melt season" because of the drop from snow maximum to no snow in the catchment during this time.

Pg. 23, Section 6. In the conclusions section, I would recommend adding some comments about the case study.

RESPONSE:

We will add this in the revised manuscript.

REVIEW:

We have reworded the conclusions and added summarizing comments on the case study.

[revised manuscript text omitted]

We first present an overview over the hydrological model used in this study and the newly developed snow algorithm to treat LAISI in the snowpack in Sect. 2. A description of the catchment used for our study and the  input data sets is given in Sect. 3. Sect. 4 describes the 1-D model experiments and the model settings and calibration process in the case study. Lastly our results are presented together with the discussion  first for the model experiments, followed by the case study within Sect. 5.

**2 Modeling framework and the snowpack algorithm**

In the following section we provide descriptions of the hydrologic model (Sect. 2.1) and the formulation of a novel snowpack module used for the analyses (Sect. 2.2).

**2.1 Hydrologic Model Framework**

For the  analysis, we use Statkraft's hydrologic forecasting toolbox Shyft; https://github.com/statkraft/shyft), a model framework developed for hydropower forecasting. The concept of Shyft follows the idea that a hydrological model can be expressed as a sequence of well known routines, each describing a certain aspect of the represented hydrological processes. Which processes are represented depend on the purpose

of the model and the requirements of the user. The sequence of routines, the so called "methods-stack", is then run on a cell by cell basis, where the cell loosely represents an area of similar time-invariant geographical data (e.g. topographic properties or land type) with no specific restriction to cell geometry or area.  The Shyft framework allows for  the paradigm

5 of distributed,  conceptual parameter models, and more physically based approaches.  Standard model input variables are temperature, precipitation, wind speed, relative humidity and shortwave radiation. The methods-stack used herein consists of (i) a single-equation implementation to determine the potential evapotranspiration, (ii) a newly developed snowpack algorithm using an online radiative transfer solution for

10 snow to account for the effect of LAISI on the snow albedo, and (iii) a first order nonlinear differential equation to calculate the catchment response to precipitation, snow melt and evapotranspiration. (i) and (iii) are described in more detail herein, while (ii) is described in detail in Sect. 2.2.

To determine the potential evapotranspiration, $E_{pot}$, we use the method according to Priestley and Taylor (1972)

$$E_{pot} = \frac{\alpha}{\lambda} \frac{a}{\lambda} \cdot \frac{s(T_a)}{s(T_a) + \gamma} \cdot R_n \tag{1}$$

15 with  $a$ = 1.26 being a dimensionless empirical multiplier, $\gamma$ the psychrometric constant, $s(T_a)$ the slope of the relationship between the saturation vapour pressure and the temperature $T_a$, $\lambda$ the latent heat of vaporization and $R_n$ the net radiation.

The catchment response to precipitation and snow melt is determined using the approach of Kirchner (2009), who describes catchment discharge from a simple first order nonlinear differential equation.

20 $$Q = f(S)$$

 Following Kirchner's suggestion, we solve the log transformed formulation

$$\frac{d(ln(Q))}{dt} = g(Q)(\frac{P - E}{Q} - 1) \tag{2}$$

due to numerical instabilities of the original formulation. In Eq. (2), $Q$ is the catchment discharge,

25

$$\frac{dS}{dt} = P - E - Q$$

$$\frac{dQ}{dt} = g(Q)(P - E - Q),$$

where $g(Q)$ (called the "sensitivity function") is the derivative with respect to $S$ of the inverse of $f(S)$. $g(Q)$ can be estimated from the observed discharge alone for periods of the discharge time series for which the catchment precipitation ($P$) and evapotranspiration ($E$) can be neglected. Kirchner (2009) uses the discharge time series of two catchments governed by humid climate and mild, snow poor winters (the Plynlimon catchments in mid-Wales; for more information see Robinson et al. (2013) )

5 and recession plots to estimates $E$ the evapotranspiration, and $P$ the precipitation.

We assume that the sensitivity function, $g(Q)$. He finds, has the same form as described in Kirchner (2009) :

$$ln(g(Q)) \approx c_1 + c_2 ln(qQ) + c_3 (ln(Q))^2 \tag{3}$$

with $c_1$, $c_2$ and $c_3$ being the only catchment specific parameters. To then solve Eq. (??) numerically using Eq. (3), Kirchner suggests to log-transform Eq. (??) due to a "smoother" profile of the log-transformed function:

$$\frac{d(ln(Q))}{dt} = \frac{1}{Q}\frac{dQ}{dt} = g(Q)(\frac{P-E}{Q} - 1)$$

, which we estimate by standard model calibration of simulated discharge against observed discharge. In contrast to Kirchner Kirchner (20 approach, we apply a slight adjustment. Firstly, we use the outflow response use the liquid water outflow from the snow routine described in Sect. 2.2 instead of precipitation , $P$ , to integrate in Eq. (2) . This outflow (Kirchner (2009) used snow-free catchments in his analysis). The outflow from the snow routine can be liquid precipitation, melt water, or a combination of 15 both. In the catchments used by Kirchner (2009) "persistent snow cover is rare". For this reason, a contribution to the liquid water storage from snow melt is not considered in Eq. (??). Our study catchment is a high mountain catchment in Norway with a long lasting snow cover (typically until end of June; see Sect. 3). Thus, during spring and partly during summer, snow melt significantly contributes to the change in the liquid water storage, making the aforementioned adaptation necessary. Furthermore, the presence of a permanent snow layer and snow melt leads to a more challenging identification of periods when 20 the change in liquid water storage is governed by discharge only.

Secondly, we assume that the sensitivity function, $g(Q)$, has the same form as described in Kirchner (2009) (see Eq. (3)) and estimate the parameters $c_1$, $c_2$ and $c_3$ by standard model calibration of simulated discharge against observed discharge using the Nash-Sutcliffe model efficiency as objective function, rather than using recession plots. Since we use a daily time step in our simulation, the identification of periods with negligible storage contribution from precipitation (

25 **2.2 A new snowpack module for LAISI**

To account for snow in the model, we developed a snow-algorithm to solve the energy balance

$$\frac{\delta F}{\delta t} = K_{in}(1-\alpha) + L_{in} + L_{out} + H_s + H_l + R \tag{4}$$

with the incoming shortwave radiation flux $K_{in}$, the incoming and outgoing longwave radiation fluxes $L_in$ and $L_in$, the sensible and latent heat fluxes $H_s$ and $H_l$, and the heat contribution from rain $R$ (fluxes are considered to be positive when directed into the snowpack and as such an energy source to the snowpack). $\frac{\delta F}{\delta t}$ is the net energy flux into (or out of) the snowpack (fluxes are considered to be positive when directed into the snowpack).

$L_{in}$ and $L_{out}$ are calculated using the Stephan-Boltzmann law, with $L_{in}$ depending on the air temperature $T_a$ and $L_{out}$ on the snow surface temperature $T_{ss}$, calculated as $T_{ss} = 1.16 \cdot T_a - 2.09$ (Hegdahl et al., 2016) . The latent and sensible heat fluxes are calculated using a bulk-transfer approach that depends on wind speed, temperature and relative humidity (Hegdahl et al., 2016) .

**2.3**

 The main addition provided in the algorithm described herein is the implementation of a radiative transfer solution dynamically.  for the dynamical calculation of snow albedo, $\alpha$. The implementation allows a new class of  model input variables, wet and dry deposition rates of light absorbing aerosols . From this, the model is able to simulate the impact of dust, black carbon, volcanic ash or other aerosol deposition on snow albedo, snow melt and runoff. To account for the mass balance of LAISI in the snowpack while maintaining a representation of sub-grid snow variability and snow cover fraction (SCF), the energy balance based snow algorithm underlies a tiling approach, where a grid-cell's snowfall is apportioned to sub-grid units following a gamma distribution.

In the following we present: (i) an  introduction to the radiative transfer calculations required to represent LAISI in the snowpack (Sect. 2.2.1), and  ii) the sub-gridscale tiling approach to represent snowpack spatial variability (Sect. 2.2.2).

**2.2.1**

**2.2.1 Aerosols in the snowpack**

$$\frac{\delta F}{\delta t} = K + L + H_s + H_l + R$$

 Wiscombe and Warren (1980) and Warren and Wiscombe (198 a robust and elegant model for snow albedo that remains today as a standard. Critical to their approach was the ability to account

for: (i) wide variability in ice absorption with wavelength, (ii) the forward scattering of snow grains, and (iii) both diffuse and direct beam radiation at the surface. Furthermore, and of particular importance to the success of the approach, the

5

 model relies on observable parameters.

$$K = K_{in} - K_{out} = K_{in}(1 - \alpha)$$

10

$$L = L_{in} - L_{out} = \epsilon_a \sigma T_a^4 - \epsilon_s \sigma T_s^4$$

 Both the albedo of clean snow and the effect of LAISI on the snow albedo strongly depend on the snow grain effective radius (or optical grain size) $r$ (Warren and Wiscombe, 1980), which alters as snow

15 ~~above the surface and $\epsilon_a$ is then called the effective clear sky emissivity of the atmosphere (e.g. Unsworth and Monteith (1975) ) . In our model approach, $T_s$ is calculated as a function of the air temperature ($T_a$) rather than resolving heat conduction in multiple snow layers. Raleigh et al. (2013) found a high correlation between the air temperature measured at standard heights above the surface and $T_s$ at various study sites with different characteristics. Following his finding, we assume a linear relationship between $T_a$ and $T_s$:~~

20 $$T_s = m + n \cdot T_a$$

$$\epsilon_a = a \cdot (e_a/T_a)^b.$$

25

$$e_a = e_s \cdot r_h$$

ages. $r$ can be related to the specific surface area (SSA), representing the ratio of surface area per unit mass of the snow grain (Roy et al., 2013),

$$r = \frac{3}{\rho_{ice} \cdot SSA} \tag{5}$$

5   with $\rho_{ice}$ the density of ice.

In our model, we compute the evolution of SSA in dry snow following Taillandier et al. (2007) as

$$SSA(t) = [0.629 \cdot SSA_0 - 15.0 \cdot (T_s - 11.2)] - [0.076 \cdot SSA_0 - 1.76 \cdot (T_s - 2.96)]$$
$$\ln \left\{ t + \exp \left( \frac{-0.371 \cdot SSA_0 - 15.0 \cdot (T_s - 11.2)}{0.076 \cdot SSA_0 - 1.76 \cdot (T_s - 2.96)} \right) \right\}, \tag{6}$$

10   where $e_s$ is the equilibrium water pressure and $r_h$ $t$ is the relative humidity. The latter is a common variable measured at meteorological observation stations. An approximation for $e_s$ over water and ice is given by Bosen (1960) and Bosen (1964) . Radiative exchanges dominate the snow melt rate in most snow melt scenarios. However, age of the snow layer (hours), $SSA_0$ is the fluxes of sensible and latent heat often contribute significantly due to vertical gradients in the air temperature and the vapour pressure. They are largely due to turbulent exchange processes and as such strongly dependent on the wind

15   speed. The physically consistent determination of $H_s$ $SSA$ at $t$=0 (cm$^2$ g$^{-1}$), and $H_l$ over snow is rather difficult and requires complex instrumentation (e.g., Eddy Correlation Method). Various attempts have been made to ease the calculation (e.g., Gray and Male, 1981) ; we have followed Anderson (1976) and employ a bulk-transfer approach to approximate the turbulent fluxes of sensible and latent heat as functions of wind speed, temperature and air humidity, where the impact of the wind speed is represented in a linear, two-parametric wind-function. The parameters of the wind function (intercept and

20   slope) are then determined by model calibration.

For the calculation of the heat contribution from rain $R$, we assume that rain falling on top of snow is cooled from atmospheric temperature $T_a$ to the freezing temperature of water $T_f$, releasing the sensible heat $T_s$ is the snow temperature (°C). The evolution of SSA in wet snow is calculated according to Eq. 5 and Brun (1989) as

$$R = \rho_w c_w \Delta r (T_a - T_f) = \frac{C_1 + C_2 \cdot \Theta^3}{r^2 \cdot 4\pi}, \tag{7}$$

25   where $\rho_w$ and $c_w$ are the density and heat capacity of water, respectively. If Eq. (4) results in an energy surplus, we assume that the surplus is consumed by snow melt, expressed in snow water equivalent (SWE), less the change in the cold content of the top 30 mm of SWE of the snowpack.

**2.2.2   Aerosols in the snowpack**

Wiscombe and Warren (1980) and Warren and Wiscombe (1980) developed a robust and elegant model for snow albedo that remains today as a standard. Critical to their approach was the ability to account for: (i) wide variability in ice absorption with wavelength, (ii) the forward scattering of snow grains, and (iii)both diffuse and direct beam radiation at the surface. Furthermore, and of particular importance to the success of the approach, the model relies on observable parameters $C_1$=1.1·

5 $10^{-3}$ mm$^3$ d$^{-1}$ and $C_2 = 3.7\cdot10^{-5}$ mm$^3$ d$^{-1}$ are empirical coefficients. $\Theta$ is the liquid water content of snow in mass percentage. $SSA_0$ is set to 73.0 m$^2$ kg$^{-1}$ (Domine et al., 2007) and we set the minimum snowfall required to reset the SAA to 5 mm snow water equivalent (SWE).

To solve for the effect of light absorption of LAISI in the snowpack on the snow albedo, we have integrated a two-layer adaption of the Snow, Ice, and Aerosol Radiative (SNICAR) model (Flanner et al., 2007, 2009) into the energy and mass budget

10 calculationsof Sect. ??. . By providing the solar zenith angle of the sun, the optical grain size $r$ of snow, mixing ratios of LAISI in the snow layers and SWE of each layer, SNICAR is calculates the snow albedo for a number of spectral bands. To achive this, SNICAR utilizes the theory from Wiscombe and Warren (1980) and the two-stream, multilayer radiative approximation of Toon et al. (1989). Following Flanner et al. (2007), our implementation of SNICAR uses five spectral bands (0.3-0.7, 0.7-1.0, 1.0-1.2, 1.2-1.5, and 1.5-5.0 um) in order to maintain computational efficiency, and individual broadband optical ice and aerosol

15 properties were weighted by incident solar flux following the Chandrasekhar mean approach (Thomas and Stamnes, 1999) . The incident flux were simulated offline assuming mid-latitude winter clear- and cloudy-sky conditions. Flanner et al. (2007) compared results from 5 bands scheme to the default 470 bands scheme in SNICAR and concluded that relative errors are less than 0.5%. The incident flux were simulated offline assuming mid-latitude winter clear- and cloudy-sky conditions.

Both the albedo of clean snow and the The absorbing effect of LAISI on the snow albedo strongly depend on the snow grain

20 effective radius (or optical grain size) $r$. The snow grain effective radius $r$ in turn alters as snow ages. To represent the effect of snow ageing on the evolution of the snow grain effective radius, we use a fast exponential limited growth for air temperatures above 0°C and a slow linear growth for air temperatures below or equal to 0°C:

with $r_t$ is most efficient when the LAISI reside at or close to the snow surface (Warren and Wiscombe, 1980) . As snow melts LAISI can remain near the surface due to inefficient melt scavenging, which leads to an increase in the near surface

25 concentration of LAISI and thus a further decrease in the snow albedo; the so called melt amplification (e.g., Xu et al., 2012; Doherty et al., Field observations suggest that the magnitude of this effect is determined by the particle size and the hydrophobicity of the respective LAISI (Doherty et al., 2013) . Conway et al. (1996) observed vertical redistribution and the effect on the snow albedo by adding volcanic ash and hydrophilic and hydrophobic BC to the snow surface of a natural snowpack. Flanner et al. (2007) used the results from Conway et al. to determine the scavenging ratios, specifying the ratio of BC contained in the melting snow that

30 is flushed out with the melt water, of both hydrophilic and hydrophobic BC. They found the scavenging ratio for hydrophobic BC, $k_{phob}$, to be 0.03, and $r_{t-1}$ being the snow grain effective radius at time $t$ and $t-1$, respectively, $r_{min}$ and $r_{max}$ the snow grain effective radius of fresh and old snowfor hydrophilic BC, $k_{phil}$, respectively, and $d_{fast}$ and $d_{slow}$ the fast and the slow growth rates, which are determined by model calibration0.2. Doherty et al. (2013) found similar results by observing BC mixing ratios close to the surface of melting snow. Recent studies report efficient removal of BC with melt

35 water (Lazarcik et al., 2017) , revealing large gaps in the understanding of the process.

 To represent the evolution of LAISI mixing ratio near the snow surface, we treat LAISI in two layers in our model: (i) a surface layer with a time invariant maximum depth (in mm SWE), where the concentration of each LAISI species is calculated from a uniform mixing of the layer's snow with  either falling snow with a certain mixing ratio of aerosol (wet deposition) or aerosol from atmospheric dry  deposition; and (ii) a bottom layer, representing the snow exceeding the maximum depth of the surface layer.  Following Krinner et al. (2006), we apply a maximum surface layer thickness of 8 mm . Krinner et al. (2006) suggests this value based on observations of 1 cm thick dirty layers in alpine firn cores used to identify summer horizons.  Due to potential accumulation of LAISI in surface snow via dry deposition and melt amplification, we expect the simulated surface mixing ratios of LAISI to be sensitive to the surface layer thickness of our model . For this reason, we use a factor of 2 to the maximal surface layer thickness  to account for the uncertainty.

 To allow for melt amplification in the model, we include LAISI mass fluxes between the two layers during snow accumulation and snow melt.  Generalizing Jacobson (2004)'s representation of LAISI mass loss due to meltwater scavenging  for multiple snow layers (Flanner et al., 2007), we characterize the magnitude of melt scavenging using the scavenging ratio $k$ and calculate the temporal change of BC mass $m_s$ in the surface layer as

$$\frac{dm_s}{dt} = -kq_sc_s + D, \tag{8}$$

and the change of BC mass $m_b$ in the bottom layer as

$$\frac{dm_b}{dt} = k(q_sc_s - q_bc_b)_{~,}. \tag{9}$$

 Herein, $q_s$ and $q_b$ are the mass fluxes of melt water from the surface to the bottom layer and out of the bottom layer, respectively, and $c_s$ and $c_b$ are the mass mixing ratios of BC in the respective layer. $D$ is the atmospheric deposition mass flux. A value for $k$ of <1 is equal to a scavenging efficiency of less than 100% and hence allows for accumulation of LAISI in the surface layer  during melt. In our

 analysis, we account for hydrophobic and hydrophilic

5 ~~and initial and final BC mass in the top 2 cm. Using the $k_{phob}/k_{phil}$ ratio from analysis of observations in the top 50 cm of snow, he estimated $k_{phil}$ to 0.2. To account for the uncertainty in the estimations, Flanner et al. (2007) used a order of magnitude variation on these estimates. These uncertainty might seem large, however, Flanner et al. 's calculations of the scavenging ratios of hydrophilic and hydrophobic BC are based on only one dataset (presented in Conway et al. (1996) ), and accurate measurements that allow an uncertainty estimate of the scavenging don't exist to the knowledge of the authors.~~

10  by distinguishing between  the type of deposition mechanism (hydrophilic BC predominantly from wet deposition, hydrophobic BC for dry deposition ). By following Flanner et al. (2007) , we set $k_{phob}$ to 0.03 and $k_{phil}$ to 0.2, and account for the large uncertainty by using an order of magnitude variation on $k_{phob}$

15 and $k_{phil}$. Like Flanner et al. (2007) , we treat aged, hydrophilic BC as sulphate coated to account for the net increase in the mass absorption cross section (MAC) by 1.5 at $\lambda$=550 nm compared to hydrophobic BC caused by the ageing of BC (reducing effect on MAC) and particle coating from condensation of weakly absorbing compounds (enhancing effect on MAC) suggested by Bond et al. (2006). As a consequence, hydrophilic BC absorbs stronger than hydrophobic BC under the same conditions. On the other hand, hydrophilic BC  undergoes a more efficient melt scavenging. The competing mechanisms are subjects

20 of the 1-D sensitivity study in Sect. 5.1.3.

**2.2.2 Sub-grid variability in snow depth and snow cover**

The representation of sub-grid snow variability can play a key role in modelling the hydrology of areas with a seasonal snow-pack (e.g., Hartmann et al., 1999). Several approaches exist to capture the sub-grid snow covered fraction (SCF) and distribution of SWE. Statistical approaches often use so called snow depletion curves to describe a relationship be-

25 tween a prognostic snow variable (e.g SWE, accumulated melt depth) and regional observations of SCF,  (e.g., Liston, 2004; Luce and Tarboton, 2004; Kolberg and Gottschalk, 2010) . However, such approaches do not allow for explicit treatment of snow layers, which is required when simulating the  mixing ratios of LAISI. In our model, we

30 follow (Aas et al., 2017) by assuming that the sub-grid spatial distribution of each single event of solid precipitation follows a certain probability distribution function. From this distribution we calculate multiplication factors, which then are used to assign the snowfall of a model grid cell to a number of  sub-grid computational elements, the so called tiles  (Aas et al., 2017) . The snow algorithm described  herein is executed for each of the tiles separately. This implies that variables related to the snow state, such as SWE, liquid

water content, impurity content, and snow albedo  differ among the tiles. This also allows to simulate the  sub-grid variability in impurity content. To calculate the multiplication factors, we  assume that the  sub-grid redistributed snow follows a gamma distribution (see e.g., Kolberg and Gottschalk, 2010; Gisnås et al., 2016) , deter-

5    mined by the coefficient of variation (CV). CV values were derived based on work done by Gisnås et al. (2016), who used Winstral and Marks (2002)'s terrain-based parametrization to model snow redistribution in Norway by accounting for wind effects during the snow accumulation period over a digital elevation model with 10 m resolution.  Gisnås et al. (2016) calibrated the redistribution model with snow depth data from Airborne Laser Scanning (ALS) over the Hardangervidda mountain plateau (see Melvold and Skaugen (2013)) and evaluated with snow depth data from ground

10    penetrating radar observations at Finse, both located in Southern Norway. The detailed scheme is described in Gisnås et al. (2016). In the case study presented in Sect. 5.2, we use the CV values from Gisnås et al. (2016) to derive a linear relationship between the model cell's elevation and the corresponding CV value by simple linear regression (see  Fig.1a), which results in a $R^2$-value of 0.71 and a p-value of smaller than 2.0e-5 for the study area. The linear relationship is only applied to cells with an areal forest cover fraction of lower than or equal to 0.5. For cells with a forest cover fraction of higher than 0.5, a constant

15    snow CV value of 0.17 is used, following the findings of Liston (2004) for high latitude, mountainous forest. Examples of multiplication factors for forested cells and forest free cells  for a different CV values are shown in  Fig. 1b.

**3   Site description, meteorologic  model input and atmospheric deposition data**

We selected the unregulated upper Atna catchment for our analysis. This catchment is located in a high elevation region

[revised manuscript text omitted]

In the snow algorithm used in this study, dry deposition and sedimentation are treated the same way. Herein, dry deposition  refers to the sum of REMO-HAM dry deposition and sedimentation.

**4 Model experiments and calibration**

Our analysis is in two parts in Sect. 5. First we present a 1-D sensitivity study investigating the impact of parameters and variables specific to the algorithm determining the effect of LAISI (Sect. 5.1). We then demonstrate the significance of BC in snow radiative forcing on the catchment scale in a case study by simulating the impact of wet and dry deposition of BC in a remote south Norwegian catchment (Sect. 5.2).

We assume uncertainties of the LAISI radiative forcing to originate mainly from the model representation of surface layer thickness, melt scavenging of BC, and uncertainties in the deposition  input data. To account for the uncertainties, we declare minimum (min), central (mid), and maximum (max) effect estimates to each of the critical parameters, outlined together with further model parameters in Table ??. The min, mid, and max estimates are both subjects of analysis in the sensitivity study (further described in Sect. 4.1) and used in the case study to give an uncertainty estimate of the LAISI effect on the hydrologic variables (further described in Sect. 4.2). We investigate the impact of BC impurities on the response variables by comparing the results from Aerosol Radiative Forcing model experiments ("ARF" scenarios) to simulations in which all BC deposition rates are set to zero ("no-ARF" scenario).

**4.1 1-D sensitivity study experiments**

For the 1-D sensitivity study presented in Sect. 5.1, we use synthetic ~~forcing data according to Table ??. The forcing data is divided into two periods, the snow accumulation period and the snow melt period, and held constant during each of the periods. The forcing applied during the snow accumulation period of 180 days results in 250 mm of SWE at the end of the accumulation period. This value is representative of the mean SWE of the upper 50of tiles (factor Nr. 5 to 10 in right Fig. 1) at winter snow maximum in the Atnsjoen catchment during the study period of the case study . Deposition rates during the snow accumulation period were set to the average BC deposition rate during snow accumulation periods in the Atnsjoen catchment simulated with the regional aerosol-climate model REMO-HAM (see Sect. 3.1). After the snow accumulation period, we invoked a time invariant forcing to slowly melt the snowpack until meltout. The forcing applied for melt is based on the average forcing during the melt season from mid March until mid July of the Atnsjoen catchment and results in a melt period of ca. 25-35 days, depending on the scenario applied. This is in the range of the average time period it takes from snow maximum in a tile to meltout averaged over all snow tiles and melt seasons in the Atnsjoen catchment. For the melt period, different model setups are applied, investigating how the snowpack evolution depends onof the model (Sect. 5.1.1)of BC (Sect. 5.1.2), sole,BC species (hydrophobic or hydrophilic; Sect. 5.1.3) and (iv) the amount of snow at melt season start (Sect. 5.1.4). For simplicity and comparability~~

  impact of the scavenging ratio with respect to the BC species. Furthermore, we investigate how LAISI impact snowpacks of different depths, but same LAISI mixing ratio at melt onset. We run the model with model parameters as outlined in Table **??** if not otherwise specified.

5 .

**4.1.1**

 The model input applied for melting is based on the average meteorological conditions during the melt season from mid March until mid July of the

10  Atnsjoen catchment. In our sensitivity experimants, all snowpacks have 250 mm SWE

15

**4.1.1**

20    of snow with a mixing ratio of 35 ng g$^{-1}$ in both surface and bottom layer at melt onset. These values are representative of the upper 50% of tiles at winter snow maximum in the Atnsjoen catchment during the study period of the case study

25  . During the melt period, we exclude fresh snowfall and dry deposition, in order to isolate the effect of the

**4.1.1**

30

**4.1.1**

5  tested model parameters on the snowpack evolution under melt conditions. This might lead to an underestimation of total BC mass in the snow  column.

**4.2 Case study model setup and calibration**

 We investigate the impact of BC aerosol deposition on the catchment

10 hydrology of a Norwegian catchment over a study period of 6 years, from September 2006 to September 2012. The station based  input data described above is interpolated to the simulation cells ( 1x1 km$^2$ and accordingly smaller cells at the catchment boarders; right Fig. 2) using  Shyft's interpolation algorithms. For temperature  Bayesian Kriging (Diggle and Ribeiro, 2007) is used. For precipitation, BC deposition rates, wind speed, and relative humidity  interpolation to the model cells is via inverse distance weighting

15 . A 5% increase in precipitation for every 100 m increase in altitude (Førland, 1979) is used for the precipitation interpolation.

To calibrate the model against observed discharge, we first run a  split-sample calibration (Klemes, 1986) using the first 3 years (1 September 2006 to 31 October 2009) of the study period as calibration period and the following 3 years (1 September 2009 to 31 October 2012) for model validation.

20 For parameter estimation, we use the BOBYQA algorithm for bound constrained optimization (Powell, 2009) . To asses the predictive efficiency of the model we use the Nash-Sutcliffe model efficiency (NSE).

$$NSE = 1 - \frac{\sum_{t=0}^{T}(Q_o^t - Q_s^t)^2}{\sum_{t=0}^{T}(Q_o^t - \overline{Q_o})^2} \tag{10}$$

25 where $Q_o^t$ and $Q_s^t$ are the observed and simulated discharge at time t, respectively, and $\overline{Q_0}$ is the mean observed discharge over the assessed period.  Model calibration is run with mid-estimates for all model parameters impacting the handling and effect of LAISI in the snowpack and aerosol depositions as simulated from REMO-HAM during model calibration. Those parameters and further model parameters, including the parameters estimated during calibration, are listed in the left column of Table ??. We investigate the uncertainty in the effect of LAISI on snow melt  by using the min and max effect

30 parameter estimates from Table ??, while holding constant all  other model parameters as estimated during calibration. To assess the gross effect of LAISI we compare the simulations to equivalent simulations in which ARF is not included.

**5 Results and Discussion**

In the following, we first present in Sect. 5.1 the role of model parameters and variables critical to the effect of LAISI on the development of a melting snowpack by using our new snow algorithm as a point model. We then present the results of the case study in Sect. 5.2, where we examine the significance of the LAISI radiative forcing for hydrological processes by simulating the impact of BC deposition on the snow melt and discharge generation in a snow dominated mountain catchment (Sect. 5.2).

**5.1 1-D sensitivity studies**

**5.1.1 Sensitivity to surface layer thickness**

To investigate the impact of the maximum surface layer thickness of the model, we run simulations with synthetic forcing and use maximal surface layer thicknesses of 4.0 mm SWE (max estimate, see Tabel **??**), 8.0 mm SWE (mid estimate), 16.0 mm SWE (min estimate). Additionally we include a single layer model with a vertically uniform distribution of BC in the analysis and for comparison a simulation with clean snow. Fig. 3a shows the effect of the different maximum surface layer thicknesses on the melting snowpack, with mid-estimates for further model parameters according to Table **??**. The maximum surface layer thickness strongly determines the surface BC mixing ratio over the melt season. During snow melt, surface BC increases up to a factor of circa 10, 20 and about 30 for maximum surface layer thicknesses of 16.0 mm SWE, 8.0 mm SWE, and 4.0 mm SWE, compared to the pre-melt season BC mixing ratio ($35 \text{ ng g}^{-1}$). Since the model input used in the sensitivity study during the melt period does exclude fresh snowfall and dry deposition, the increase in surface BC mixing ratio is due to melt amplification solely. The importance of BC accumulation in surface snow is discussed controversially in the literature. While several studies report a significant increase in surface BC mixing ratio during melt (Doherty et al., 2013; Sterle et al., 2013) of up to an order of magnitude (Sterle et al., 2013) and more (Xu et al., 2012), others report highly efficient scavenging with melt (Lazarcik et al., 2017). Over most of the melt period, our results show a factor increase between 5 and 15. Only at the end of the melt season, higher factor increases are reached. To this point of time, however, the snowpack is typically very thin and effects on discharge generation due to very high increase in surface BC should be small.

For the three 2-layer scenarios (green, purple and red curves in Fig. 3a), the resulting difference on the albedo and melt rate are small, even though the increase in surface layer mixing ratio during the melt season differs strongly among the scenarios

 . The relatively small differences in snowpack evolution among the two-layer models, despite the large differences in surface BC, result from the fact that for all two-layer models ~~the surface layer is. Aerosol closer to the surface absorb more effectively due to the higher radiative intensity near the surface, which explains the stronger stronger albedo decrease and melt rate increase with thinner surface layer: the mean radiative intensity diminishes with depth due to absorption in snow and LAISI and scattering, leading to a less effective absorption of LAISI in deeper snow. By what means the radiative intensity diminishes with depth depends, among other variables, on the optical grain size of the snow~~the surface layer thickness is much thinner than the penetration depth of shortwave radiation. For example, in clean snow with an optical grain size of 50 um, the radiative intensity diminishes to $\frac{1}{e}$ of its surface value (the so called penetration depth) in 25.5 mm SWE. For snow with an optical grain size of 1000 um, the penetration depth increases to 117 mm SWE (both results from Flanner et al., 2007, assuming a wavelength of 550 nm and a solar zenith angle of 60°). ~~For this reason, LAISI generally absorb more efficient in snow with a larger optical grain size. Thus, the differences in albedo and subsequent implications for melt of ARF scenarios compared to the no ARF scenario (black lines in Fig. 3a) are partly due to the increasing grain size during the melt period, and partly due to the accumulation of BC in the top layer. The relatively small differences in albedo, melt rate and snowpack development among the two-layer models (green, purple and red lines in top and bottom Fig. 3a) (despite the large differences in surface BC ; central Fig. 3a), result from the fact that for all two-layer models, the surface layer thickness is much thinner than the penetration depth. Thus, LAISIno ARFLAISILAISILAISILAISIby the LAISIlowering effect on the albedo in the beginning of the melt season(a bit less than five days; yellow line in bottom graph of Fig. 3a). Note that by simply adding a second layer, a doubling of the surface layer LAISI concentration occurs already when the accumulated melt equals the surface layer thickness, and thus the sensitivity~~

 compared to a clean snowpack than in the 2-layer scenarios (about five days).

The sensitivity study using different values for the maximum surface layer thickness provides three important results. First, when the properties of the  included LAISI are prone to melt amplification (scavenging  ratio below 1), a minimum of two layers is required to simulate the effect of efficient absorption resulting from LAISI located close to the snow surface. Second, the surface layer thickness only plays a minor role for the effect on the albedo,  as long as the assumption that the surface layer thickness is much smaller than the penetration depth of  shortwave radiation into the snowpack is justifiable. Third,  by varying the surface layer thickness in a reasonable range, we cover a large range of BC increase in surface snow during melt, yet the effect on albedo, snow melt and snowpack evolution is minimal. Observed LAISI concentrations often are sampled in the top few centimetres of the snowpack and compared to surface layer concentration of models (e.g., Flanner et al., 2007; Forsström et al., 2013) , even though the surface layer is not a measurable snow property. Our results show that the comparison of observed surface concentrations with simulations is critical due to the  large impact of the model surface layer thickness on the  surface  concentration - while the effect on key snowpack variables such as the snow albedo remain nearly unaffected. This highlights the need for including a surface layer variation in the uncertainty estimation of the comparison with snow sampled in the surface layer.

**5.1.2 Sensitivity to scavenging ratio of BC**

 Field measurements indicate that only a fraction of  BC is flushed out with the melt water and BC can accumulate near the snow surface (e.g., Xu et al., 2012; Doherty et al., 2013; Sterle et al., 2013; Doherty et al., 2016) . Our model is able to simulate this process by taking the scavenging ratio of BC during meltwater movement into account. In this section we explore the scavenging processes further

 , by investigating the impact of different BC scavenging ratios on the snowpack evolution. Fig. 3b. In the range of  investigated scavenging ratios, we find sensitivity of the BC surface mixing ratio, the albedo, and the subsequent snow melt to this parameter. When applying a melt scavenging factor typical for hydrophobic BC (green lines in graphs of Fig. 3b) there is little effect compared to the scenario without melt scavenging (purple lines; both show circa a factor 30 increase in surface BC concentration to the end of the melt season and only little differences in the development of albedo and snow melt). However, a distinction exists when using a scavenging ratio estimate for hydrophilic BC. In contrast to the no melt scavenging and hydrophobic

scenarios, surface BC does not increase as rapidly  during the melt period (red line, central graph of Fig. 3b) and in fact is completely flushed  when applying the max-estimate of hydrophilic scavenging (yellow line).

The changes in the scavenging ratio  lead to a considerable effect on the albedo and the snow melt (meltout delayed by circa 1 (green lines), 2.5 (red lines), and 7 days (yellow lines) for scavenging ratios of 0.03, 0.2, and 2.0, respectively, compared to no melt scavenging (purple lines in Fig. 3b)). Compared to the  no-ARF experiment (black lines), the presence of  BC still causes an earlier meltout of circa 8, 6.5, and 2 days for scavenging ratios of 0.03, 0.2, and 2.0, respectively, in our simulation. This implies a significant effect of BC on the albedo in all scenarios applied. Only when the melt scavenging is set to the upper limit (2.0; yellow lines in graphs of Fig. 3b), the surface concentration drops continuously during the melt period due to the highly efficient melt scavenging. As a consequence, the albedo converges against the albedo of the  no-ARF case, before it drops roughly one day earlier to a value of circa 0.2 due to the earlier exposure of the underling ground (solid yellow and black line in top graph of Fig. 3b). The slight increasing in difference in the melt rate between the  no-ARF and the upper  scavenging during the first 7 days  of melt are due to the increasing absorption efficiency of BC with increasing optical snow grain size  (e.g., Flanner et al., 2007). The following convergence (day 7 until 17 from melt onset) of both melt rates are due to the decreasing  BC concentration in the upper  scavenging scenario due to ongoing removal of  BC (compare the dashed yellow and black line in top graph of Fig. 3b). However, even though nearly all  BC is removed from the snow by the end of the melt period when using the upper scavenging ratio, the melt out still happens circa two days earlier compared to the  no-ARF experiment. This reveals that small amounts of BC in snow can impact the snowpack evolution over the whole melt period even under efficient scavenging.

**5.1.3  Sensitivity to BC species**

Hydrophilic BC absorbs stronger than hydrophobic BC under the same conditions due to an increased MAC compared to hydrophobic BC caused by the ageing of BC during atmospheric transport (Bond et al., 2006). On the other hand, as we previously explored, hydrophilic BC undergoes more efficient melt scavenging (Flanner et al., 2007), which impacts the snowpack evolution significantly. The column of graphs in Fig. 3c illustrates the net effect of these competing processes by applying the mid estimate of the scavenging ratio of hydrophobic BC (0.03) to both the hydrophobic BC (green curve) and the hydrophilic BC (purple curves) species. In this manner these curves show the isolated effect of the different absorption properties of the two species. We further apply the mid estimate for hydrophilic BC scavenging ratio (0.2) to hydrophilic BC (red curves) to quantify the gross effect. As in other cases, we include the  no-ARF scenario (black curves) to highlight the overall effect on the albedo and melt of the different scenarios.

The isolated effect of the stronger absorption of hydrophilic BC leads to an earlier meltout by circa two days compared to hydrophobic BC (purple and green curves in graphs of Fig. 3c). However, when applying the mid estimate of the scavenging ratio for hydrophilic BC (0.2),  the combined effects

 leads to a masking of the isolated effect of stronger absorption by hydrophilic BC (and vice versa). During the melt period,  snow albedo, melt rate and the snowpack SWE barely differ between the scenarios with the mid estimate scavenging  for hydrophobic and hydrophilic BC applied (red and green curves in top and bottom graphs of Fig. 3c). This reveals that both scenarios, hydrophobic BC with low scavenging

5  efficiency and hydrophilic BC with high scavenging efficiency, lead roughly to an earlier meltout by circa 6 days. We interpret this that a clear distinction between the both species might play a secondary role in the determination of the overall impact of BC on snow melt.

**5.1.4 Sensitivity to snowpack SWE at melt onset**

 In the following we explore the shortening of the melt period duration of snowpacks with constant BC mixing ratio at melt onset but different SWE relative to clean snowpacks with similar SWE. Results are shown in Fig. 4 for different scavenging ratios. Apart from  SWE

15 and scavenging ratio, all initial snowpack properties and the model input data are the same among the different scenarios, including an initial BC concentration of 35 ng g$^{-1}$. BC is distributed uniformly throughout the entire snowpack at melt onset. With respect to the range of snowpack SWE at

20  melt onset presented here, the  melt period shortening is stronger the smaller the scavenging ratio applied, and increasing with increasing SWE at melt onset. Results show a melt period reduction of up to 30% for the mid-estimate hydrophilic BC was applied for small

25  scavenging, end even higher when applying the mid-estimate hydrophobic BC scavenging.

With increasing SWE at melt onset, the increase in the  melt period shortening gets less pronounced  (dashed and dashed-dotted curves in Fig. 4), and differences between the melt scavenging scenarios become larger. When applying very efficient melt scavenging (dotted curve in Fig. 4), the effect on the  reduction is smallest over the range of SWE values shown, however, still

30 leading to a melt period shortening between 4-8%.

The results suggest that not only the BC concentration and distribution, the snow properties, and the radiative properties and hydrophobicity of the aerosol control how  BC in snow impacts the melt. The amount of snow accumulated

also plays an important role, with thicker snowpacks and similar LAISI mixing ratio at melt onset showing a stronger response to the LAISI induced processes.

**5.2 Case study: Impact of BC deposition on the hydrology of a south Norwegian catchment**

**5.2.1 Performance of the model**

 In the split-sample test, the model performs reasonably well during both calibration and validation, with NSEs of 0.86 during the calibration period (green line in Fig. 5a) and 0.82 during the validation period (red line in Fig. 5a). However, in  the  winter season (circa November until March) the model generally underestimates the discharge and peaks in the beginning of the  melt season are slightly underestimated. The scatter plot in Fig. 6  confirms the underestimation of low flow situations. For the case study analysis, we use model parameters  from a calibration over the  full period (1 September 2006 to 31 October 2012; Fig. 5b), which results in a NSE of 0.84. We use mid-estimates for all LAISI-relevant parameters. The optimized parameters are listed in Table **??**. Note that switching ARF off entirely (no BC deposition) leads to a slight decrease of the model quality (NSE of 0.83 over the whole period; not shown).

**5.2.2 Evolution of surface BC mixing ratio**

The evolution of surface albedo driven by BC deposition is distinct in the accumulation period vs. the melt period. During the snow accumulation period (circa until end of March), only slight differences in albedo are noticeable. The average annual snow albedo from January $1^{st}$ until March $22^{nd}$ is 0.871 for the no-ARF experiment (Fig. 7a), while during the same time period, min, mid, and max scenarios show relative albedo reductions of 0.003, 0.010, and 0.014, respectively from the no-ARF case. The differences in snow albedo during the accumulation season are mostly due to differences in deposition and in the maximum surface layer thickness of the snowpack, and lead to average surface layer concentrations of 12, 49, and 98 ng g$^{-1}$ (min, mid, and max estimates; Fig. 7b) at the beginning of the melt period.

With the start of the melt season, the difference in albedo is larger between model experiments. This has two reasons: (i) with increasing grain size during the melt season, the absorbing effect of BC gets more efficient due to deeper penetration of radiation into the snowpack leading to a stronger effect of the BC deposition on albedo (snow of larger grains has a larger

extinction coefficient and more effective forward scattering properties (Flanner et al., 2007) ). (ii) with the start of the melt season there is a widespread decrease of snow thickness, allowing BC to accumulate in the surface layer. This latter effect is strongly depended on the applied scavenging ratios, as we demonstrated in the 1-D sensitivity study (cf. Sect. 5.1). During the melt season, the mid-scenario spatially averaged surface BC mixing ratio increases from 49 ng g$^{-1}$ to about 250 ng g$^{-1}$ (factor 5 increase) at the end of the melt season (beginning of July). For the max-scenario, the increase is from roughly 100 ng g$^{-1}$ to over 2500 ng g$^{-1}$ (factor 25 increase), while the min-scenario leads to a decrease in BC surface mixing ratio. At the end of the melt season, the large differences in surface BC mixing ratio cause a relative decrease from the no-ARF case of about 0.03, 0.1 and over 0.3 for the min, mid, and max scenario, respectively.

For the min- and mid-scenario the model simulates an average annual surface BC mixing ratio of about 18 ng g$^{-1}$ and 71 ng g$^{-1}$, respectively. Forsström et al. (2013) found for mainland Scandinavia values of the same magnitude, with seasonal means for different measurement locations and time periods ranging from about 10 ng g$^{-1}$ to 80 ng g$^{-1}$. This places our results well within those presented in Forsström et al. (2013) . Our max-scenario yields 198 ng g$^{-1}$ which lies above average values expected from Forsström et al. (2013) . However, Flanner et al. (2007) evaluated the global impact of the radiative forcing of BC in snow using a model which was compared with globally distributed surface BC measurements. For south Norway, Flanner et al. (2007) predicted an annual mean surface BC concentration between 46 and 215 ng g$^{-1}$ for the year 1998. Including Flanner et al. (2007) 's results, our simulations reproduce a reasonable range of values.

Still, we recognize our max-scenario results in a strong increase in surface BC mixing ratios mostly due to low BC scavenging with melt (note the strong increase from end of March on in Fig. 7). This divergent evolution of surface BC mixing ratios in the min, mid, and max scenarios reveals uncertainty in the representation of the fate of BC in snow during melt. This uncertainty is also reflected in the literature. On the one hand, some studies report of high accumulation of BC in surface snow with implications for snow melt. Doherty et al. (2013) reported a factor 5 increase in surface BC mixing ratio under melt conditions, and in a subsequent study found increases of over an order of magnitude (Doherty et al., 2016) . These findings were similar to Xu et al. (2012) who also found post-depositional enrichment of BC in surface snow over an order of magnitude. On the other hand, Lazarcik et al. (2017) observe efficiently scavenged BC, leading to decreased surface mixing ratios. In fact, they report BC leaching from the snow more rapidly than the snow melt and summarize that that surface enrichment of BC is not linked to SWE decreases during melt. The large differences in the evolution of surface BC mixing ratio during melt in our study reflects the range of uncertainties shown in previous studies. However, while the surface BC mixing ratio evolution during melt for min- and mid-scenario is within reason, it appears our max-scenario results in an overestimate of melt amplification.

**5.2.3  BC induced radiative forcing**

The radiative forcing in snow (RFS) induced by the presence of BC is calculated from the average radiative forcing over snow bearing tiles only. The RFS represents the additional uptake of energy from solar radiation per area snow cover due to the presence of BC in the snow compared to clean snow with the same properties. Fig. 8a shows the daily mean RFS and demonstrates the increase effect of RFS during snow melt. Low RFS is observed during the snow accumulation period then

steadily increasing through spring snow melt, reaching values of approximately 8, 18, and 57 $Wm^{-2}$ for the min, mid, and max scenarios, respectively (see red solid line and shaded area in Fig. 8a). The strong increase in RFS during spring melt results from the combination of: (i) the decrease in snow albedo due to the increase in surface BC concentrations (e.g. melt amplification and the increasing optical grain size in melting snow as discussed in Sect. 5.2.2) and, (ii) the increasing daily solar irradiation due to a lower solar zenith angle and longer days.

However, most relevant for discharge generation (see Sect. 5.2.4), is the catchment-wide total daily energy uptake due to BC, calculated as the mean radiative forcing over all grid cells. As the snow cover fraction (SCF) in the catchment drops during spring (dotted line and yellow shaded area in Fig. 7 and 8), the effect of the RFS on the melt generation is limited by the increasing area of bare ground. The net effect is shown in Fig. 8b. The catchment mean daily energy uptake due to the presence of BC in snow shows a strong annual cycle and reaches a maximum of 1.3, 4.9, and 8.8 $Wm^{-2}$ (min, mid, and max scenario, respectively) around the beginning of May. Radiative forcing in mid winter is small due low surface BC mixing ratios and low solar irradiate. (Qian et al., 2011) also reports a similar strong annual cycle with values in the same range for BC radiative forcing over the Tibetan Plateau using a global climate model, but with higher values in winter time. Annual mean values are 0.284, 0.844, and 1.391 $Wm^{-2}$ for the min, mid, and max scenario. Averaged over entire Scandinavia (including Finland), Hienola et al. (2016) calculated lower values around 0.145 $Wm^{-2}$. However, Hienola et al. (2016) study includes large areas with shorter snow cover. Since the value is strongly depended on the snow cover evolution, higher values compared to Hienola et al. (2016) are expected due to the long lasting snow cover in our case study region.

**5.2.4   BC impact on catchment discharge and snow storage**

Fig. 9a shows the simulated daily discharge and catchment SWE  averaged over the 6 years simulation period for the mid (red lines); min and max estimates (bounds of the shaded areas); and the no-ARF scenario (black lines). The  differences in daily discharge and catchment SWE of the min, mid, and max scenarios to the  no-ARF scenario are shown in Fig. 9b. All simulations with ARF  show higher daily discharge from end of March until end of May and lower discharge from end of  May until mid August  relative to the no-ARF simulation. For the rest of the year, no effect on the discharge is noticeable. The  net impact of RFS results in a shift in the timing of discharge. Higher discharge early in the melt season  is observed, yet offset by lower discharge following May. The cumulative annual discharge remains nearly identical.

Min, mid, and max scenarios all show the change from higher  to lower discharge compared to the  no-ARF scenario approximately at the same  time (at the end of  May; see blue marker in Fig. 9b).  Therefore, we can quantify the absolute and relative effect of RFS on the

discharge during the two periods.  : the early melt season from circa March 22 until May 29 and  the late melt season from circa May 30 until August  an average percentage increase in daily discharge of 2.5 %, 9.9 % and 21.4 % for the min, mid, and max scenario  for the early melt season and a decrease in discharge  of -0.8 %, -3.1 %, and -6.7 % ~~, respectively. Maximum increase in daily discharge during the 6 years simulation period is 1.4 m$^3$ s$^{-1}$ (3.6 ), 5.6 m$^3$ s$^{-1}$ (17.3 ), and 11.9 m$^3$ s$^{-1}$ (42.7 ) for the min, mid and max estimates, respectively (not shown). The maximum decrease in daily discharge during the simulation is determined to -1.9 m$^3$ s$^{-1}$ (-8.1 ), -8.2 m$^3$ s$^{-1}$ (-11.4 ), -14.8 m$^3$ s$^{-1}$ (-20.9 ) for the min, mid and max estimates, respectively (not shown).~~ during the late melt season.

 The differences in discharge among the scenarios can be explained  by understanding the evolution of the snowpack.  In the all scenarios the catchment SWE (Fig. 9a reaches a peak reduction relative to the no-ARF scenario of -4.6 %, -13.4 % and -34.4 % at mid May. The average difference in catchment SWE of the min, mid, and max scenarios compared to the  no-ARF scenario during the entire melt season is  -1.5, -5.1, and -10.3 mm; or an average of 2.1 %, 7.4 %, and 15.1 %  (see Table ??). From mid May on, the differences in catchment SWE between  scenarios drop continuously, which is equivalent to a higher catchment averaged snow melt rate in the  no-ARF scenario compared to the ARF scenarios.

 The difference at the beginning of the melt season can be attributed to  RFS. However, from  mid May on we see a decrease in the differences in catchment SWE between the ARF and  no-ARF scenarios (Fig. 9b).  To understand this counter-intuitive result, we need to evaluate the impact of BC deposition at the catchment scale . The dynamics driven by the SCF of the catchment  is a limiting factor to the catchment averaged snow melt.

 Looking at Fig. 7a we see the development of  average snow albedo and the  SCF in the catchment. During the melt period, the catchment averaged albedo in all of the scenarios, decreases  (and RFS is continually increasing). Intuitively, one would expect more melting due to enhanced solar radiative

forcing. However,  the SCF decrease with increased melt due to ARF counteracts the RFS effect itself, due to the  reduction in area from which snow can actually melt

 . This is also indicated in Fig. 8b, where the additional energy uptake due to BC in snow peaks in the beginning of May. We can see the same result for the discharge: the increased discharge of the ARF scenarios  during the beginning of the melt season  may simply be attributed to RFS, whereas the decreased discharge later in the season  attributed to melt limitation caused by the  SCF retreat.

**5.2.5**

 Compared to observations, all simulations (ARF and no-ARF) tend to underestimate discharge during early melt season and overestimate discharge during late melt season (Fig. 9a).  However, the magnitude of over- and underestimation strongly differs between the scenarios. By including ARF the volume error is reduced in both the early melt season (by increasing melt), and in late melt season (by subsequently decreasing melt generation in the catchment due to reduced SCF). Expressed as seasonal mean volume error for early and late melt season, ~~49 and 98 ng g$^{-1}$ (Fig. 7b) at the beginning of the melt period. With the start of the melt season , the difference in albedo gets larger between the different model experiments. This has several combined reasons: (i) with increasing grain size during the melt season, the absorbing effect of BC gets more efficient due to deeper penetration of radiation into the snowpack (snow of larger grains has a larger extinction coefficient and more effective forward scattering properties (Flanner et al., 2007) ). This leads to a stronger effect of the pre-melt season BC concentrations on the albedo. (ii) With the start of the melt season and thus widespread retreat of the vertical snow extent, BC can accumulate in the surface layer. This effect is strongly depended on the applied scavenging ratios of hydrophilic and hydrophobic BC as we demonstrated in the 1-D sensitivity study in Sect. 5.1. The magnitude of the scavenging ratio determines if BC can accumulate in the surface layer and acts to decrease on the albedo (scavenging ratio below 1) or if BC is efficiently removed by melt water, leading (as isolated effect) to an increase of albedo (scavenging ratio above 1). The applied scavenging ratios of hydrophobic and hydrophilic BC in the mid (0.03~~ the difference to observed discharge is largest for the no-ARF scenario and smallest for max scenario. The max scenario reduces the volume

error by -75.1% during early melt season and 0.2, respectively) and max -89.9% during late melt season, relative to the no-ARF scenario (0.003 and 0.02, respectively) model experiments are below 1 and accumulation of BC in the surface layer results. For the mid scenario , the spatially averaged surface BC increases from a pre-melt season value of about 49 ng g$^{-1}$ to a surface BC concentration of circa 250 ng g$^{-1}$ (factor 5 increase) to the end of the melt season(beginning of July). For the max experiment,

5   the simulated surface BC concentration increases from roughly 100 ng g$^{-1}$ to over 2500 ng g$^{-1}$ (factor 25 increase). For the min scenario the scavenging ratio for hydrophilic BC is 2.0 - leading to a decrease in the surface concentration of hydrophilic BC. Even though the surface concentration of hydrophobic BC increases, the total surface concentration of BC decreases due to the higher - circa factor 20 - hydrophilic BC concentration to the beginning of the melt season compared to the surface concentration of hydrophobic BC (see lower boundary of the shaded area in Fig. 7b). (iii)A third reason for the enhanced

10  albedo is the strong increase in BC at the end of the melt season. The sub-grid snow variability plays an important role due to the fact that BC is predominately wet deposited. The mid and max scenario show a roughly linear increase of surface BC concentration on a log-scale see Table **??**). The min and mid scenarios also reduce the volume error. Thus, on average, an improvement in simulated discharge is achieved during the melt season . The tiles bearing little snow melt out more quickly than the tiles containing large snow accumulation. At the same time, tiles bearing large quantities of snow tend to also bear

15  large quantities of BC (in terms of total BC mass) due toby accounting for BC RFS.

**5.3   Uncertainties**

Both the literature and our analysis demonstrates numerous uncertainties and we urge further studies to address RFS-induced uncertainty. In our model study, uncertainties result principally from uncertainty of the mixing ratio of BC in the snowpack due to:

20  **i) - prescribed BC deposition**

   In the approach presented here, we use prescribed BC deposition mass fluxes. Even though this is common practice (e.g., Goldenson et al., 2012; Lee et al., 2013; Jiao et al., 2014) , it was showing by Doherty et al. (2014) that the dominantly wet-depositioned BC, which we chose in the model to follow the same redistribution as snow. Only dry deposition is assumed to deposit spatially homogeneous over the sub grid tiles. Late in the melt season, the snow albedo is predominantly computed

25  from tiles, that due to a high accumulation factor, were rich in snow after the snow accumulation period, and thus rich in BC mass. That leads to high accumulation in the top layer when combined with a scavenging ratio of below 1. This effect amplifies the catchment averaged surface BC concentration increase during the melt season in the mid and max estimate scenarios and contributes to the large differences in surface BC among the three scenarios to the end of the melt season. This large difference in surface BC between the different scenarios is then causing the wide spread in snow albedo to the end of the melt season,

30  lowering the average snow albedo in the catchment by about 0.03decoupling of aerosol deposition from the water mass flux of falling snow can lead to an overestimation of surface mixing ratios by a factor of 1.5-2.5. However, we would like to highlight an important difference between our approach and the one (Doherty et al., 2014) claim to be problematic: First, the high bias in surface snow BC mixing ratios described by (Doherty et al., 2014) refers to global climate model simulations with prescribed aerosol deposition rates (wet and dry), 0.1 and over 0.3 in the min, mid and max estimate scenarios to the end of the melt

season due to ARF. Qualitatively, we feel this represents reality well, in that if we think about snow patches in a catchment at the end of the season, they tend to be 'dirty', as the concentration of impurities increases while the water melts away.

The range of the catchment mean surface BC concentrations in the min, mid and max estimate becomes extremely high to the end of the melt period, ranging over more than 3 orders of magnitude between the min and max estimate of the simulations (see Fig. 7b). However, at the point of time when these extreme differences are reached, the SCF of the catchment of all scenarios is converging toward zero - making the concentrations to this point of time not representative for the development throughout the melt period. But where the input aerosol fields are interpolated in time from monthly means. Therefore, the episodic nature of aerosol deposition due to wet deposition is generally absent in the prescribed-aerosol fields. The coupling of the interpolated fields with highly variable meteorology (in particular precipitation) results in the high bias (Doherty et al., 2014) . In our case study, on the other hand, we use deposition fields originating from the regional aerosol climate model REMO-HAM, forced with ERA-Interim reanalysis data at the boundaries. REMO-HAM output is 3-hourly, which we re-sampled to daily means in order to have consistency between the deposition fields and the observed daily precipitation used as input data in the extreme diverging results highlight the high uncertainty that comes with simulation of the fate of LAISI in the snowpack and the ARF they are causing.

A significant challenge when evaluating these results is the severe lack of observations - not only in hydrological simulations. The daily timestep allows us to preserve the episodic nature of aerosol deposition. Moreover, the catchment used herein as case study, but in general when simulating the impact of LAISI on the snowpack over a melt season - especially when the approach involves the determination of the LAISI concentrations in the snowpack from aerosol depositionrates. In the study on the global impact of the radiative forcing of BC in snow, Flanner et al. (2007) compare the model results with various measurement of surface BC , representing many cryospheric regions of the globe - with overall good agreement with observations. For south Norway, Flanner et al. (2007) predict central estimate annual mean surface BC concentrations between 46 and 215 ng $g^{-1}$ for the year 1998. Our simulations show concentrations in the range of Flanner et al. (2007) 's results (71 ng $g^{-1}$ for the mid estimate average annual surface BC concentration over the 6 years period, and 18 daily BC wet deposition rates should not be biased due to major inaccuracies in precipitation as REMO-HAM has been shown to reproduce the Scandinavian precipitation realistically (Pietikäinen et al., 2012) . The high bias occurring when using interpolated monthly averages as input should therefore be minimized.

Additionally, and 198 ng $g^{-1}$ for the min and max estimate, respectively) . Our results further agree with the range of surface BC observations in mainland Norway presented in Forsström et al. (2013) .

Our model further suggests that melt amplification can have severe implications on the impact of LAISI on both, the snowpack evolution and the discharge regime of a catchment, which means that the seasonal cycle of surface BC concentration is of great importance. Especially for the impact on significantly, (Doherty et al., 2014) (and the critiques therein) address an objective with consideration to climate impacts. Our analysis is focused on the impact to the hydrological cycle, the fate of the LAISI in melting snow is essential – which leads to great importance of surface BC concentrations during spring. The increase of the mid estimate surface BC during the melt season agree with observations from Doherty et al. (2013) , who measured a roughly 5 times increase in surface BC of a melting snow. The experiments conducted by Conway et al. (1996) investigating

artificially added BC and on melting snow show similar results. Forsström et al. (2013) associates large spikes in observed surface BC with with snow melt , which supports the course of the mean surface BC concentration in the catchment resulting from the mid estimate simulation.

However, the surface BC concentrations during spring melt are also the most uncertain (see differing course of BC surface concentration of min, mid and max estimate in . Our simulations suggest that BC RFS is mostly important during spring time, where surface BC mixing ratio are predominantly controlled by melt processes, and not by deposition processes (as shown in Fig. 3 and Fig. 7b): the parameters quantifying the effect of melt amplification are are based on the results of a sole field experiment campaign only (namely the experiments conducted by Conway et al. (1996) ). This relatively weak basis for the mid estimates of the model parameters, combined with lacking observational data of surface BC in our study region during the melt season leads then to the high uncertainties our model results are showing.

**5.3.1  BC induced radiative forcing in snow and catchment**

**ii) - LAISI other than BC**

Fig. 8a shows the daily mean radiative forcing in the catchment snow induced by the presence of BC in snow averaged over snow bearing tiles only (herein after referred to as RFS, radiative forcing in snow). The RFS represents the additional uptake of energy from solar radiation due to the presence of BC in the snow compared to clean snow with By including only BC deposition in our simulation, we potentially underestimate the additional effect of further LAISI species such as mineral dust (Di Mauro et al., 2015; Painter et al., 2010) , mixing of the same properties. Our simulations suggest that the RFS underlies a strong annual cycle with low values during the snow accumulation period and steadily increasing values during spring snow melt, reaching values of approximately 8, 18 and 57 Wm$^{-2}$ for the min, mid, and max effect estimates, respectively, to the end of the spring melt season (see red solid line and shaded area in Fig. 8a). The strong increase in RFS during spring melt results from two combined processes: (i) the decrease in snow albedo due to the catchment wide increase in surface BC concentrations (melt amplification) and the increasing optical grain size in melting snow as discussed in Sect. 5.2.2 and (ii) the increasing daily solar irradiation due to a lower solar zenith angle and longer days. The RFS averaged over the 6 years simulation period is 0.50, 1.48 and 2.43 Wm$^{-2}$ for the min, mid, max scenarios, respectively. However, for the snow with soil from the underlying ground, or local sources (Wang et al., 2013) and biological processes (Lutz et al., 2016) . Neglecting additional RFS from LAISI other than BC is likely to result in an underestimation of the overall effect of LAISI on snow melt and discharge generation. However, this implies that our approach gives a conservative estimate of the effect on the discharge generation, a more relevant variable is the SCF normalized daily radiative forcing in snow. As the SCF drops, the effect of the RFS on the melt generation in the catchment gets limited by the increasing area of bare ground. The net effect is shown in Fig. 8b, where the radiative forcing is normalized with the SCF. The results can be seen as a measure for the catchment wide additional energy uptake due to the presence of BC in snow, which on average reaches a maximum of 1.3, 4.9 and 8.8 Wm$^{-2}$ (min, mid and max scenario, respectively) in around the beginning of May of LAISI, with BC being a proxy for the overall effect of LAISI on snow melt and discharge generation on the catchment scale.

**6  Conclusions**

Herein we presented a newly developed snow algorithm for application in hydrologic models that allows a new class of  variables: the deposition rates of light absorbing aerosols. By coupling a radiative transfer model for snow to an energy balance based snowpack model, we are providing a tool that can be used to determine the effect of various species of LAISI (herein shown for BC) on the hydrologic cycle  at the catchment scale. From a 1-D model study, presented in Sect. 5.1, we conclude that:

  i -  the implementation of at least two layers (a thin surface layer and a bottom layer) is of outstanding importance to capture the potential effect of melt amplification on the near surface LAISI evolution. The maximum surface layer thickness (in SWE)  has a rather little effect on the snow albedo and melt rate as long as  it is sufficiently small (smaller than the penetration depth of shortwave radiation). However, the evolution of the LAISI surface  mixing ratio is highly sensitive to the  maximum surface layer thickness. For this reason, we suggest to include a surface layer thickness variation in model studies when comparing simulated to observed LAISI mixing ratios sampled in the snow surface.

  ii -  The determination on how LAISI is washed out of the snowpack with melt water has great effect on the evolution of LAISI concentration near the surface, snow albedo and melt rate. Due to rare observations of this effect the uncertainties are high and our findings show the need for more detailed understanding of the processes involved due to the high importance for the overall effect of LAISI in the snowpack.

  iii -  Snow rich  areas are likely to be more effected by LAISI than  areas with medium snow accumulation due to a nonlinear relationship between melt season duration and SWE at melt onset.

To prove the significance of the  radiative forcing from BC for the hydrologic cycle  at the catchment scale we demonstrated the effect of BC deposition and the subsequent implications for snow melt and discharge generation due to impacts on the snow albedo on a remote mountain catchment. ~~Even though our model approach is conservative due the lacking implementation of the effect of LAISI on the grain size growth and due to the choice of a remote northern catchment of only medium snow accumulation (compared to other Norwegian mountain catchments), we could show that the effect on the discharge generation is significant, even in low deposition regions like Norway, leading to a shift inannual water balance~~ discharge regime of a catchment, which means that the seasonal cycle of surface BC mixing ratio is of great importance. However, large uncertainties are connected with the representation of surface enrichment of BC. Especially for the impact on the hydrological cycle, the fate of the BC in melting snow is essential.

Including radiative forcing from BC in the simulations leads to a reduction in volume error during the early and late melt season in our simulations. We conclude from  our study that hydrological modelling can potentially be improved by including the effect of LAISI, especially when the model approach implicates a physically based representation of the snowpack in general and the snow albedo in particular. However, more research in the area of catchment scale impact of LAISI is needed to support this. The model tool presented in this study allows to target this in future applications.

*Acknowledgements.* This work was conducted within the Norwegian Research Council's INDNOR program under the Hydrologic sensitivity to Cryosphere-Aerosol interaction in Mountain Processes (HyCAMP) project (NFR no. 222195). We thank the Mitigation of Arctic warming by controlling European black carbon emissions (MACEB) project for their help concerning the REMO-HAM simulations. Furthermore, we thank the International Institute for Applied System Analysis (IIASA), especially Kaarle Kupiainen and Zbigniew Klimont, for providing the the emissions data. Sigbjorn Helset and Statkraft AS, in general, have been vital resources in the development of the algorithm and, in particular, the implmentation into Shyft. The ECHAM-HAMMOZ model is developed by a consortium composed of ETH Zurich, Max Planck Institut für Meteorologie, Forschungszentrum Jülich, University of Oxford, and the Finnish Meteorological Institute and managed by the Center for Climate Systems Modeling (C2SM) at ETH Zurich. We also would like to thank Sigbjørn Helseth from Statkraft for the support during the implementation of our algorithm into the Shyft framework.

[revised manuscript text omitted]

---

## Referee Report (RR1)

**Review of "Modelling hydrological impacts of light absorbing aerosol deposition on snow at the catchment scale"**

**General comments**

I have found this manuscript much improved from the previous version. The structure is far clearer and more precise. Additionally, I believe that the reviewer addressed many of the comments that reviewers addressed in the first version.

However, two subjects that were addressed in the previous review should be further explored. Although this aspect has been improved since the original manuscript, I believe that the authors should engage more with observed processes within the snowpack. First, in the first review the authors were asked to put the research in the context of field studies that have been done on this topic. This was recommended as a means to show the robustness or ability of their model. Although this work and the associated processes have been cited and mentioned, the further comparison and description with model results should be done. Specific instances are described below.

Second, I understand the constraints of this project, and realize that the focus is on black carbon, however, other LAISI are very important and likely have a larger role than black carbon in reducing albedo. For the sensitivity studies this should not be a large problem. However, for the catchment modeling I think this is a bigger issue. I believe that the framework presented here could be quite a useful tool to implement LAISI in to hydrological modeling, so it seems important to explain its ability in the best possible way. To me it seems that since these sources are not accounted, the efficiency of the model when compared to real discharge data might not be accurate. Aside from dramatically changing the paper, I am not sure the best way to improve this. However, further commentary on this or work by the authors to address this subject would improve the paper and make the work more relevant and usable in other applications.

**Scientific Comments**

- Pg 3, Line 19. What is deposition field?

- Pg 4, Lines 1-15. This comment was made in the previous review. I think some citations need to be added here. Have other researchers published with "Statkraft"? What are "methods-stacks"? I think this paragraph can be improved.

- Pg 4, Line 19. add 2009 (if this is the correct year of the paper) following Kirchner's.

- Pg 7, Line 3. Describe "surface layer thickness".

- Pg 7, Line 7. Describe "melt amplification"

- Pg 12, Lines 10-35. I think the are interesting results and the paper would be strengthened if these were compared to real observed black carbon trends. For instance comments in line 23-25 align with observations made by Sterle et al., 2013 and Delaney et al., 2015. Especially given the lack of mineral dust and other LAISI in the model, I believe the confidence in the model would increase if such processes and results would related to observed data and work.

- Pg 13. Once again, the authors should engage with the body of research regarding field measurements. How do the scavenging ratios used in the sensitivity study compare to those observed?

- Pg 14. Section 5.1.4. This section I found confusing and am not sure of its intent and actual meaning It seems strange to state that thicker snowpacks have more LAISI. I assume this is due to your prescribed initial concentration and scavenging assumptions. However, black carbon in the snowpack is a manifestation of atmospheric conditions and scavenging ratio, regardless of snowpack quantity. Of course thicker snowpack will have a lower concentration. Additionally, BC in the snowpack is not steady through out the profile(Sterle et al. 2013, Xu et al. 2012, Delaney et al. 2015) and can often occur in events as dust on snow work in Colorado by Painter, as well as by Delaney et al. 2015, Kaspari et al. 2015 and Hadley et al. 2007 among others show. As a result I believe that this assessment should be refined.

- Pg 15, Section 5.2.2. There are many places where these model results could be compared more with field measurements for the literature. Additionally, maybe make some comments about how well these concentrations agree with literature values. To me they seem on the upper end of snowpacks outside of Asia, but some commentary would be good.

- Pg 19, Section 5.3. As mentioned above, the non-BC LAISI species in the model is a notable short-coming. Is it possible to determine what the main source of LAISI is in the catchment?

- Pg 19, Line 28-19. I cannot easily see where this point is explored in the text. See my comment regarding Section 5.1.4.

- Pg. 20 Line 11. "model tool" is a bit funny. Maybe "tool" or "model" or "modeling tool."

---

## Author Response (AR2)

**General comments**

We are thankful for the thoughtful comments received during this discussion. The comments have allowed us to improve our analysis and provided an important perspective in framing our discussion. Individual comments are addressed in further detail directly below or in the revised text.

**I have found this manuscript much improved from the previous version. The structure is far clearer and more precise. Additionally, I believe that the reviewer addressed many of the comments that reviewers addressed in the first version.**
Thank you for recognizing our efforts. We do feel strongly that the manuscript has improved with the help of the reviewers.

**However, two subjects that were addressed in the previous review should be further explored. Although this aspect has been improved since the original manuscript, I believe that the authors should engage more with observed processes within the snowpack.**

**First, in the first review the authors were asked to put the research in the context of field studies that have been done on this topic. This was recommended as a means to show the robustness or ability of their model. Although this work and the associated processes have been cited and mentioned, the further comparison and description with model results should be done. Specific instances are described below.**
We have put efforts in broadening our discussion and give a more detailed comparison with model studies. This accounts for the sensitivity study, where we lay focus on relating the investigated processes more accurately to observational studies, as well as for the case study. In the case study, we aim to discuss our results in a broader context.

**Second, I understand the constraints of this project, and realize that the focus is on black carbon, however, other LAISI are very important and likely have a larger role than black carbon in reducing albedo. For the sensitivity studies this should not be a large problem. However, for the catchment modeling I think this is a bigger issue. I believe that the framework presented here could be quite a useful tool to implement LAISI in to hydrological modeling, so it seems important to explain its ability in the best possible way. To me it seems that since these sources are not accounted, the efficiency of the model when compared to real discharge data might not be accurate. Aside from dramatically changing the paper, I am not sure the best way to improve this. However, further commentary on this or work by the authors to address this subject would improve the paper and make the work more relevant and usable in other applications.**
Thank you for recognizing our contribution. We feel the herein presented approach has a large potential in hydrologic applications in many regions of the world.
The study was initiated from a body of work (including transport modeling) that focused on BC deposition. Deposition data for other species is not available presently, and requires a significant effort.
We acknowledge that the focus on only black carbon is a potential shortcoming with respect to the overall effect of LAISI on snow melt, and the predicted impact on the hydrology may be underestimated. On the other hand, we show that BC alone has the potential to significantly impact discharge generation in a range of snow surface BC values that is reasonable. Further LAISI would increase this effect to a certain degree. For this reason we feel that our approach can be seen as a conservative estimate and demonstrates the need for further research to better quantify the total effect of LAISI. It is our intention to provide this algorithm as a tool for such quantification - which may be addressed in future studies. As to the concern that "the efficiency of the model when compared to real discharge data might not be accurate": The efficiency itself is an estimate on how "accurate" our simulation predict observed discharge. The high NSE values

reflect that our simulations are able to simulate discharge quite well, while including BC leads to improved simulations (improved NSE). We have now add further discussion (see Sect. 5.3. ) regarding the importance of the implementation of other LAISI species and the potential to increase the model efficiency. We agree this is an important point, and appreciate the reviewers insistence to include an improved discussion. We hope that it will be picked up in further studies.

Scientific Comments

1. **Pg 3, Line 19. What is deposition field?**
   By deposition field we mean deposition input time series then then are interpolated to the model domain. We have clarified this in the text:
   *Before:* "Aside from enabling the user to optionally apply a deposition field, the algorithm depends on standard atmospheric input variables (precipitation, temperature,  short wave radiation, wind speed, and relative humidity)"
   *Replaced with:* "Aside from enabling the user to optionally apply deposition mass fluxes as model input, the algorithm depends on standard atmospheric input variables (precipitation, temperature,  short wave radiation, wind speed, and relative humidity)."

2. **Pg 4, Lines 1-15. This comment was made in the previous review. I think some citations need to be added here. Have other researchers published with "Statkraft"?**
   So far, students have used the model and describe model specifics in their thesis. We added those as references. Also, a contribution from the AGU 2016 Fall meeting describing the model has been added. Furthermore, we are working on a manuscript describing the model in detail and aim to submit the manuscript within the next two months to *Geoscientific Model Development* (GMD). We hope to be able to add the resulting reference (e.g. the discussion paper) to the here presented paper by the time it is accepted.

3. **What are "methods-stacks"? I think this paragraph can be improved.**
   We have rewritten and shortened the paragraph. We avoid the expression "method-stacks" since this would raise the need of a detailed explanation of Shyft's paradigms, as pointed out by the reviewer. Since this is not the focus of the paper, we feel that shortening this paragraph in combination with the added references leads to an improvement of readability without sacrificing important information.

4. **Pg 4, Line 19. add 2009 (if this is the correct year of the paper) following Kirchner's.**
   We have corrected this.

5. **Pg 7, Line 3. Describe "surface layer thickness".**
   We have rephrased the sentence prior to the statement referenced herein in order to describe the "surface layer thickness":
   *Before:* "To represent the evolution of LAISI mixing ration near the surface, we treat LAISI in two layers in our model: (I) a surface layer with a time invariant maximum depth (in mm SWE), where the concentration …"
   *Replaced with*: "… : (i) a surface layer with a time invariant maximum thickness (further called maximum surface layer thickness). In the surface layer, the concentration …"

Furthermore, we sometimes referred to the "surface layer thickness" with "depth of the surface layer" in this paragraph. We changed "depth" to "thickness" in order to be consistent and clear.

6. **Pg 7, Line 7. Describe "melt amplification"**
Melt amplification is described in the paragraph prior to the one commented on herein (Pg 6, Line 24-26):
"As snow melts LAISI can remain near the surface due to inefficient melt scavenging, which leads to an increase in the near surface concentration of LAISI and thus a further decrease in the snow albedo; the so called melt amplification."

7. **Pg 12, Lines 10-35. I think there are interesting results and the paper would be strengthened if these were compared to real observed black carbon trends. For instance comments in line 23-25 align with observations made by Sterle et al., 2013 and Delaney et al., 2015. Especially given the lack of mineral dust and other LAISI in the model, I believe the confidence in the model would increase if such processes and results would related to observed data and work.**
We have partly restructured the paragraph and put effort into comparing the simulated BC evolution to observed trends. We also included the here suggested publications in the discussion.

8. **Pg 13. Once again, the authors should engage with the body of research regarding field measurements. How do the scavenging ratios used in the sensitivity study compare to those observed?**
We have included a description on how the used scavenging ratios relate to field observations. We also extended the discussion in the context of other publications about this topic.

9. **Pg 14. Section 5.1.4. This section I found confusing and am not sure of its intent and actual meaning It seems strange to state that thicker snowpacks have more LAISI. I assume this is due to your prescribed initial concentration and scavenging assumptions. However, black carbon in the snowpack is a manifestation of atmospheric conditions and scavenging ratio, regardless of snowpack quantity. Of course thicker snowpack will have a lower concentration. Additionally, BC in the snowpack is not steady through out the profile(Sterle et al. 2013, Xu et al. 2012, Delaney et al. 2015) and can often occur in events as dust on snow work in Colorado by Painter, as well as by Delaney et al. 2015, Kaspari et al. 2015 and Hadley et al. 2007 among others show. As a result I believe that this assessment should be refined.**
In the first version of the manuscript we conducted the analysis differently, by leaving the total mass of BC in the snowpack constant (which results in thicker snowpacks having lower concentrations, as suggested here). However, reviewer 1 suggested to "increase BC deposition in proportion to the increase in SWE so one can see to what degree having a equally-polluted but deeper snowpack" and we adapted our analysis accordingly. We conclude from this that both our analyses has some shortcomings, but neither approach demonstrates a strong sensitivity in relation to other parameters. We still believe in the relevance of both analyses, however, since the subject of this sub-section is not of great importance for the publication, we have decided to remove this paragraph from the paper. We hope it also helps to further shorten the manuscript, which was a recommendation in the initial revisions.

10. **Pg 15, Section 5.2.2. There are many places where these model results could be compared more with field measurements for the literature. Additionally, maybe make some comments about how well these concentrations agree with literature values. To me they seem on the upper end of snowpacks outside of Asia, but some commentary would be good.**
We have restructured this Section, included further literature comparisons of the results, and broadened the discussion.

11. **Pg 19, Section 5.3. As mentioned above, the non-BC LAISI species in the model is a notable short-coming. Is it possible to determine what the main source of LAISI is in the catchment?**
We have rewritten and broadened the discussion. We believe in the results of Forstrom 2013 that demonstrate dust deposition may not be significant, but also acknowledge a fair degree of uncertainty due to the limited extent of the observations.

12. **Pg 19, Line 28-19. I cannot easily see where this point is explored in the text. See my comment regarding Section 5.1.4.**
We have removed this conclusion (compare with our response to point 9.)

13. **Pg. 20 Line 11. "model tool" is a bit funny. Maybe "tool" or "model" or "modeling tool."**
We have modified this sentence.

[revised manuscript text omitted]
15   In this study we address this deficiency by introducing a rainfall-runoff model with a newly developed snow algorithm that allows for a new class of model input variables: the deposition mass flux of different species of light absorbing aerosols. The model integrates snowpack dynamics forced by LAISI and allows for analysis at the catchment scale. The algorithm uses a radiative transfer model for snow to account dynamically for the impact of LAISI on the snow albedo and the subsequent impacts on the snow melt and discharge generation. Aside from enabling the user to optionally apply deposition mass
20   fluxes as model input, the algorithm depends on standard atmospheric input variables (precipitation, temperature, short wave radiation, wind speed, and relative humidity). To enable a critical evaluation of the newly developed snowpack algorithm, we conduct two independent analyses: i) a 1-D sensitivity study of critical model parameters, and ii) a catchment scale analysis of the impact of LAISI. In both analysis we use BC in snow from wet and dry deposition as a proxy for the impact of LAISI.

We first present an overview over the hydrological model used in this study and the newly developed snow algorithm to treat
25   LAISI in the snowpack in Sect. 2. A description of the catchment used for our study and the input data sets is given in Sect. 3. Sect. 4 describes the 1-D model experiments and the model settings and calibration process in the case study. Lastly our results are presented together with the discussion first for the model experiments, followed by the case study within Sect. 5.

**2    Modelling framework and the snowpack algorithm**

In the following section we provide descriptions of the hydrologic model (Sect. 2.1) and the formulation of a novel snowpack
30   module used for the analyses (Sect. 2.2).

**2.1 Hydrologic Model Framework**

For the analysis, we use Statkraft's hydrologic forecasting toolbox (Shyft; https://github.com/statkraft/shyft), a model framework developed for hydropower forecasting ~~. The concept of Shyft follows the idea that a hydrological model can be expressed as a sequence of well known routines, each describing a certain aspect of the represented hydrological processes. Which processes are represented depend on the purpose of the model and the requirements of the user. The sequence of routines , the so called "methods-stack", is then run on a cell by cell basis, where the cell loosely represents an area of similar time-invariant geographical data (e.g. topographic properties or land type) with no specific restriction to cell geometry or area. The Shyft framework allows for the paradigm of distributed,The methods-stack used herein consists of~~

[revised manuscript text omitted]
 distinguishing between the type of deposition mechanism (hydrophilic BC predominantly from wet deposition, hydrophobic BC for dry deposition). By following Flanner et al. (2007), we set $k_{phob}$ to 0.03 and $k_{phil}$ to 0.2, and account for the large uncertainty by using an order of magnitude variation on $k_{phob}$ and $k_{phil}$. Like Flanner et al. (2007), we treat aged, hydrophilic BC as sulphate coated to account for the net increase in the mass absorption cross section (MAC) by 1.5 at $\lambda$=550 nm compared to hydrophobic BC caused by the ageing of BC (reducing effect on MAC) and particle coating from condensation of weakly absorbing compounds (enhancing effect on MAC) suggested by Bond et al. (2006). As a consequence, hydrophilic BC absorbs stronger than hydrophobic BC under the same conditions. On the other hand, hydrophilic BC undergoes a more efficient melt scavenging. The competing mechanisms are subjects of the 1-D sensitivity study in Sect. 5.1.3.

**2.2.2 Sub-grid variability in snow depth and snow cover**

The representation of sub-grid snow variability can play a key role in modelling the hydrology of areas with a seasonal snow-pack (e.g., Hartmann et al., 1999). Several approaches exist to capture the sub-grid snow covered fraction (SCF) and distribution of SWE. Statistical approaches often use so called snow depletion curves to describe a relationship between a prognostic snow variable (e.g SWE, accumulated melt depth) and regional observations of SCF, (e.g., Liston, 2004; Luce and Tarboton, 2004; Kolberg and Gottschalk, 2010). However, such approaches do not allow for explicit treatment of snow layers, which is required when simulating the mixing ratios of LAISI. In our model, we follow (Aas et al., 2017) by assuming that the sub-grid spatial distribution of each single event of solid precipitation follows a certain probability distribution function. From this distribution we calculate multiplication factors, which then are used to assign the snowfall of a model grid cell to a number of sub-grid computational elements, the so called tiles (Aas et al., 2017). The snow algorithm described herein is executed for each of the tiles separately. This implies that variables related to the snow state, such as SWE, liquid water content, impurity content, and snow albedo differ among the tiles. This also allows to simulate the sub-grid variability in impurity content. To calculate the multiplication factors, we assume that the sub-grid redistributed snow follows a gamma distribution (see e.g., Kolberg and Gottschalk, 2010; Gisnås et al., 2016), determined by the coefficient of variation (CV). CV values were derived based on work done by Gisnås et al. (2016), who used Winstral and Marks (2002)'s terrain-based parametrization to model snow redistribution in Norway by accounting for wind effects during the snow accumulation period over a digital elevation model with 10 m resolution. Gisnås et al. (2016) calibrated the redistribution model with snow depth data from Airborne Laser Scanning (ALS) over the Hardangervidda mountain plateau (see Melvold and Skaugen (2013)) and evaluated with snow depth data from ground penetrating radar observations at Finse, both located in Southern Norway. The detailed scheme is described in Gisnås et al. (2016). In the case study presented in Sect. 5.2, we use the CV values from Gisnås et al. (2016) to derive a linear relationship between the model grid cell's elevation and the corresponding CV value by simple linear regression (see Fig.1a), which results in a $R^2$-value of 0.71 and a p-value of smaller than 2.0e-5 for the study area. The linear relationship is only applied to grid cells with an areal forest cover fraction of lower than or equal to 0.5. For grid cells with a forest cover fraction of higher than 0.5, a constant snow CV value of 0.17 is used, following the findings of Liston (2004) for high latitude, mountainous forest. Examples of multiplication factors for forested grid cells and forest free grid cells for a different CV values are shown in Fig. 1b.

**3 Site description, meteorologic model input and atmospheric deposition data**

We selected the unregulated upper Atna catchment for our analysis. This catchment is located in a high elevation region of southern Norway (left Fig. 2). The watershed covers an area of 463 km$^2$ and ranges in elevation from 700 masl at the outlet at lake Atnsjoen to over 2000 masl in the Rondane mountains in the western part of the watershed (right Fig. 2), with approximately 90 % of the area above the forest limit. The average annual precipitation in the watershed during the study period is approximately 655 mm, where most precipitation falls as rain in summer. The mean annual discharge is approximately

[revised manuscript text omitted]

In the snow algorithm used in this study, dry deposition and sedimentation are treated the same way. Herein, dry deposition refers to the sum of REMO-HAM dry deposition and sedimentation.

**4    Model experiments and calibration**

10    Our analysis is in two parts in Sect. 5. First we present a 1-D sensitivity study investigating the impact of parameters and variables specific to the algorithm determining the effect of LAISI (Sect. 5.1). We then demonstrate the significance of BC in snow radiative forcing on the catchment scale in a case study by simulating the impact of wet and dry deposition of BC in a remote south Norwegian catchment (Sect. 5.2).

We assume uncertainties of the LAISI radiative forcing to originate mainly from the model representation of surface layer

15    thickness, melt scavenging of BC, and uncertainties in the deposition input data. To account for the uncertainties, we declare minimum (min), central (mid), and maximum (max) effect estimates to each of the critical parameters, outlined together with further model parameters in Table 2. The min, mid, and max estimates are both subjects of analysis in the sensitivity study (further described in Sect. 4.1) and used in the case study to give an uncertainty estimate of the LAISI effect on the hydrologic variables (further described in Sect. 4.2). We investigate the impact of BC impurities on the response variables by comparing

20    the results from Aerosol Radiative Forcing model experiments ("ARF" scenarios) to simulations in which all BC deposition rates are set to zero ("no-ARF" scenario).

**4.1    1-D sensitivity study experiments**

For the 1-D sensitivity study presented in Sect. 5.1, we use synthetic input data to study the evolution of snowpacks under constant melting conditions in order to identify the impact of different model settings: the impact of (i) the maximum surface

25    layer thickness, (ii) the scavenging ratio sole, and (iii) the impact of the scavenging ratio with respect to the BC species.  We run the model with model parameters as outlined in Table 2 if not otherwise specified.

The model input applied for melting is based on the average meteorological conditions during the melt season from mid March until mid July of the Atnsjoen catchment. In our sensitivity experimants, all snowpacks have 250 mm SWE of snow

30    with a mixing ratio of 35 ng g$^{-1}$ in both surface and bottom layer at melt onset. These values are representative of the upper 50% of tiles at winter snow maximum in the Atnsjoen catchment during the study period of the case study. During the melt

period, we exclude fresh snowfall and dry deposition, in order to isolate the effect of the tested model parameters on the snowpack evolution under melt conditions. This might lead to an underestimation of total BC mass in the snow column.

**4.2 Case study model setup and calibration**

We investigate the impact of BC aerosol deposition on the catchment hydrology of a Norwegian catchment over a study period of 6 years, from September 2006 to September 2012. The station based input data described above is interpolated to the simulation grid cells (1x1 km$^2$ and accordingly smaller cells at the catchment boarders; right Fig. 2) using Shyft's interpolation algorithms. For temperature Bayesian Kriging (Diggle and Ribeiro, 2007) is used. For precipitation, BC deposition rates, wind speed, and relative humidity interpolation to the model grid cells is via inverse distance weighting. A 5% increase in precipitation for every 100 m increase in altitude (Førland, 1979) is used for the precipitation interpolation.

To calibrate the model against observed discharge, we first run a split-sample calibration (Klemes, 1986) using the first 3 years (1 September 2006 to 31 October 2009) of the study period as calibration period and the following 3 years (1 September 2009 to 31 October 2012) for model validation. For parameter estimation, we use the BOBYQA algorithm for bound constrained optimization (Powell, 2009). To asses the predictive efficiency of the model we use the Nash-Sutcliffe model efficiency (NSE).

$$NSE = 1 - \frac{\sum_{t=0}^{T}(Q_o^t - Q_s^t)^2}{\sum_{t=0}^{T}(Q_o^t - \overline{Q_o})^2} \qquad (10)$$

where $Q_o^t$ and $Q_s^t$ are the observed and simulated discharge at time t, respectively, and $\overline{Q_0}$ is the mean observed discharge over the assessed period. Model calibration is run with mid-estimates for all model parameters impacting the handling and effect of LAISI in the snowpack and aerosol depositions as simulated from REMO-HAM during model calibration. Those parameters and further model parameters, including the parameters estimated during calibration, are listed in the left column of Table 2. We investigate the uncertainty in the effect of LAISI on snow melt by using the min and max effect parameter estimates from Table 2, while holding constant all other model parameters as estimated during calibration. To assess the gross effect of LAISI we compare the simulations to equivalent simulations in which ARF is not included.

**5 Results and Discussion**

In the following, we first present in Sect. 5.1 the role of model parameters and variables critical to the effect of LAISI on the development of a melting snowpack by using our new snow algorithm as a point model. We then present the results of the case study in Sect. 5.2, where we examine the significance of the LAISI radiative forcing for hydrological processes by simulating the impact of BC deposition on the snow melt and discharge generation in a snow dominated mountain catchment (Sect. 5.2).

**5.1   1-D sensitivity studies**

**5.1.1   Sensitivity to surface layer thickness**

To investigate the impact of the maximum surface layer thickness of the model, we run simulations with synthetic forcing and use maximal surface layer thicknesses of 4.0 mm SWE (max estimate, see Tabel 2), 8.0 mm SWE (mid estimate), 16.0 mm SWE (min estimate). Additionally we include a single layer model with a vertically uniform distribution of BC in the analysis and for comparison a simulation with clean snow. Since the model input used in the sensitivity study during the melt period does exclude fresh snowfall and dry deposition, increases in surface BC mixing ratio is due to melt amplification solely. Fig. 3a shows the effect of the different maximum surface layer thicknesses on the melting snowpack, with mid-estimates for further model parameters according to Table 2. The maximum surface layer thickness strongly determines the surface BC mixing ratio over the melt season. During snow melt, surface BC increases up to a factor of circa 10, 20 and about 30 for maximum surface layer thicknesses of 16.0 mm SWE, 8.0 mm SWE, and 4.0 mm SWE, compared to the pre-melt season BC mixing ratio (35 ng g$^{-1}$). ~~Since the model input used in the sensitivity study during the melt period does exclude fresh snowfall and dry deposition, the increase in surface BC mixing ratio is due to melt amplification solely. The importance of BC accumulation in surface snow is discussed controversially in the literature. While several studies report a significant increase in surface BC mixing ratio during melt (Doherty et al., 2013; Sterle et al., 2013) of up to an order of magnitude (Sterle et al., 2013) and more (Xu et al., 2012) , others report highly efficient scavenging with melt (Lazarcik et al., 2017) . Over most of the melt period, our results show a factor increase between 5 and 15. Only at the end of the melt season, higher factor increases are reached. To this point of time, however, the snowpack is typically very thin and effects on discharge generation due to very high increase in surface BC should be small.~~

For the three 2-layer scenarios (green, purple and red curves in Fig. 3a), the resulting difference on the albedo and melt rate are small, even though the increase in surface layer mixing ratio during the melt season differs strongly among the scenarios. The relatively small differences in snowpack evolution among the two-layer models, despite the large differences in surface BC, result from the fact that for all two-layer models the surface layer thickness is much thinner than the penetration depth of shortwave radiation. For example, in clean snow with an optical grain size of 50 um, the radiative intensity diminishes to $\frac{1}{e}$ of its surface value (the so called penetration depth) in 25.5 mm SWE. For snow with an optical grain size of 1000 um, the penetration depth increases to 117 mm SWE (both results from Flanner et al., 2007, assuming a wavelength of 550 nm and a solar zenith angle of 60°). Thus, BC in the surface layer absorb efficiently in all 2-layer scenarios and the difference in the albedo is relatively large compared to the no-ARF scenario (solid black line in top graph of Fig. 3a), but relatively small among the two-layer scenarios (solid green, purple, and red line in top graph of Fig. 3a). This is a critical difference when a single layer model is used (solid yellow lines in Fig. 3a). With only one layer, aerosol is distributed uniformly over the snowpack. The BC concentration is slowly increasing, however, it stays comparably low in contrast to the two-layer models  until shortly before meltout (solid yellow line in the center graph of Fig. 3a). Due to the uniform distribution of BC in the single layer model, a large fraction of the BC is located at depths where the radiative

intensity is much lower than in the top few mm of the snowpack, leading to a weaker absorption efficiency. This leads to a less pronounced decrease of albedo compared to the two layer models (solid yellow line in the top graph of Fig. 3a) and thus to a shorter meltout shift compared to a clean snowpack than in the 2-layer scenarios (about five days).

Observations of BC in melting snow support the accumulation of BC near the surface (Xu et al., 2012; Doherty et al., 2013; Sterle et al., 2013). In a sequence of snow pits, Sterle et al. (2013) showed that during the ablation season, BC mixing ratios increase significantly near the snow surface (sampled in the top two centimeter) relative to bulk BC concentrations. They suggest that most likely a large fraction of previously deposited BC becomes concentrated near the surface. Delaney et al. (2015) also report of surface BC increase during melt, to which BC being trapped at the snow surface is likely to contribute. BC increase in surface snow of up to an order of magnitude (Sterle et al., 2013; Doherty et al., 2016) and more (Xu et al., 2012) have been observed in natural snow during melt. This aligns reasonably well with the here presented evolution of BC in the surface layer. Over most of the melt period, our results show a factor increase between 5 and 15 for the 2-layer scenarios. Higher values are mainly predicted shortly before meltout, when the snowpack is typically very thin and effects on discharge generation due to high increase in surface BC should be small.

The results presented herein demonstrate that simulating BC accumulation near the snow surface using a thin surface layer (2-layer model) can have a significant impact on the albedo compared to a model that does not resolve near surface processes (single layer model). Furthermore, by varying the model's maximum surface layer, we show that simulated surface mixing ratios of BC are highly sensitive to this model parameter. Since evaluation of model predictions for BC in snow is commonly performed by comparing simulated with observed BC mixing ratios in surface snow (e.g., Flanner et al., 2007; Forsström et al., 2013) , this is a critical result. Snow is often sampled in top few centimeters (typically 2 to 5 cm, e.g., Doherty et al., 2010; Aamaas et al., 2011; Forsström et al., 2013). This raises an interesting challenge give that the surface layer assumed in models is not a measurable property of snow. A comparison of model simulations with observations should therefore include some quantification of the uncertainty resulting from the layer thickness parametrization.

The sensitivity study using different values for the maximum surface layer thickness provides three important results. First, when the properties of the included LAISI are prone to melt amplification (scavenging ratio below 1), a minimum of two layers is required to simulate the effect of efficient absorption resulting from LAISI located close to the snow surface. Second, the surface layer thickness  and the connected surface BC evolution plays only a minor role for the effect on the albedo, as long as the assumption that the surface layer thickness is much smaller than the penetration depth of shortwave radiation into the snowpack is justifiable. Third, by varying the surface layer thickness in a reasonable range, we cover a large range of BC increase in surface snow during melt, yet the effect on albedo, snow melt and snowpack evolution is minimal. ~~Observed LAISI concentrations often are sampled in the top few centimetres of the snowpack and compared to surface layer concentration of models (e.g., Flanner et al., 2007; Forsström et al., 2013) , even though the surface layer is not a measurable snow property. Our results show that the comparison of observed surface concentrations with simulations is critical due to the large impact of the model surface layer thickness on the surface concentration - while the effect on key snowpack variables such as the snow albedo remain nearly unaffected. This highlights the need for including a surface layer variation in the uncertainty estimation of the comparison with snow sampled in the surface layer.~~

**5.1.2   Sensitivity to scavenging ratio of BC**

Field measurements indicate that only a fraction of BC is flushed out with the melt water and BC can accumulate near the snow surface (e.g., Xu et al., 2012; Doherty et al., 2013; Sterle et al., 2013; Doherty et al., 2016). Our model is able to simulate this process by taking the scavenging ratio of BC during meltwater movement into account (Eq. 8 and 9). In this

5   section we explore the scavenging processes further, by investigating the impact of different BC scavenging ratios on the snowpack evolution.  The scavenging ratio applied for hydrophobic BC (0.03) is based on analysis conducted by Flanner et al. (2007) using data from Conway et al. (1996) . The same accounts for the applied hydrophilic BC scavenging ratio (0.2), which also compares well to field observations from Doherty et al. (2013) . We further include Flanner et al. (2007) 's upper bound uncertainty estimate for hydrophilic BC (2.0; efficient scavenging) in the analyses, and for comparison a scenario

10   in which BC does not undergo any scavenging (0.0). In the range of investigated scavenging ratios, we find sensitivity of the BC surface mixing ratio, the albedo, and the subsequent snow melt to this parameter (Fig. 3b). When applying a melt scavenging factor typical for hydrophobic BC ( purple lines in graphs of Fig. 3b) there is little effect compared to the scenario without melt scavenging ( green lines; both show circa a factor 30 increase in surface BC concentration to the end of the melt season and only little differences in the development of albedo and snow melt). However, a distinction exists when using a

15   scavenging ratio estimate for hydrophilic BC. In contrast to  no scavenging and hydrophobic  scavenging, surface BC does not increase as rapidly during the melt period (red line, central graph of Fig. 3b) and in fact is completely flushed when applying the max-estimate of hydrophilic scavenging (yellow line).

    The changes in the scavenging ratio lead to a considerable effect on the albedo and the snow melt (meltout delayed by circa  0.5 (purple lines),  3 (red lines), and  8 days (yellow lines) for scavenging ratios of 0.03, 0.2, and 2.0, respectively,

20   compared to no  scavenging (green lines in Fig. 3b)). Compared to the no-ARF experiment (black lines), the presence of BC  causes an earlier meltout of circa  9.5, 7, and 2 days for scavenging ratios of 0.03, 0.2, and 2.0, respectively, in our simulation. This implies a significant effect of BC on the albedo in all scenarios applied.  When the melt scavenging is set to the upper limit (2.0; yellow lines in graphs of Fig. 3b), the surface concentration drops continuously during the melt period due to the highly efficient melt scavenging. As a consequence, the albedo converges

25   against the albedo of the no-ARF case, before it drops roughly  two days earlier to a value of circa 0.2 due to the earlier exposure of the underling ground (solid yellow and black line in top graph of Fig. 3b). ~~The slight increasing in difference in the melt rate between the no-ARF and the upper scavenging during the first 7 days of melt are due to the increasing absorption efficiency of BC with increasing optical snow grain size (e.g., Flanner et al., 2007) . The following convergence (day 7 until 17 from melt onset) of both melt rates are due to the decreasing BC concentration in the upper scavenging scenario due to ongoing~~

30    Even though nearly all BC is removed from the snow by the end of the melt period, the melt out still happens circa two days earlier compared to the no-ARF experiment.

In the literature, the scavenging efficiency of BC is discussed controversially. Flanner et al. (2007) 's estimates for scavenging ratios of hydrophilic and hydrophobic BC, which are used in this study, are based on data from field experiments using artificially added soot (Conway et al., 1996) . Parameters derived from artificially added soot might not be directly transferable to the scavenging properties of naturally occurring BC. Even though field observations from Doherty et al. (2013) agree well with the estimates of Flanner et al. (2007) , and further studies highlight the importance of BC retention in the snow pack (e.g., Xu et al., 2012; Sterle et al., 2013) , a large uncertainty remains on the magnitude of this effect (Lazarcik et al., 2017) . For this reason, we use a factor of 10 in the min and max effect estimates in the case study of Sect. 5.2 (see Table 2). The results presented herein show large differences in snowpack response in the boundaries of these uncertainties. This reveals the need for more detailed experimental and observational insight in order to reduce uncertainties. However, our results further show that that even if BC undergoes efficient scavenging, the impact on snowpack evolution can still be significant.

**5.1.3 Sensitivity to BC species**

Hydrophilic BC absorbs stronger than hydrophobic BC under the same conditions due to an increased MAC compared to hydrophobic BC caused by the ageing of BC during atmospheric transport (Bond et al., 2006). On the other hand, as we previously explored, hydrophilic BC undergoes more efficient melt scavenging (Flanner et al., 2007), which impacts the snowpack evolution significantly. The column of graphs in Fig. 3c illustrates the net effect of these competing processes by applying the mid estimate of the scavenging ratio of hydrophobic BC (0.03) to both the hydrophobic BC (green curve) and the hydrophilic BC (purple curves) species. In this manner these curves show the isolated effect of the different absorption properties of the two species. We further apply the mid estimate for hydrophilic BC scavenging ratio (0.2) to hydrophilic BC (red curves) to quantify the gross effect. As in other cases, we include the no-ARF scenario (black curves) to highlight the overall effect on the albedo and melt of the different scenarios.

The isolated effect of the stronger absorption of hydrophilic BC leads to an earlier meltout by circa two days compared to hydrophobic BC (purple and green curves in graphs of Fig. 3c). However, when applying the mid estimate of the scavenging ratio for hydrophilic BC (0.2), the combined effects leads to a masking of the isolated effect of stronger absorption by hydrophilic BC (and vice versa). During the melt period, snow albedo, melt rate and the snowpack SWE barely differ between the scenarios with the mid estimate scavenging for hydrophobic and hydrophilic BC applied (red and green curves in top and bottom graphs of Fig. 3c). This reveals that both scenarios, hydrophobic BC with low scavenging efficiency and hydrophilic BC with high scavenging efficiency, lead roughly to an earlier meltout by circa 6 days. We interpret this that a clear distinction between the both species might play a secondary role in the determination of the overall impact of BC on snow melt.

**5.1.4 Sensitivity to snowpack SWE at melt onset**

In the following we explore the shortening of the melt period duration of snowpacks with constant BC mixing ratio at melt onset but different SWE relative to clean snowpacks with similar SWE. Results are shown in Fig. ?? for different scavenging ratios. Apart from SWE and scavenging ratio, all initial snowpack properties and the model input data are the same among the different scenarios, including an initial BC concentration of 35 ng g$^{-1}$. BC is distributed uniformly throughout the entire

snowpack at melt onset. With respect to the range of snowpack SWE at melt onset presented here, the melt period shortening is stronger the smaller the scavenging ratio applied, and increasing with increasing SWE at melt onset. Results show a melt period reduction of up to 30for the mid-estimate hydrophilic BC scavenging, end even higher when applying the mid-estimate hydrophobic BC scavenging.

With increasing SWE at melt onset, the increase in melt period shortening gets less pronounced (dashed and dashed-dotted curves in Fig. **??**), and differences between the melt scavenging scenarios become larger. When applying very efficient melt scavenging (dotted curve in Fig. **??**), the effect on the reduction is smallest over the range of SWE values shown, however, still leading to a melt period shortening between 4-8.

The results suggest that not only the BC concentration and distribution, the snow properties, and the radiative properties and hydrophobicity of the aerosol control how BC in snow impacts the melt. The amount of snow accumulated also plays an important role, with thicker snowpacks and similar LAISI mixing ratio at melt onset showing a stronger response to the LAISI induced processes.

[revised manuscript text omitted]

5 ~~For the min- and mid-scenario the model simulates an average annual surface BC mixing ratio of about 18 ng g$^{-1}$ and 71 ng g$^{-1}$, respectively. Forsström et al. (2013) found for mainland Scandinavia values of the same magnitude, with seasonal means for different measurement locations and time periods ranging from about 10 ng g$^{-1}$ to 80 ng g$^{-1}$. This places our results well within those presented in Forsström et al. (2013) . Our max-scenario yields 198 ng g$^{-1}$ which lies above average values expected from Forsström et al. (2013) . However, Flanner et al. (2007) evaluated the global impact of the radiative forcing of~~
10

 We recognize our max-scenario results in a strong increase in surface BC mixing ratios mostly due to low BC scavenging with melt (note the strong increase from end of March on in Fig. 6). This divergent evolution of surface BC mixing
15 ratios in the min, mid, and max scenarios reveals uncertainty in the representation of the fate of BC in snow during melt. This uncertainty is also reflected in the literature. On the one hand, some studies report of high accumulation of BC in surface snow with implications for snow melt. Doherty et al. (2013) reported a factor 5 increase in surface BC mixing ratio under melt conditions, and in a subsequent study found increases of over an order of magnitude (Doherty et al., 2016). These findings were similar to Xu et al. (2012) who also found post-depositional enrichment of BC in surface snow over an order of magnitude.
20 On the other hand, Lazarcik et al. (2017) observe efficiently scavenged BC, leading to decreased surface mixing ratios. In fact, they report BC leaching from the snow more rapidly than the snow melt and summarize that that surface enrichment of BC is not linked to SWE decreases during melt. The large differences in the evolution of surface BC mixing ratio during melt in our study reflects the range of uncertainties shown in previous studies. However, while the surface BC mixing ratio evolution during melt for min- and mid-scenario is within reason, it appears our max-scenario  is very likely
25 to overestimate melt amplification.

**5.2.3   BC induced radiative forcing**

The radiative forcing in snow (RFS) induced by the presence of BC is calculated from the average radiative forcing over snow bearing tiles only. The RFS represents the additional uptake of energy from solar radiation per area snow cover due to the presence of BC in the snow compared to clean snow with the same properties. Fig. 7a shows the daily mean RFS and
30 demonstrates the increase effect of RFS during snow melt. Low RFS is observed during the snow accumulation period then steadily increasing through spring snow melt, reaching values of approximately 8, 18, and 57 Wm$^{-2}$ for the min, mid, and max scenarios, respectively (see red solid line and shaded area in Fig. 7a). The strong increase in RFS during spring melt results from the combination of: (i) the decrease in snow albedo due to the increase in surface BC concentrations (e.g. melt

amplification and the increasing optical grain size in melting snow as discussed in Sect. 5.2.2) and, (ii) the increasing daily solar irradiation due to a lower solar zenith angle and longer days.

However, most relevant for discharge generation (see Sect. 5.2.4), is the catchment-wide total daily energy uptake due to BC, calculated as the mean radiative forcing over all grid cells. As the snow cover fraction (SCF) in the catchment drops during

5   spring (dotted line and yellow shaded area in Fig. 6 and 7), the effect of the RFS on the melt generation is limited by the increasing area of bare ground. The net effect is shown in Fig. 7b. The catchment mean daily energy uptake due to the presence of BC in snow shows a strong annual cycle and reaches a maximum of 1.3, 4.9, and 8.8 $Wm^{-2}$ (min, mid, and max scenario, respectively) around the beginning of May. Radiative forcing in mid winter is small due low surface BC mixing ratios and low solar irradiate. (Qian et al., 2011) also reports a similar strong annual cycle with values in the same range for BC radiative

10   forcing over the Tibetan Plateau using a global climate model, but with higher values in winter time. Annual mean values are 0.284, 0.844, and 1.391 $Wm^{-2}$ for the min, mid, and max scenario. Averaged over entire Scandinavia (including Finland), Hienola et al. (2016) calculated lower values around 0.145 $Wm^{-2}$. However, Hienola et al. (2016) study includes large areas with shorter snow cover. Since the value is strongly depended on the snow cover evolution, higher values compared to Hienola et al. (2016) are expected due to the long lasting snow cover in our case study region.

15   **5.2.4   BC impact on catchment discharge and snow storage**

Fig. 8a shows the simulated daily discharge and catchment SWE averaged over the 6 years simulation period for the mid (red lines); min and max estimates (bounds of the shaded areas); and the no-ARF scenario (black lines). The differences in daily discharge and catchment SWE of the min, mid, and max scenarios to the no-ARF scenario are shown in Fig. 8b. All simulations with ARF show higher daily discharge from end of March until end of May and lower discharge from end of May until mid

20   August relative to the no-ARF simulation. For the rest of the year, no effect on the discharge is noticeable. The net impact of RFS results in a shift in the timing of discharge. Higher discharge early in the melt season is observed, yet offset by lower discharge following May. The cumulative annual discharge remains nearly identical.

Min, mid, and max scenarios all show the change from higher to lower discharge compared to the no-ARF scenario approximately at the same time (at the end of May; see blue marker in Fig. 8b). Therefore, we can quantify the absolute and relative

25   effect of RFS on the discharge during the two periods: the early melt season from circa March 22 until May 29 and the late melt season from circa May 30 until August 10 (Fig. 8b and see Table 3). This yields an average percentage increase in daily discharge of 2.5 %, 9.9 % and 21.4 % for the min, mid, and max scenario for the early melt season and a decrease in discharge of -0.8 %, -3.1 %, and -6.7 % during the late melt season.

The differences in discharge among the scenarios can be explained by understanding the evolution of the snowpack. In the

30   all scenarios the catchment SWE (Fig. 8a) reaches a peak reduction relative to the no-ARF scenario of -4.6 %, -13.4 % and -34.4 % at mid May. The average difference in catchment SWE of the min, mid, and max scenarios compared to the no-ARF scenario during the entire melt season is -1.5, -5.1, and -10.3 mm; or an average of 2.1 %, 7.4 %, and 15.1 % (see Table 3). From mid May on, the differences in catchment SWE between scenarios drop continuously, which is equivalent to a higher catchment averaged snow melt rate in the no-ARF scenario compared to the ARF scenarios.

The difference at the beginning of the melt season can be attributed to RFS. However, from mid May on we see a decrease in the differences in catchment SWE between the ARF and no-ARF scenarios (Fig. 8b). To understand this counter-intuitive result, we need to evaluate the impact of BC deposition at the catchment scale. The dynamics driven by the SCF of the catchment is a limiting factor to the catchment averaged snow melt. Looking at Fig. 6a we see the development of average snow albedo and the SCF in the catchment. During the melt period, the catchment averaged albedo in all of the scenarios, decreases (and RFS is continually increasing). Intuitively, one would expect more melting due to enhanced solar radiative forcing. However, the SCF decrease with increased melt due to ARF counteracts the RFS effect itself, due to the reduction in area from which snow can actually melt. This is also indicated in Fig. 7b, where the additional energy uptake due to BC in snow peaks in the beginning of May. We can see the same result for the discharge: the increased discharge of the ARF scenarios during the beginning of the melt season may simply be attributed to RFS, whereas the decreased discharge later in the season is attributed to melt limitation caused by the SCF retreat.

Compared to observations, all simulations (ARF and no-ARF) tend to underestimate discharge during early melt season and overestimate discharge during late melt season (Fig. 8a). However, the magnitude of over- and underestimation strongly differs between the scenarios. By including ARF the volume error is reduced in both the early melt season (by increasing melt), and in late melt season (by subsequently decreasing melt generation in the catchment due to reduced SCF). Expressed as seasonal mean volume error for early and late melt season, the difference to observed discharge is largest for the no-ARF scenario and smallest for max scenario. The max scenario reduces the volume error by -75.1% during early melt season and -89.9% during late melt season, relative to the no-ARF scenario (see Table 4). The min and mid scenarios also reduce the volume error. Thus, on average, an improvement in simulated discharge is achieved during the melt season by accounting for BC RFS.

**5.3 Uncertainties**

Both the literature and our analysis demonstrates numerous uncertainties and we urge further studies to address RFS-induced uncertainty. In our model study, uncertainties result principally from uncertainty of the mixing ratio of BC in the snowpack due to:

[revised manuscript text omitted]

---

## Author Response (AR3)

**Authors' response to editor's comments**

**Thanks for the continued efforts revising this potentially very interesting manuscript. To be fully honest, I have to say that the revisions in this and especially also the previous rounds are a bit on the minimalistic side. There are several comments where more substantial changes and/or new computations would be possible/desired.**

During prior reviews we received constructive feedback regarding the structure of the manuscript and readability. Referee #3 has provided positive feedback and noted improved readability. However, we acknowledge that certain aspects of the manuscript benefit from further restructuring and a deeper analysis of uncertainty. We feel confident we have addressed the editors concerns in this round of review. Changes to the manuscript include:

- splitting of results/discussion section into separate sections (Sect. 5 and 6) per the editor's recommendation,
- the addition of new calculations concerning estimates of radiative forcing in snow from BC mixing ratios found in Scandinavian snowpacks (see Sect. 6.2.2.),
- a more critical discussion regarding the model improvement which results from accounting for BC radiative forcing and associated uncertainty (see Sect. 6.2.3. and 6.3), and importantly,
- a completely new analysis of uncertainty using the GLUE approach assessing uncertainty of model predictions coming from the parameter space prior and posterior to the usage of radiative forcing calculations from LAISI (see Sect. 6.2.3 and 6.3).

**There are still multi-letter variable names (Eq 5, 6, 10), please follow the author guidelines.**

We have replaced the multi-letter variables.

**All reviewers state that the manuscript is hard to read. Splitting results and discussion would clearly help to make the manuscript more readable (if you do not believe me, almost all scientific writing advice books recommend this for 'normal' journal papers!)**

We have restructured the manuscript according to the Editor's recommendation, i.e. split the former results/discussion section into separate sections of results and discussion (now Sect. 5 and 6).

**"the efficiency of the model when compared to real discharge data might not be accurate" The point raised by the reviewer here is that there is a clear risk of being right for the wrong reason (see J. Kirchner, 2006, WRR). Since you want to make the point that including BC leads to better simulations it is crucial that the rest of the model works for the right reasons. Otherwise the improved fit might just be because some other error is compensated.**

In order to address this, we have conducted new simulations to estimate uncertainty coming from the model parameters using the GLUE method (Beven, 1992). We have included an extended discussion about the problem of being right for the wrong reason, i.e. discussion potential implications from structural deficits of the model. It should be noted that while we do find in our case study that the simulations were improved when incorporating BC, we recognize that better simulations do not always result from increased complexity and there is an appropriate time and place to apply such complexity. Our main intent is not to argue this is essential for improved forecasting, but rather to provide a mechanism to address the potential impacts of LAISI in a more robust manner than presently available today.

**While it is good that model uncertainties now are better discussed than in previous versions,**

**the reviewer comments actually would have motivated more new calculations. Especially a quantification of uncertainties would be useful. (see Pappenberger and Beven, 2006)**

As described above, we have quantified uncertainties prior and posterior using radiative forcing from LAISI calculations due to model parameter uncertainty – a large source of uncertainties in conceptual modelling.  We agree that this analysis, in conjunction with the refined discussion about further uncertainties, further improves the quality of the manuscript and appreciate the encouragement of the editor to include such an analysis.

References

[revised manuscript text omitted]
. Since the model input used in the sensitivity study during the melt period does exclude fresh snowfall and dry deposition, increases in surface BC mixing ratio is due to melt amplification solely., with mid-estimates for further model parameters~~ with other parameters set according to Table 2. The maximum surface layer thickness strongly determines the surface BC mixing ratio over the melt season. During snow melt, surface BC increases up to a factor of circa 10, 20, and about 30 for maximum surface layer thicknesses of 16.0 mm SWE, 8.0 mm SWE, and 4.0 mm SWE, compared to the pre-melt season BC mixing ratio (35 ng g$^{-1}$).

 For those three 2-layer scenarios (green, purple and red curves in Fig. 3a), the resulting  differences in albedo and melt rate are small, even though the increase in surface layer mixing ratio during the melt season differs strongly among the scenarios. ~~The relatively small differences in snowpack evolution among the two-layer models, despite the large differences in surface BC, result from the fact that for all two-layer models the surface layer thickness is much thinner than the penetration depth of shortwave radiation. For example, in clean snow with an optical grain size of 50 um, the radiative intensity diminishes to $\frac{1}{e}$ of its surface value (the so called penetration depth) in 25.5 mm SWE. For snow with an optical grain size of 1000 um, the penetration depth increases to 117 mm SWE (both results from Flanner et al., 2007 , assuming a wavelength of 550 nm and a solar zenith angle of 60°). Thus, BC in the surface layer absorb efficiently in all 2-layer scenarios and the difference in the albedo is relatively large compared to the no-ARF scenario (solid black line in top graph of Fig. 3a), but relatively small among the two-layer scenarios (solid green, purple, and red line in top graph of Fig. 3a). This is a critical difference when ais used (solid yellow lines in Fig. 3a). With only one layer, aerosol is distributed uniformly over the snowpack. The BC concentration is slowly increasing, however, ittwo-layer(solid yellow line in the center graph of Fig. 3a). Due to the uniform distribution of BC in the single layer model, a large fraction of the BC is located at depths where the radiative intensity is much lower than in the top few mm of the snowpack, leading to a weaker absorption efficiencytwo layer models (solid yellow line in the top graph of Fig. 3a)than in the 2-layer scenarios (about five days ).~~

alarge fraction of previously deposited BC becomes concentrated near the surface. Delaney et al. (2015) also report of surface BC increase during melt, to which BC being trapped at the snow surface is likely to contribute. BC increase in surface snow of up to an order of magnitude (Sterle et al., 2013; Doherty et al., 2016) and more (Xu et al., 2012) have been observed in natural snow during melt. This aligns reasonably well with the here presented evolution of BC in the surface layer. Over most of the melt period, our results show a factor increase between 5 and 15 for of about 5 days (yellow curves in Fig. 3a), whereas the 2-layer scenarios . Higher values are mainly predicted shortly before meltout, when the snowpack is typically very thin and effects on discharge generation due to high increase in surface BC should be small.

The results presented herein demonstrate that simulating BC accumulation near the snow surface using a thin surface layer (2-layer model) can have a significant impact on the albedo compared to a model that does not resolve near surface processes (single layer model). Furthermore, by varying the model's maximum surface layer, we show that simulated surface mixing ratios of BC are highly sensitive to this model parameter. Since evaluation of model predictions for BC in snow is commonly performed by comparing simulated with observed BC mixing ratios in surface snow (e.g., Flanner et al., 2007; Forsström et al., 2013) , this is a critical result. Snow is often sampled in top few centimeters (typically 2 to 5 cm, e.g., Doherty et al., 2010; Aamaas et al., 2011; For This raises an interesting challenge give that the surface layer assumed in models is not a measurable property of snow. A comparison of model simulations with observations should therefore include some quantification of the uncertainty resulting from the layer thickness parametrization.

The sensitivity study using different values for the maximum surface layer thickness provides three important results. First, when the properties of the included LAISI are prone to melt amplification (scavenging ratio below 1), a minimum of two layers is required to simulate the effect of efficient absorption resulting from LAISI located close to the snow surface. Second, the surface layer thickness and the connected surface BC evolution plays only a minor role for the effect on the albedo, as long as the assumption that the surface layer thickness is much smaller than the penetration depth of shortwave radiation into the snowpack is justifiable. Third, by varying the surface layer thickness in a reasonable range, we cover a large range of BC increase in surface snow during melt, yet the effect on albedo, snow melt and snowpack evolution is minimalshow earlier meltouts of about 7 days.

**5.1.2   Sensitivity to scavenging ratio of BC**

Field measurements indicate that only a fraction of BC is flushed out with the melt water and BC can accumulate near the snow surface (e.g., Xu et al., 2012; Doherty et al., 2013; Sterle et al., 2013; Doherty et al., 2016) . Our model is able to simulate this process by taking the scavenging ratio of BC during meltwater movement into account (Eq. 8 and 9). In this section we explore the scavenging processes further, by investigating the impact of different BC scavenging ratios on the snowpack evolution. The scavenging ratio applied for hydrophobic BC (0.03) is based on analysis conducted by Flanner et al. (2007) using data from Conway et al. (1996) . The same accounts for the applied hydrophilic BC scavenging ratio (0.2), which also compares well to field observations from Doherty et al. (2013) . We further include Flanner et al. (2007) 's upper bound uncertainty estimate for hydrophilic BC (2.0; efficient scavenging) in the analyses, and for comparison a scenario in which BC does not undergo any scavenging (0.0). In the range of investigated scavenging ratios, we find sensitivity of the BC surface surface BC mixing

ratio, the albedo, and the subsequent snow melt to this parameter (Fig. 3b). When applying a melt scavenging factor typical for the lower bound of hydrophilic BC (0.02, purple lines) there is little effect compared to the scenario without melt scavenging (green lines). Both show circa a factor 30 increase in surface BC concentration to the end of the melt season and only little differences in the development of albedo and snow melt. Similar results are achieved when using the mid estimate scavenging factor for hydrophobic BC (0.03, not shown). A distinction exists when using  the mid estimate scavenging factor for hydrophilic BC (0.2, red line). In contrast to no scavenging and  the lower bound hydrophilic scavenging, surface BC does not increase as rapidly during the melt period  and in fact is completely flushed when applying  a melt scavenging factor typical for the upper bound of hydrophilic BC (yellow line, the surface concentration drops continuously during the melt period).

The changes in the scavenging ratio lead to a considerable effect on the albedo and the snow melt. Meltout is delayed by circa 0.5 (purple lines), 3 (red lines), and 8 days (yellow lines) for scavenging ratios of 0.03, 0.2, and 2.0, respectively, compared to no scavenging (green lines). Compared to the no-ARF experiment (black lines), the presence of BC causes an earlier meltout of circa 9.5, 7, and 2 days for scavenging ratios of 0.03, 0.2, and 2.0, respectively, in our simulation. ~~This implies a significant effect of BC on the albedo in all scenarios applied. When the melt scavenging is set to the upper limit (2.0; yellow lines in graphs of Fig. 3b), the surface concentration drops continuously during the melt period due to the highly efficient melt scavenging. As a consequence, the albedo converges against the albedo of the no-ARF case, before it drops roughly two days earlier to a value of circa 0.2 due to the earlier exposure of the underling ground (solid yellow and black line in top graph of Fig. 3b). Even though nearly all BC is removed from the snow by the end of the melt period, the melt out still happens circa two days earlier compared to the no-ARF experiment.~~

~~In the literature, the scavenging efficiency of BC is discussed controversially. Flanner et al. (2007) 's estimates for scavenging ratios of hydrophilic and hydrophobic BC, which are used in this study, are based on data from field experiments using artificially added soot (Conway et al., 1996) . Parameters derived from artificially added soot might not be directly transferable to the scavenging properties of naturally occurring BC. Even though field observations from Doherty et al. (2013) agree well with the estimates of Flanner et al. (2007) , and further studies highlight the importance of BC retention in the snow pack (e.g., Xu et al., 2012; Sterle et al., 2013) , a large uncertainty remains on the magnitude of this effect (Lazarcik et al., 2017) . For this reason, we use a factor of 10 in the min and max effect estimates in the case study of Sect. 5.2 (see Table 2). The results presented herein show large differences in snowpack response in the boundaries of these uncertainties. This reveals the need for more detailed experimental and observational insight in order to reduce uncertainties. However, our results further show that that even if BC undergoes efficient scavenging, the impact on snowpack evolution can still be significant.~~

**5.1.3 Sensitivity to BC species**

[revised manuscript text omitted]